**Holocene climates of the Iberian Peninsula: pollen-based reconstructions of changes in**
**the west-east gradient of temperature and moisture**
Mengmeng Liu[1,*], Yicheng Shen[2], Penelope González-Sampériz[3], Graciela Gil-Romera[3],
Cajo J. F. ter Braak[4], Iain Colin Prentice[1], Sandy P. Harrison[2]
1: Department of Life Sciences, Imperial College London, Silwood Park Campus, Buckhurst
Road, Ascot SL5 7PY, UK
2: Geography & Environmental Science, Reading University, Whiteknights, Reading, RG6
6AH, UK
3: Instituto Pirenaico de Ecología-CSIC, Avda. Montañana 1005, 50059, Zaragoza, Spain
4: Biometris (Applied Mathematics and Applied Statistics Centre), Wageningen University &
Research, 6708 PB Wageningen, The Netherlands
*: Corresponding author: Mengmeng Liu (m.liu18@imperial.ac.uk)
Ms for: *Climate of the Past*
**Abstract**
The Iberian Peninsula is characterised by a steep west-east moisture gradient today, reflecting
the dominance of maritime influences along the Atlantic coast and more Mediterranean-type
climate further east. Holocene pollen records from the Peninsula suggest that this gradient was
less steep during the mid-Holocene, possibly reflecting the impact of orbital changes on
circulation and thus regional patterns in climate. Here we use 7214 pollen samples from 117
sites covering part or all of the last 12,000 years to reconstruct changes in seasonal temperature
and in moisture across the Iberian Peninsula quantitatively. We show that there is an increasing
trend in winter temperature at a regional scale, consistent with known changes in winter
insolation. However, summer temperatures do not show the decreasing trend through the
Holocene that would be expected if they were a direct response to insolation forcing. We show
that summer temperature is strongly correlated with plant-available moisture (α), as measured
by the ratio of actual evapotranspiration to equilibrium evapotranspiration, which declines
through the Holocene. The reconstructions also confirm that the west-east gradient in moisture
was considerably less steep than today during the mid-Holocene, indicating that atmospheric
circulation changes (possibly driven by orbital changes) have been important determinants of
the Holocene climate of the region.

## 1. Introduction

The Iberian Peninsula is characterised by a steep west-east gradient in temperature and moisture today, reflecting the dominance of maritime influences along the Atlantic coast and more Mediterranean-type climate further east. Projections of future climate change suggest that the region will become both warmer and drier, but nevertheless show that this west-east differentiation is maintained (Andrade et al., 2021a). The changes in temperature are projected to be larger and the occurrence of extreme temperature episodes more frequent in the south-central and eastern parts of Iberia than in Atlantic coastal areas (Carvalho et al., 2021). Similar gradients are seen in future projections of precipitation change, with largest reductions in precipitation in the south-central region (Andrade et al., 2021b). However, the stability of these west-east gradients during the Holocene has been questioned. In particular, the west-east gradient in moisture appears to have been less pronounced during the mid-Holocene (8~4 ka) when cooler summers and wetter conditions in the Atlantic zone (e.g. Martínez-Cortizas et al., 2009; Mauri et al., 2015) coincided with the maximum development of mesophytic vegetation further east and south (Aranbarri et al., 2014, 2015; Carrión et al., 2010, 2009; González-Sampériz et al., 2017).

However, much of the evidence for Holocene climates of the Iberian Peninsula is based on qualitative interpretations of vegetation changes, generally interpreted as reflecting changes in moisture availability (Morellón et al., 2018; Ramos-Román et al., 2018; Schröder et al., 2019). These records are extensive and they seem to indicate fairly complex spatial patterns of change. Kaufman et al. (2020) provides quantitative reconstructions of summer and winter temperature in their compilation of Holocene climate information, but there are only 5 terrestrial sites from the Iberian Peninsula. Iberia was also included in the quantitative pollen-based reconstructions of European climate through the Holocene in Mauri et al. (2015), which is an update of Davis et al. (2003). However, the geographical distribution of sites included is uneven and a large fraction of the records were from the Pyrenees and the Cantabrian mountains, with additional clustering of sites in coastal regions. Thus, the inferred patterns of climate over most of the central part of the Peninsula are therefore largely extrapolated. Tarroso et al. (2016) has provided reconstructions of summer and winter temperature and mean annual precipitation since the Last Glacial Maximum for the Iberian Peninsula, by using modern species distribution data to develop climate probability distribution functions (PDFs) and applying these to 31 fossil records. However, although they identified trends in precipitation during the Holocene, the

temperature reconstructions do not seem to be reliable since they show no changes through time (9~3 ka), either for the Iberian Peninsula as a whole or for individual sub-regions, in contradiction to the other reconstructions. The current state of uncertainty about Holocene climate changes in Iberia is further exacerbated because quantitative reconstructions of summer temperature made at individual sites using chironomid data (Muñoz Sobrino et al., 2013; Tarrats et al., 2018) are not consistent with reconstructed summer temperatures based on pollen for the same sites.

We used the method Tolerance-weighted Weighted Average Partial Least-Squares regression with a sampling frequency correction (fxTWA-PLS), introduced by Liu et al. (2020) as an improvement of the widely used Weighted Average Partial Least-Squares (WAPLS: ter Braak and Juggins, 1993) method for reconstructing past climates from pollen assemblages. As presented in depth by Liu et al. (2020), this method is a more complete implementation of the theory underlying WA-PLS because it takes greater account of the climatic information provided by taxa with more limited climatic ranges and also applies the sampling frequency correction to reduce the impact of uneven sampling in the training data set. Liu et al. (2020) showed that fxTWA-PLS does indeed provide better reconstructions than WA-PLS.

Here we have further modified the algorithm implementing fxTWA-PLS, achieving an additional gain in performance. In the algorithm as published by Liu et al. (2020), sampling frequencies were extracted from a histogram. In the modified algorithm they are estimated using P-splines smoothing (Eilers and Marx, 2021), which makes the estimates almost independent on the chosen bin width (see Appendix A for details). In addition, the modified method applies the sampling frequency correction at two separate steps – the estimation of optima and tolerances, and the regression step – a measure intended to produce more stable results. Indeed, the modified method produces both improved $R^2$ values and reduced compression and maximum bias in reconstructed climate variables (see Table A1 and Figs A1–A2). We will return to this point in the Discussion.

We have used this improved method to reconstruct Holocene climates across Iberia, and re-examined the trends in summer and winter temperature and plant-available moisture, using a new and relatively comprehensive compilation of pollen data (Shen et al., 2022) with age models based on the latest radiocarbon calibration curve (IntCal20: Reimer et al., 2020). We explicitly test whether there are significant differences in the west-east gradient of moisture and temperature through time. We then analyse the relationships between the changes in the

three climate variables and how trends in these variables are related to external climate forcing.
These analyses allow us to investigate whether the west-east gradient in moisture was less steep
during the mid-Holocene and explore what controls the patterns of climate change across the
region.
**2. Methods**
Multiple techniques have been developed to make quantitative climate reconstructions from
pollen (see reviews in Bartlein et al., 2011; Chevalier et al., 2020; Salonen et al., 2011). Modern
analogue techniques (MAT: Overpeck et al., 1985) tend to produce rapid shifts in reconstructed
values corresponding to changes in the selection of the specific analogue samples, although
this tendency is less marked in the conceptually analogous response surface technique (Bartlein
et al., 1986). Regression-based techniques, including weighted averaging methods such as
Weighted Average Partial Least-Squares (WAPLS: ter Braak and Juggins, 1993), do not
produce step-changes in the reconstructions but suffer from the tendency to compress the
reconstructions towards the central part of the sampled climate range. However, this tendency
can be substantially reduced by accounting for the sampling frequency (fx) and the climate
tolerance of the pollen taxa present in the training data set (fxTWA-PLS: Liu et al., 2020).
Machine-learning and Bayesian approaches have also been applied to derive climate
reconstructions from pollen assemblages (Peyron et al., 1998; Salonen et al., 2019). However,
comparison of fxTWA-PLS with the Bayesian model BUMPER (Holden et al., 2017), shows
that fxTWA-PLS performs better in capturing the climate of the modern training data set from
Europe (Liu et al., 2020).
Although fxTWA-PLS has clear advantages over other quantitative reconstructions techniques,
there is still a slight tendency towards compression. We have therefore made a further
modification to the approach as described in Liu et al. (2020). In the original version of
fxTWA-PLS, the fx correction is applied as a weight with the form of $1/fx^2$ in the regression
(step 7 in Table 1 in Liu et al., 2020). Here (see Appendix A) we make a further modification
of fxTWA-PLS by (a) applying the fx correction separately in both the taxon calculation and
the regression (step 2 and 7 in Table 1 in Liu et al., 2020) as a weight with the form of $1/fx$ and
(b) applying P-splines smoothing (Eilers and Marx, 2021) in order to reduce the dependence
of the fx estimation on bin width. The modified version further reduces the biases at the
extremes of the sampled climate range.
There are no generally accepted rules as to the choice of variables for palaeoclimate
reconstruction. No systematic comparison of these choices has been made. However, it is
widely understood that plant taxon distributions reflect distinct, largely independent controls
by summer temperatures, winter temperatures, and moisture availability (see e.g. Harrison et
al., 2010). Therefore, in common with many other studies (Cheddadi et al., 1997; Jiang et al.,
2010; Peyron et al., 1998; Wei et al., 2021; Zhang et al., 2007), we have chosen bioclimatic
variables that reflect these independent controls, with mean temperature of the coldest month
(MTCO) to represent winter temperatures, mean temperature of the warmest month (MTWA)
to represent summer temperatures and $\alpha$, an estimate of the ratio of actual evapotranspiration
to equilibrium evapotranspiration, to represent plant-available moisture. We choose not to use
mean annual air temperature (MAAT) because it is a composite of summer and winter
conditions; and we prefer to use an index of effective moisture availability (our estimate of $\alpha$
being one such index) to mean annual precipitation (MAP), whose significance for plant
function depends strongly on potential evaporation (a function of temperature and net
radiation). Our calculation of $\alpha$ takes account of this dependence. Growing degree days above
a baseline of 0 °C ($GDD_0$) would be a possible alternative to MTWA as an expression of
summer conditions but is most relevant as a predictor of "cold limits" of trees in cool climates,
whereas MTWA better reflects the high-temperature stress on plants in Mediterranean-type
climates.
We used the modified version of fxTWA-PLS to reconstruct these three climate variables. The
individual and joint effects of MTCO, MTWA and $\alpha$ were tested explicitly using canonical
correspondence analysis (CCA). The modified version further reduces the biases at the
extremes of the sampled climate range, while retaining the desirable properties of WA-PLS in
terms of robustness to spatial autocorrelation (fxTWA-PLS: Liu et al., 2020).
The modern pollen training dataset was derived from the SPECIAL Modern Pollen Data Set
(SMPDS: Harrison, 2019). The SMPDS consists of relative abundance records from 6458
terrestrial sites from Europe, northern Africa, the Middle East and northern Eurasia (SI Fig.
S1) assembled from multiple different published sources. The pollen records were
taxonomically standardized, and filtered (as recommended by Chevalier et al., 2020) to remove
obligate aquatics, insectivorous species, introduced species, and taxa that only occur in
cultivation (see SI Table S1 for the list). Taxa (mainly herbaceous) with only sporadic
occurrences were amalgamated to higher taxonomic levels (genus, sub-family or family) after

ensuring consistency with their distribution in climate space. As a result of these amalgamations, the SMPDS contains data on 247 pollen taxa. For our analysis, we use the 195 taxa that occur at more than 10 sites.

Modern climate data at each of the sites in the training data set were obtained from Harrison (2019). This data set contains climate reconstructions of MTCO, growing degree days above a baseline of $0°$ C ($GDD_0$) and a moisture index (MI), defined as the ratio of annual precipitation to annual potential evapotranspiration. The climate at each site was obtained using geographically weighted regression (GWR) of the CRU CL v2.0 gridded dataset of modern (1961-1990) surface climate at 10 arc minute resolution (New et al., 2002) in order to (a) correct for elevation differences between each pollen site and the corresponding grid cell and (b) make the resulting climate independent of the resolution of the underlying data set. The geographically weighted regression used a fixed bandwidth kernel of $1.06°$ (~140km) to optimize model diagnostics and reduce spatial clustering of residuals relative to other bandwidths. The climate of each pollen site was then estimated based on its longitude, latitude, and elevation. MTCO and $GDD_0$ was taken directly from the GWR regression and MI was calculated for each pollen site using a modified code from SPLASH v1.0 (Davis et al., 2017) based on daily values of precipitation, temperature and sunshine hours again obtained using a mean-conserving interpolation of the monthly values of each. For this application, we used MTCO directly from the data set but calculated MTWA from MTCO and $GDD_0$, based on the relationship between MTCO, MTWA and $GDD_0$ given by Appendix 2 of Wei et al. (2021). We derived $\alpha$ from MI following Liu et al. (2020). The modern training data set provides records spanning a range of MTCO from $-42.4$ °C to 14.8 °C, of MTWA from 4.2 °C to 33.5 °C, and of $\alpha$ from 0.04 to 1.25 (Fig. 1, SI Fig. S1).

The fossil pollen data from the Iberian Peninsula were compiled by Shen et al. (2021) and the data set was obtained from Harrison et al. (2022). The taxonomy used by Shen et al. (2021) is consistent with that employed in the SMPDS. Shen et al. (2021) provides consistent age models for all the records based on the IntCal20 calibration curve (Reimer et al., 2020) and the BACON Bayesian age-modelling tool (Blaauw et al., 2021; Blaauw and Christeny, 2011) using the supervised modelling approach implemented in the `ageR` package (Villegas-Diaz et al, 2021). We excluded individual pollen samples with large uncertainties (standard error larger than 100 years) on the attributed in the new age model. As a result, the climate reconstructions are based on a fossil data set of 7384 pollen samples from 117 records covering part or all of the last

12,000 years (Fig. 2), with 42 individual records provided by the original authors, 73 records
obtained from the European Pollen Database (EPD, www.europeanpollendatabase.net) and 2
records from PANGAEA (www.pangaea.de/). Details of the records are given in Table 1. The
average temporal resolution of these records is 101 years. We then excluded a few samples
where the reconstructed values of α exceed the natural limit of 0 and 1.26. Finally, 7214
samples from 117 records are used for the analyses of the climate reconstructions. Summer
insolation and winter insolation are also calculated using the PAST software based on the age
and latitude of each sample (Hammer et al., 2001).
Variance inflation factor (VIF) scores are calculated for both the modern climates and the
climates reconstructed from fossil pollen records, in order to avoid multicollinearity problems
and thus guarantee the climate variables (MTCO, MTWA, α) used here represent independent
features of the pollen records.
In addition to examining the reconstructions for individual sites, we constructed composite
curves for the Iberian Peninsula as a whole. The composite curves provide a way of comparing
the relationship between trends in the reconstructed climate changes and insolation changes.
The curves were constructed after binning the site-based reconstructions using ± 500-year bins.
We did 1000 bootstrap resampling of the reconstructed climate values in each ± 500-year bin
to avoid the influence of a single value or a single site on the mean climate value in this bin,
and use the standard deviation of the 1000 values to represent the uncertainty of the mean
climate value. We constructed linear regression plots to examine the longitudinal and
elevational patterns in the reconstructed climate variables, and assessed the significance of
differences in these trends through time compared to the most recent bin (0.5 ka ± 500 years)
based on $p$ values, with the customary threshold of 0.05. We then compared the climate trends
with changes in summer and winter insolation.

## 3. Results

The modified version of fxTWA-PLS reproduces the modern climate reasonably well (Table
2). The performance is best for MTCO ($R^2$ 0.75, RMSEP 4.70, slope 0.91) but is also good for
α ($R^2$ 0.68, RMSEP 0.16, slope 0.78) and MTWA ($R^2$ 0.57, RMSEP 3.47, slope 0.71). The
correlations between pollen records and each of the three bioclimate variables, as assessed by
CCA, were strong for both modern climate data and fossil reconstructions (Table 3). The
variance inflation factor (VIF) scores are all less than 6, so there are no multicollinearity
problems (Table 3) (Allison, 1994), making it possible to independently reconstruct all three
climate variables based on pollen data. Furthermore, the taxa that contribute most strongly to
reconstructing colder/warmer or wetter/drier climates show predictable patterns consistent with
their known ecological preferences (SI Table S2).
Winters were generally colder than present during the early to mid-Holocene, as shown by the
coherent patterns of reconstructed anomalies at individual sites (Fig. 3a, 3d). Here "present"
means the most recent pollen bin (0.5 ka ± 500 years). The composite curve also shows a
general increase in winter temperatures through time (Fig. 4a), consistent with the trend in
winter insolation (Fig. 4d). The composite curve shows that it was ca 4°C cooler than today at
11.5 ka and conditions remained cooler than present until ca 2.5 ka. Winter temperatures today
increase from north to south and are also affected by elevation; these patterns are still present
in the Holocene reconstructions, but there is no spatial differentiation between western and
eastern Iberia in the anomalies (Table 4, SI Fig. S2). The similarity of the changes compared
to present geographically is consistent with the idea that the changes in winter temperature are
driven by changes in winter insolation.
Summers were somewhat hotter than present in the west and cooler than present in the east
during the early and mid-Holocene, as shown by the reconstructed anomalies at individual sites
(Fig. 3b, 3e). This west-east difference could not arise if the changes in summer temperatures
were a direct reflection of the insolation forcing (Fig. 4e). Indeed, the composite curve shows
relatively little change in MTWA (Fig. 4b), confirming that there is no direct relationship to
insolation forcing (Fig. 4e).
There is a strong west-east gradient in α at the present day (Fig. 2), with wetter conditions in
the west and drier conditions in the east. However, the reconstructed anomalies at individual
sites (Fig. 3c, 3f) suggest that west was drier and the east was wetter than present in the mid-
Holocene, resulting in a flatter west-east gradient. The west-east gradient is significantly
different from present between 9.5 ~ 3.5 ka (Fig. 5, Table 4), implying stronger moisture
advection into the continental interior during the mid-Holocene. The change in gradient is seen
in both high and low elevation sites (SI Fig. S3). There is also significant change in α with
elevation between 9.5 ~ 4.5 ka (Table 4, SI Fig. S4).
Summer temperatures are strongly correlated with changes in α, both in terms of spatial
correlations in the modern data set at a European scale and in terms of spatial and temporal
correlations the fossil data set from Iberian Peninsula (Fig. 6). The patterns of reconstructed
anomalies in MTWA and α at individual sites are also coherent (Fig. 3b, 3c, 3e, 3f), showing
drier conditions and hotter summers than present in the west and wetter conditions with cooler
summers in the east during the early to mid-Holocene. The west-east gradient in MTWA was
significantly different from present between 9.5 and 3.5 ka except 8.5 ka (Table 4, SI Fig. S5),
roughly the interval when the gradient in α was also significantly different from present. Again,
the change in the east-west gradient is registered at both high and low elevation sites (SI Fig.
S6). However, there is no significant change in MTWA with elevation except 8.5 and 7.5 ka
(Table 4, SI Fig. S7).
**4. Discussion**
The modified version of fxTWA-PLS (fxTWA-PLS2) (Table 2, Table A1) shows a few
differences compared to the previous version (fxTWA-PLS1). Cross-validation $R^2$ values are
higher for MTCO and MTWA, and almost unchanged for α. The maximum bias shows a
decrease for all the three variables, especially for MTCO. The compression problem is also
reduced for MTCO ($b_1$ increases from 0.82 to 0.91) and MTWA ($b_1$ increases from 0.69 to
0.71) while remaining roughly the same for α. The overall performance statistics thus show
substantial improvements for MTCO and MTWA, while they show little change for α.
However, Figure A1 shows that "unphysical" reconstructions beyond the natural limits of α
(0–1.26) are greatly reduced, especially for the lower limit. There are also fewer outliers in
Figure A1 and A2 for all three variables. Thus overall, the modified version further reduces the
reconstruction biases, especially at the extremes of the sampled climate range. This
improvement probably occurs because of the separate application of 1/fx correction during
both the calculation of optima and tolerances of taxa and during the regression step – instead
of applying an overall weight of $1/fx^2$ at the regression step, which can result in some extreme
values (with low sampling frequency) being weighed too strongly and appearing as outliers.
fxTWA-PLS2 reconstructed climates have shown that there was a gradual increase in MTCO
over the Holocene, both for most of the individual sites represented in the data set (Fig. 3) and
for Iberia as a whole (Fig. 4). Colder winters in southern Europe during the mid-Holocene (6
ka) are a feature of many earlier reconstructions (e.g. Cheddadi et al., 1997; Wu et al., 2007).
A general warming trend over the Holocene is seen in gridded reconstructions of winter season
(December, January, February) temperatures as reconstructed using the modern analogue
approach by Mauri et al. (2015), although there is somewhat less millennial-scale variability
in these reconstructions (Fig. 7). Nevertheless, their reconstructions show a cooling of 3°C in
the early Holocene, comparable in magnitude to the ca 4°C cooling at 11.5 ka reconstructed
here. Although they show conditions slightly cooler than present persisting up to 1 ka, the
differences are very small (ca 0.5°C) after 2 ka, again consistent with our reconstructions of
MTCO similar to present by 2.5 ka. Quantitative reconstructions of winter temperature for the
5 terrestrial sites from the Iberian Peninsula in the Kaufman et al. (2020) compilation all show
a general trend of winter warming over the Holocene, but the magnitude of the change at some
of the individual sites is much larger (ca 10°C) and there is no assessment of the uncertainty
on these reconstructions. The composite curve of Kaufman et al. (2020) shows an increasing
trend in MTCO through the Holocene although with large uncertainties (Fig. 7). In contrast to
the consistency of the increasing trend in MTCO during the Holocene between our
reconstructions and those of Mauri et al. (2015) and Kaufman et al. (2020), there is no
discernible trend in MTCO during the Holocene reconstruction of Tarroso et al. (2016). Indeed,
there is no significant change in their MTCO values after ca 9 ka, either for the Peninsula as a
whole (Fig. 7) or for any of the four sub-regions they considered. Our reconstructed trend in
winter temperature is consistent with the changes in insolation forcing at this latitude during
the Holocene, and is also consistent with transient climate model simulations (Braconnot et al.,
2019; Carré et al., 2021; Dallmeyer et al., 2020; Parker et al., 2021) of the winter temperature
response to changing insolation forcing over the late Holocene in this region (Fig. 8, SI Fig.
S8). Thus, we suggest that changes in winter temperatures are a direct consequence of
insolation forcing.
We have shown that there is no overall trend in MTWA during the Holocene (Fig. 4).
According to our reconstructions, summer temperatures fluctuated between ca 0.5°C above or
below modern temperature. The lack of coherent trend in MTWA is consistent with the gridded
reconstructions of summer (June, July, August) temperature in the Mauri et al. (2015) data set
and also with the 5 terrestrial sites from Iberia included in the Kaufman et al. (2020) data set.
However, the patterns shown in the three data sets are very different from one another. Mauri
et al. (2015) suggest the early Holocene was colder than today, and although temperatures
similar to today were reached at 9 ka, most of the Holocene was characterised by cooler
summers. Kaufman et al. (2020), however, showed warmer than present conditions during the
early Holocene although they also show cooler conditions during the later Holocene. The
differences between the three data sets could reflect differences in the reconstruction methods,
or differences in the number of records used and in the geographic sampling. However, given
the fact that all three data sets show similar trends in winter temperature, the lack of coherency
between the data sets for MTWA points to there not being a strong, regionally coherent signal
of summer temperature changes during the Holocene. Tarroso et al. (2016) also showed no
significant changes in MTWA after ca 9 ka (Fig. 7).
The chironomid record from Laguna de la Roya covers the late glacial and terminates at 10.5
ka (Muñoz Sobrino et al., 2013). The reconstructed July temperature during the early Holocene
is ca 12~13 °C, which is considerably cooler than today at this site. However, the authors
caution that these samples have poor analogues and the record should be interpreted with
caution. Chironomid-based reconstructions of July temperature at Basa de la Mora (Tarrats et
al., 2018), a high elevation site in the Pyrenees, indicate temperatures within ± 0.5° C of the
modern during the early to mid-Holocene (10~6 ka), similar to our regional composite
reconstructions. However, they show persistently conditions cooler than present by ca 1.5 °C
between 4.5 and 2 ka, not seen in our reconstructions. Furthermore, direct comparison of our
reconstructions of MTWA at Basa de la Mora (SI Fig. S9) to the chironomid-based
reconstructions highlights that the two records show very different trajectories, since the
pollen-based reconstruction of this site shows a consistent warming trend throughout the
Holocene. Although Tarrats et al. (2018) argue that discrepancies between their temperature
reconstructions and pollen-based reconstructions reflects the fact that the vegetation of Iberia,
including the mountain areas, is largely driven by moisture changes and perhaps is not a good
indicator of temperature, we have shown that there is sufficient information in the pollen
records to reconstruct temperature and moisture independently (Table 3, Table S2). Thus, the
cause of the differences between the pollen-based and chironomid-based reconstructions at
Basa de la Mora is presumably related to methodology. In particular, the chironomid
reconstructions use a training data set that does not include samples from the Pyrenees, or
indeed the Mediterranean more generally, and may therefore not provide good analogues for
Holocene changes at this site.
The lack of a clear trend in MTWA in our reconstructions (Fig. 4b) is not consistent with
insolation forcing (Fig. 4e), which shows a declining trend during the Holocene nor is it
consistent with simulated changes in MTWA in transient climate model simulations of the
summer temperature response to changing insolation forcing over the Holocene in this region
(Fig. 8). The change in moisture gradient during the mid-Holocene, however, suggests an
alternative explanation whereby changes in summer temperature are a response to land-surface
feedbacks associated with changes in moisture (Fig. 6). Specifically, the observed increased
advection of moisture into eastern Iberia would have created wetter conditions there, which in
turn would permit increased evapotranspiration, implying less allocation of available net
radiation to sensible heating, and resulting in cooler air temperatures. Our reconstructions show
that the west-east moisture gradient in mid-Holocene (Fig. 5) was significantly flatter than the
steep moisture gradient today (Fig. 2), implying a significant increase in moisture advection
into the continental interior during this period. Mauri et al. (2015) also showed that summers
were generally wetter than present in the east but drier than present in the west at early to mid-
Holocene, supporting the idea of a flatter west-east gradient.
We have shown that stronger moisture advection is not a feature of transient climate model
simulations of the Holocene, which may explain why these simulations do not show a strong
modification of the insolation-driven changes in summer temperature (Fig. 8). Although the
amplitude differs, all of the models show a general decline in summer temperature. The failure
of the current generation of climate models to simulate the observed strengthening of moisture
transport into Europe and Eurasia during the mid-Holocene has been noted for previous
versions of these models (e.g. Bartlein et al., 2017; Mauri et al., 2014) and also shown in Fig.
S8. Mauri et al. (2014), for example, showed that climate models participating in the last phase
of the Coupled Model Intercomparison Project (CMIP5/PMIP3) were unable to reproduce
reconstructed climate patterns over Europe at 6000 yr B.P. and indicated that this resulted from
over-sensitivity to changes in insolation forcing and the failure to simulate increased moisture
transport into the continent. Bartlein et al. (2017) showed that the CMIP5/PMIP3 models
simulated warmer and drier conditions in mid-continental Eurasia at 6000 yr B.P., inconsistent
with palaeo-environmental reconstructions from the region, as a result of the simulated
reduction in the zonal temperature gradient which resulted in weaker westerly flow and reduced
moisture fluxes into the mid-continent. They also pointed out the strong feedback between drier
conditions and summer temperatures. The drying of the mid-continent is also a strong feature
of the mid-Holocene simulations made with the current generation of CMIP6/PMIP4 models
(Brierley et al., 2020). The persistence of these data-model mismatches highlights the need for
better modelling of land-surface feedbacks on atmospheric circulation and moisture.
There are comparatively few pollen-based reconstructions of moisture changes during the
Holocene from Iberia. Records from Padul show increased mean annual and winter
precipitation during the early and mid-Holocene (Camuera et al., 2022; García-Alix et al.,
2021). Reconstructions of mean annual and winter precipitation (Camuera et al., 2022) and the
ratio of annual precipitation to annual potential evapotranspiration (Wei et al., 2021) also show
wetter conditions at this time at El Cañizar de Villarquemado. Both of these sites lie in the
eastern part of the Iberian Peninsula, so these reconstructions are consistent with our
interpretation of wetter conditions in this region during the interval between 9.5 and 3.5 ka.
Ilvonen et al. (2022) provide pollen-based reconstructions of mean annual, summer and winter
precipitation from 8 sites in Iberia, using WAPLS and a Bayesian modelling approach.
Although they focus on the contrasting pattern of hydroclimate evolution between northern and
southern Iberia, the three easternmost sites (San Rafael, Navarres, and Qintanar de la Sierra)
show much wetter conditions during the early to mid-Holocene. With the exception of the
record from Monte Areo, the records from further west are relatively complacent and indeed
two sites (Zalamar, El Maillo) show decreased precipitation between 8 and 4 ka. Thus, these
records are consistent with our interpretation that the west-east gradient of moisture was
reduced between 9.5 and 4.5 ka.
Speleothem oxygen-isotope data from the Iberian Peninsula provide support for our pollen-
based reconstructions of changes in the west-east gradient of moisture through the Holocene.
The speleothem records show a progressive increase in temperature from the Younger Dryas
onwards, although the trend is less marked in the west than the east (Baldini et al., 2019). This
warming trend is consistent with our reconstructions of changes in MTCO through the
Holocene. Speleothem records also show distinctly different patterns in moisture availability,
with sites in western Iberia indicating wetter environments during early Holocene and a
transition to drier conditions from ca 7.5 cal ka BP to the present (Stoll et al., 2013; Thatcher
et al., 2020) while eastern sites record wetter conditions persisting from 9 to 4 cal ka (Walczak
et al., 2015). This finding would support the weaker west to east moisture gradient shown by
our results.
Pollen data are widely used for the quantitative reconstruction of past climates (see discussion
in Bartlein et al., 2011), but reconstructions of moisture indices are also affected by changes in
water-use efficiency caused by the impact of changing atmospheric $CO_2$ levels on plant
physiology (Farquhar, 1997; Gerhart and Ward, 2010; Prentice et al., 2017; Prentice and
Harrison, 2009). This has been shown to be important on glacial-interglacial timescales, when
intervals of lower-than-present $CO_2$ result in vegetation appearing to reflect drier conditions
than were experienced in reality (Prentice et al., 2011, 2017; Wei et al., 2021). We do not
account for this $CO_2$ effect in our reconstructions of α because the change in $CO_2$ over the
Holocene was only 40 ppm. This change relative to modern levels has only a small impact on
the reconstructions (Prentice et al., 2022) and is sufficiently small to be within the
reconstruction uncertainties. Furthermore, accounting for changes in $CO_2$ would not affect the
reconstructed west-east gradient through time.
A more serious issue for our reconstructions may be the extent to which the vegetation cover
of Iberia was substantially modified by human activities during the Holocene. Archaeological
evidence shows that the introduction of agriculture during the Neolithic transition occurred ca
7.6 ka in some southern and eastern areas of the Iberian Peninsula but spread slowly and
farming first occurred only around 6 ka in the northwest (Drake et al., 2017; Fyfe et al., 2019;
Zapata et al., 2004). Anthropogenic changes in land use have been detected at a number of
sites, based on pollen evidence of increases in weeds or the presence of cereals (e.g. Abel-
Schaad and López-Sáez, 2013; Cortés Sánchez et al., 2012; López-Merino et al., 2010; Mighall
et al., 2006; Peña-Chocarro et al., 2005) or the presence of fungal spores associated with animal
faeces which has been used to identify the presence of domesticated animals (e.g. López-Sáez
and López-Merino, 2007; Revelles et al., 2018). The presence of cereals is the most reliable
source of data on human activities, but most cereals only release pollen during threshing and
thus are not found in abundance in pollen diagrams from natural (as opposed to archaeological)
sites (Trondman et al., 2015). Indeed, it is only after ca 1 ka that the number of sites which
record cereal pollen exceeds the number of sites at which cereals are not represented (Githumbi
et al., 2022). Thus, while anthropogenic activities may have been important at the local scale
and particularly in the later Holocene (e.g. Connor et al., 2019; Fyfe et al., 2019; Githumbi et
al., 2022), most of the sites used for our reconstructions are not associated with archaeological
evidence of agriculture or substantial landscape modification. Furthermore, the consistency of
the reconstructed changes in climate across sites provides support for these being largely a
reflection of regional climate changes rather than human activities.
We have used a modified version of fxTWA-PLS to reconstruct Holocene climates of the
Iberian Peninsula because this modification reduced the compression bias in MTCO and
MTWA, and specifically reduces the maximum bias in MTCO, MTWA and α. Although this
modified approach produces better overall reconstructions (Appendix A), its use does not
change the reconstructed trends in these variables through time (SI Fig. S10). Thus, the finding
that winter temperatures are a direct reflection of insolation forcing whereas summer
temperatures are influenced by land-surface feedbacks and changes in atmospheric circulation
is robust to the version of fxTWA-PLS used. However, while we use a much larger data set
than previous reconstructions, the distribution of pollen sites is uneven and the northern part
of the Peninsula is better sampled than the southwest, which could lead to some uncertainties
in the interpretation of changes in the west-east gradient of moisture. It would, therefore, be
useful to specifically target the southwestern part of the Iberian Peninsula for new data
collection. Alternatively, it would be useful to apply the approach used here to the whole of
Eurasia, given that the failure of state-of-the-art climate models to advect moisture into the
continental interior appears to be a feature of the whole region (Bartlein et al., 2017) and not
the Peninsula alone.
**5. Conclusion**
We have developed an improved version of fxTWA-PLS which further reduces compression
bias and provides robust climate reconstructions. We have used this technique with a large
pollen data set representing 117 sites across the Iberian Peninsula to make quantitative
reconstructions of summer and winter temperature and an index of plant-available moisture
through the Holocene. We show that there was a gradual increase in winter temperature through
the Holocene and that this trend broadly follows the changes in orbital forcing. Summer
temperatures, however, do not follow the changes in orbital forcing but appear to be influenced
by land-surface feedbacks associated with changes in moisture. We show that the west-east
gradient in moisture was considerably less pronounced during the mid-Holocene (8~4 ka),
implying a significant increase in moisture advection into the continental interior resulting from
changes in circulation. Our reconstructions of temperature changes are broadly consistent with
previous reconstructions, but are more solidly based because of the increased site coverage.
Our reconstructions of changes in the west-east gradient of moisture during the early part of
the Holocene are also consistent with previous reconstructions, although this change is not
simulated by state-of-the-art climate models, implying that there are still issues to resolve the
associated land-surface feedbacks in these models. Our work provides an improved foundation
for documenting and understanding the Holocene palaeoclimates of Iberia.

**Data and Code Availability**
All the data used are public access and cited here. The code used to generate the climate
reconstructions is available at https://github.com/ml4418/Iberia-paper.git.
**Supplement.** The supplement related to this article is available online.
**Competing interests.** We declare that we have no conflict of interest.
**Author Contributions.** ML, ICP and SPH designed the study. ML, ICP and CJFtB designed
the modifications to fxTWA-PLS. PG-S and GG-R provided pollen data and insights into the
regional palaeoclimate histories. ML carried out the analyses. ML and SPH wrote the first
draft of the paper and all authors contributed to the final draft.
**Acknowledgements.** ML acknowledges support from Imperial College through the Lee
Family Scholarship. YS and SPH acknowledge support from the ERC-funded project GC 2.0
(Global Change 2.0: Unlocking the past for a clearer future; grant number 694481). ICP
acknowledges support from the ERC under the European Union Horizon 2020 research and
innovation programme (grant agreement no: 787203 REALM). This work is a contribution to
the project "Origen y Cuantification de los Cambios Paleoambientales en el Pirineo:
Variabilidad climatic e impacto humano" (PYCACHU: PID2019-106050RB-I00)" funded by
the Ministerio de Ciencia e Innovación.
**Financial support.** This research has been supported by Lee Family Scholarship fund, and
the European Research Council (grant no. GC2.0, 694481, and grant no. REALM, 787203).

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

**Figure and Table Captions**
Figure 1. Climate space represented by mean temperature of the coldest month (MTCO),
mean temperature of the warmest month (MTWA), and plant-available moisture as
represented by α, an estimate of the ratio of actual evapotranspiration to equilibrium
evapotranspiration. The grey points show climate values for a rectangular area (21° W ~ 150°
E, 29° N ~ 82° N) enclosing the SMPDS data set, derived from the Climate Research Unit
CRU CL 2.0 database (New et al., 2002). The black points show climate values of the
SMPDS dataset. The red points show climate values of the Iberian Peninsula region in the
SMPDS dataset.
Figure 2. Map showing the location of the 117 fossil sites in the Iberian Peninsula used for
climate reconstructions. Sites lower than 1000 m a.s.l. are shown as squares, sites higher than
1000 m a.s.l. are shown as triangles. The base maps show modern (a) mean temperature of
the coldest month (MTCO), (b) mean temperature of the warmest month (MTWA), and (c)
plant-available moisture as represented by α, an estimate of the ratio of actual
evapotranspiration to equilibrium evapotranspiration.
Figure 3. Reconstructed anomalies in climate at individual sites through time. The sites are
grouped into high (>1000m) and low (<1000m) elevation sites and organised from west to east.
Grey cells indicate periods or longitudes with no data. The individual plots show the anomalies
in reconstructed (a,d) mean temperature of the coldest month (MTCO), (b,e) mean temperature
of the warmest month (MTWA), and (c,f) plant-available moisture as represented by α, an
estimate of the ratio of actual evapotranspiration to equilibrium evapotranspiration. The
anomalies are expressed as deviations of the mean value in each bin (± 500 years) from the
value at 0.5 ka at each site.
Figure 4. Reconstructed composite changes (anomalies to 0.5 ka) in (a) mean temperature of
the coldest month (MTCO), (b) mean temperature of the warmest month (MTWA) and (c)
plant-available moisture as represented by α, through the Holocene compared to changes in
(d) winter and (e) summer insolation for the latitude of the Iberian Peninsula, using ± 500
years as the bin. The black lines show mean values across sites, with vertical line segments
showing the standard deviations of mean values using 1000 bootstrap cycles of site
resampling.
Figure 5. Changes in the west-east gradient of plant-available moisture as represented by
anomalies in α relative to 0.5 ka at individual sites through the Holocene. The red lines show
the regression lines. The shades indicate the 95 % confidence intervals of the regression lines
Figure 6. The relationship between mean temperature of the warmest month (MTWA) and
plant-available moisture as represented by α (a) in the modern climate data set, and (b) in the
Holocene reconstructions.
Figure 7. Comparison between reconstructed composite changes in climate anomalies. The first
column represents this paper, the second column represents Mauri et al. (2015), the third
column represents Kaufman et al. (2020), the fourth column represents Tarroso et al. (2016).
The composite curves from this paper and Kaufman et al. (2020) are calculated from individual
reconstructions, using anomalies to 0.5 ka and a bin of ± 500 years (time slices are 0.5, 1.5, …,
11.5 ka). The composite curves from Mauri et al. (2015) are converted directly from the gridded
time slices which are provided with anomalies to 0.1 ka and a bin of ± 500 years (time slices
are 1, 2, …, 12 ka). The composite curves from Tarroso et al. (2016) are also converted directly
from the gridded time slices provided, with anomalies to 0.5 ka and a bin of ± 500 years (time
slices are 3, 4, …, 12 ka). Note that Tarroso et al. (2016) applied a smoothing to the data such
that the plots in the paper do not show the excursion in MTWA at 8 ka. In all of the plots, the
black lines show mean values across sites, with vertical line bars showing the standard
deviation of mean values using 1000 bootstrap cycles of site/grid resampling.
Figure 8. Simulated mean values of mean temperature of the coldest month (MTCO), mean
temperature of the warmest month (MTWA) and mean daily precipitation in Iberian Peninsula
between 8 ka and 0 ka, smoothed using 100 year bins. Here BP means before 1950 AD. The
black lines represent Max Planck Institute Earth System Model (MPI) simulations, the red lines
represent Alfred Wagner Insitute Earth System Model (AWI) simulations, the blue lines
represent Institut Pierre Simon Laplace Climate Model (IPSL-CM5) TR5AS simulations, the
orange lines represent Institut Pierre Simon Laplace Climate Model (IPSL-CM6) TR6AV
simulations. The four simulations were forced by evolving orbital parameters and greenhouse
gas concentrations. The four models have different spatial resolution, with the finest resolution
being $1.875° \times 1.875°$ (AWI, MPI) and the coarsest resolution being $1.875° \times 3.75°$ (IPSL-
CM5, TR5AS).
Table 1. Details of the fossil pollen sites used. The fossil pollen data from the Iberian
Peninsula were compiled by Shen et al. (2021) and obtained from
https://doi.org/10.17864/1947.000343. The reference list of this table can be found in the
supplementary.
Table 2. Leave-out cross-validation (with geographically and climatically close sites
removed) fitness of the modified version of fxTWA-PLS, for mean temperature of the coldest
month (MTCO), mean temperature of the warmest month (MTWA) and plant-available
moisture (α), with p-spline smoothed fx estimation, using bins of 0.02, 0.02 and 0.002,
showing results for all the components. RMSEP is the root-mean-square error of prediction.
ΔRMSEP is the per cent change of RMSEP using the current number of components than
using one component less. p assesses whether using the current number of components is
significantly different from using one component less, which is used to choose the last
significant number of components (indicated in bold) to avoid over-fitting. The degree of
overall compression is assessed by linear regression of the cross-validated reconstructions
onto the climate variable, b1, b1.se are the slope and the standard error of the slope,
respectively. The closer the slope (b1) is to 1, the less the overall compression is.
Table 3. Canonical Correspondence Analysis (CCA) result of modern and fossil-
reconstructed MTCO, MTWA and α. The summary statistics for the ANOVA-like
permutation test (999 permutations) are also shown. VIF is the variance inflation factor, Df is
the number of degrees of freedom, $\chi^2$ is the constrained eigenvalue (or the sum of constrained
eigenvalues for the whole model), F is significance, and Pr (>F) is the probability. The CCA
plots can be found in the Supplementary (Fig. S11).
Table 4. Assessment of the significance of anomalies to 0.5 ka through time with latitude and
elevation. The slope is obtained by linear regression of the anomaly onto the longitude or
elevation. $p$ is the significance of the slope (bold parts: $p < 0.05$). $x_0$ is the point where the
anomaly is 0 in the linear equation, which indicates longitude or elevation where the anomaly
changes sign.

Figure 1. Climate space represented by mean temperature of the coldest month (MTCO), mean temperature of the warmest month (MTWA), and plant-available moisture as represented by α, an estimate of the ratio of actual evapotranspiration to equilibrium evapotranspiration. The grey points show climate values for a rectangular area (21° W ~ 150° E, 29° N ~ 82° N) enclosing the SMPDS data set, derived from the Climate Research Unit CRU CL 2.0 database (New et al., 2002). The black points show climate values of the SMPDS dataset. The red points show climate values of the Iberian Peninsula region in the SMPDS dataset.

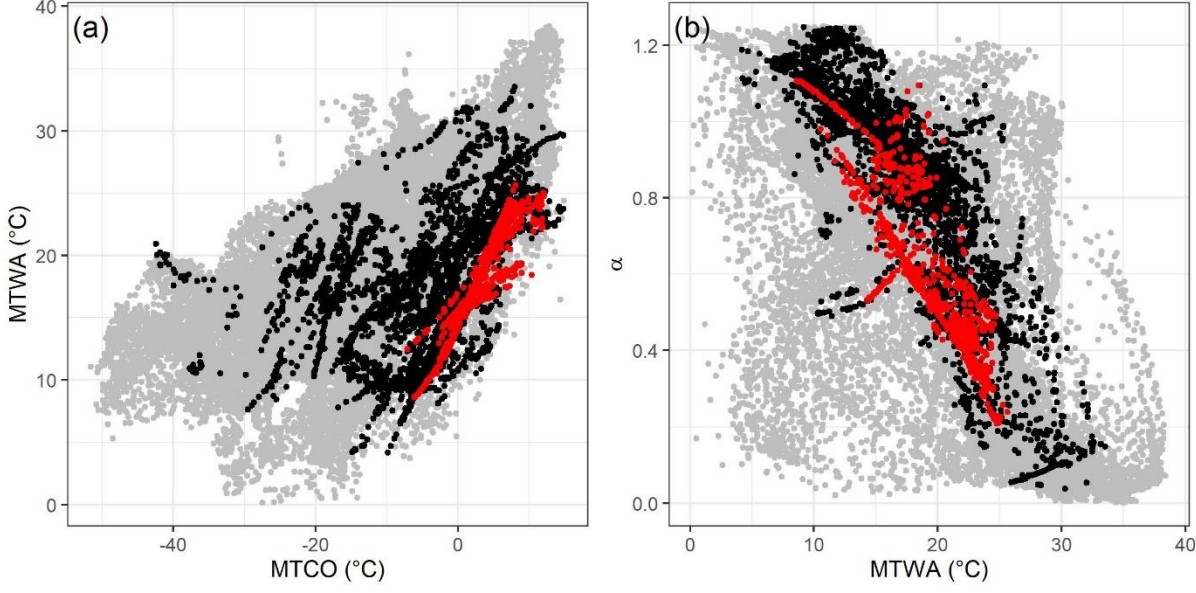

Figure 2. Map showing the location of the 117 fossil sites in the Iberian Peninsula used for
climate reconstructions. Sites lower than 1000 m a.s.l. are shown as squares, sites higher than
1000 m a.s.l. are shown as triangles. The base maps show modern (a) mean temperature of
the coldest month (MTCO), (b) mean temperature of the warmest month (MTWA), and (c)
plant-available moisture as represented by α, an estimate of the ratio of actual
evapotranspiration to equilibrium evapotranspiration.

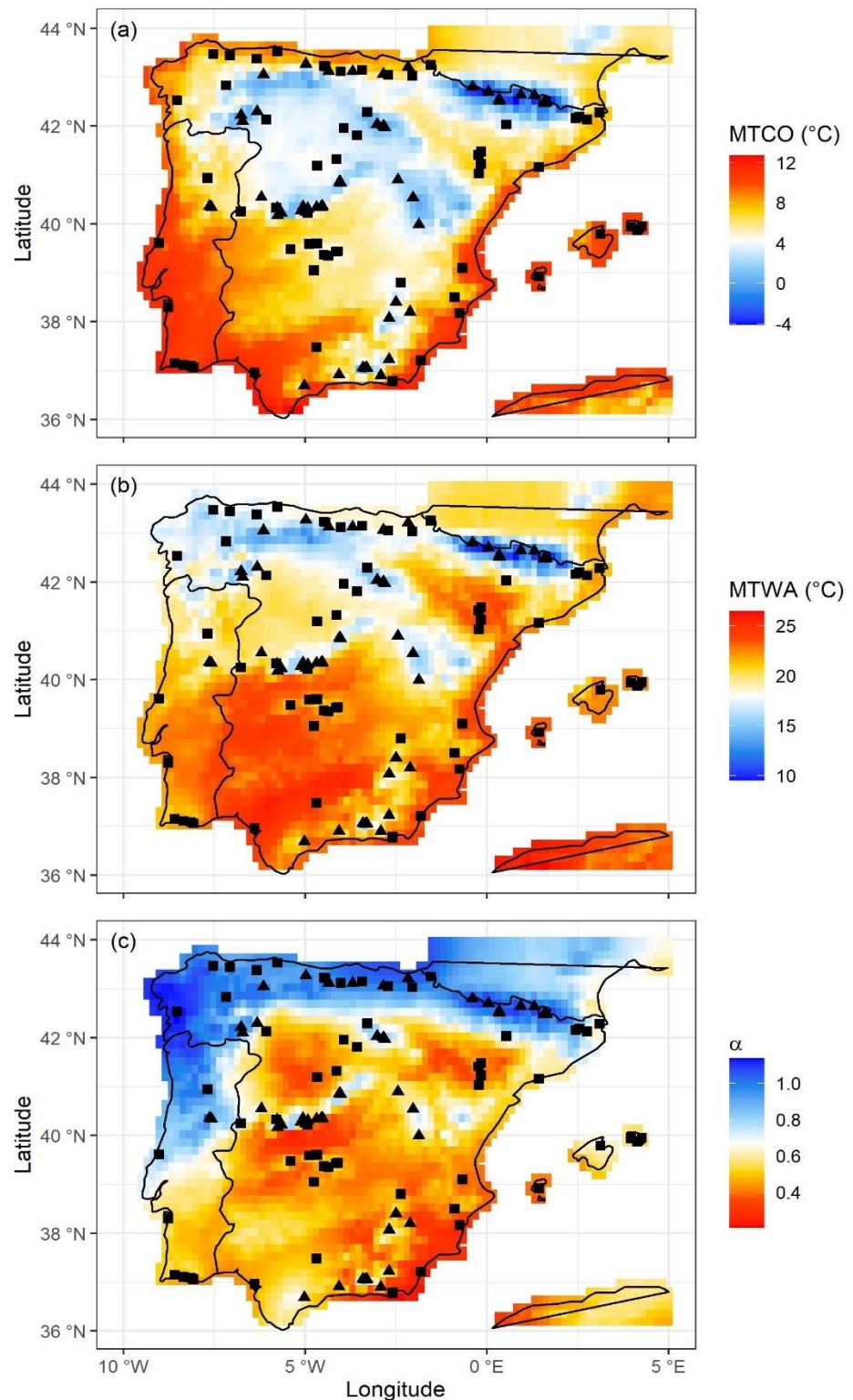


Figure 3. Reconstructed anomalies in climate at individual sites through time. The sites are grouped into high (>1000m) and low (<1000m) elevation sites and organised from west to east. Grey cells indicate periods or longitudes with no data. The individual plots show the anomalies in reconstructed (a,d) mean temperature of the coldest month (MTCO), (b,e) mean temperature of the warmest month (MTWA), and (c,f) plant-available moisture as represented by α, an estimate of the ratio of actual evapotranspiration to equilibrium evapotranspiration. The anomalies are expressed as deviations of the mean value in each bin (± 500 years) from the value at 0.5 ka at each site.

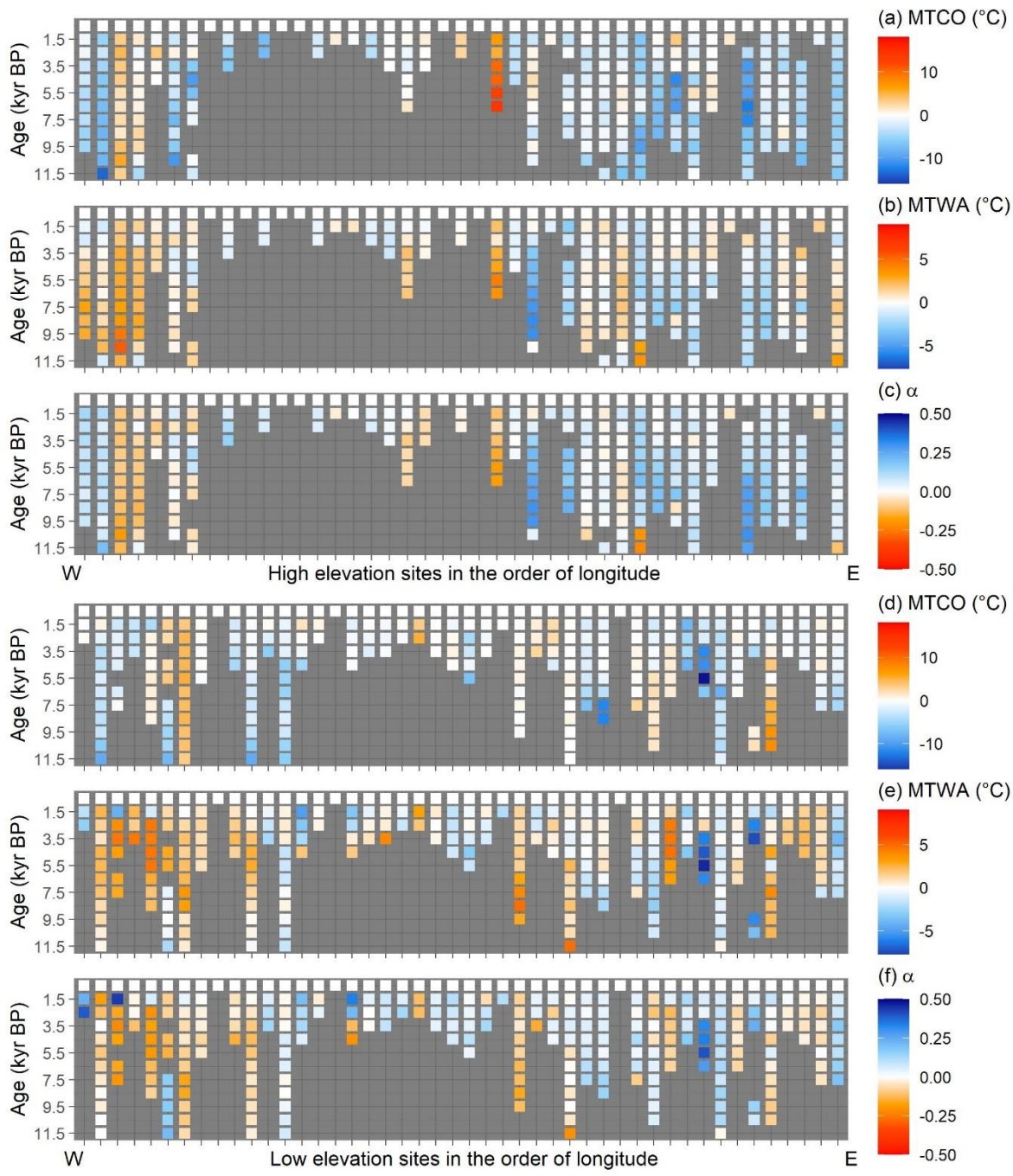

Figure 4. Reconstructed composite changes (anomalies to 0.5 ka) in (a) mean temperature of the coldest month (MTCO), (b) mean temperature of the warmest month (MTWA) and (c) plant-available moisture as represented by α, through the Holocene compared to changes in (d) winter and (e) summer insolation for the latitude of the Iberian Peninsula, using ± 500 years as the bin. The black lines show mean values across sites, with vertical line segments showing the standard deviations of mean values using 1000 bootstrap cycles of site resampling.

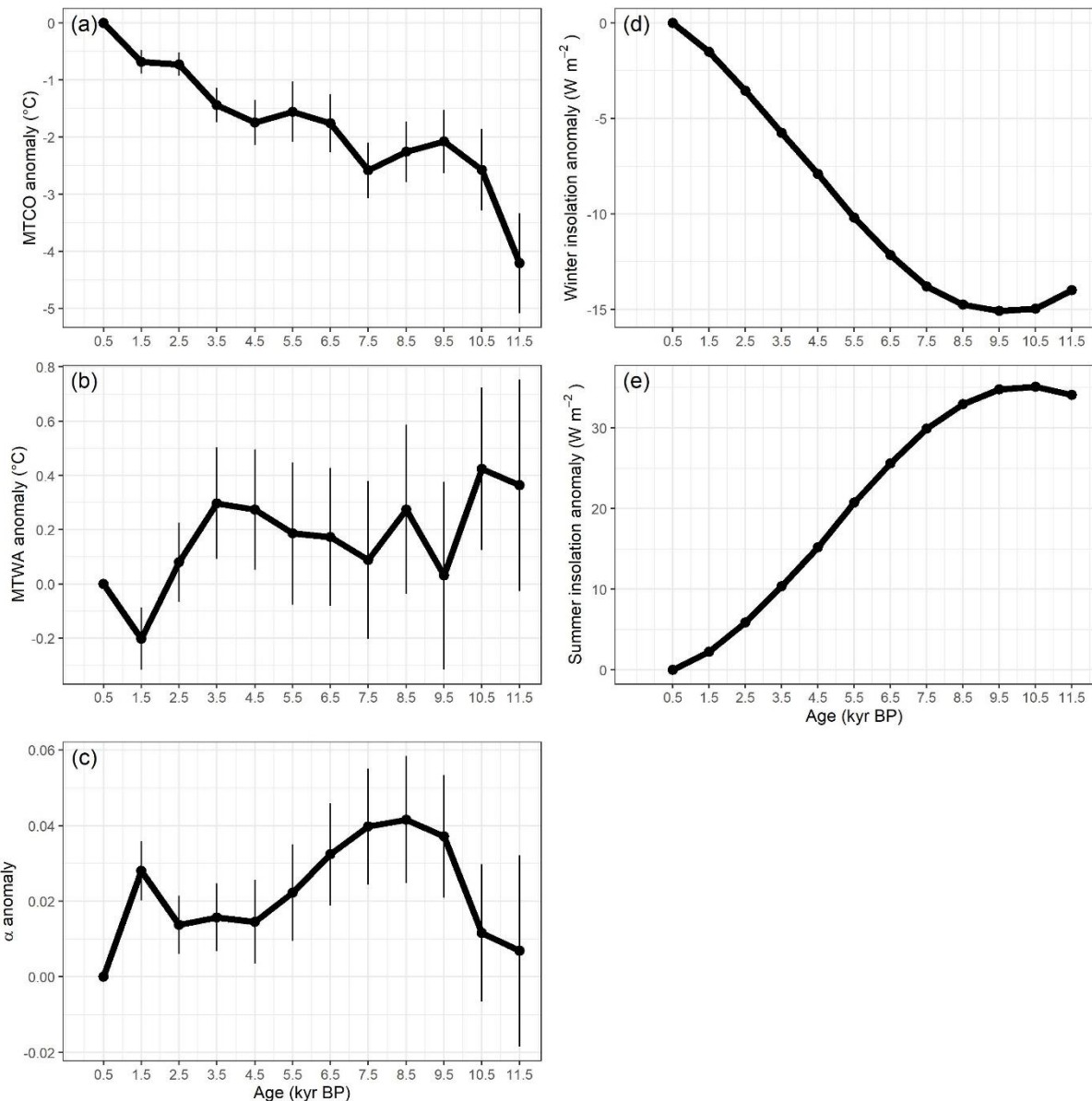

Figure 5. Changes in the west-east gradient of plant-available moisture as represented by
anomalies in α relative to 0.5 ka at individual sites through the Holocene. The red lines show
the regression lines. The shades indicate the 95 % confidence intervals of the regression lines.

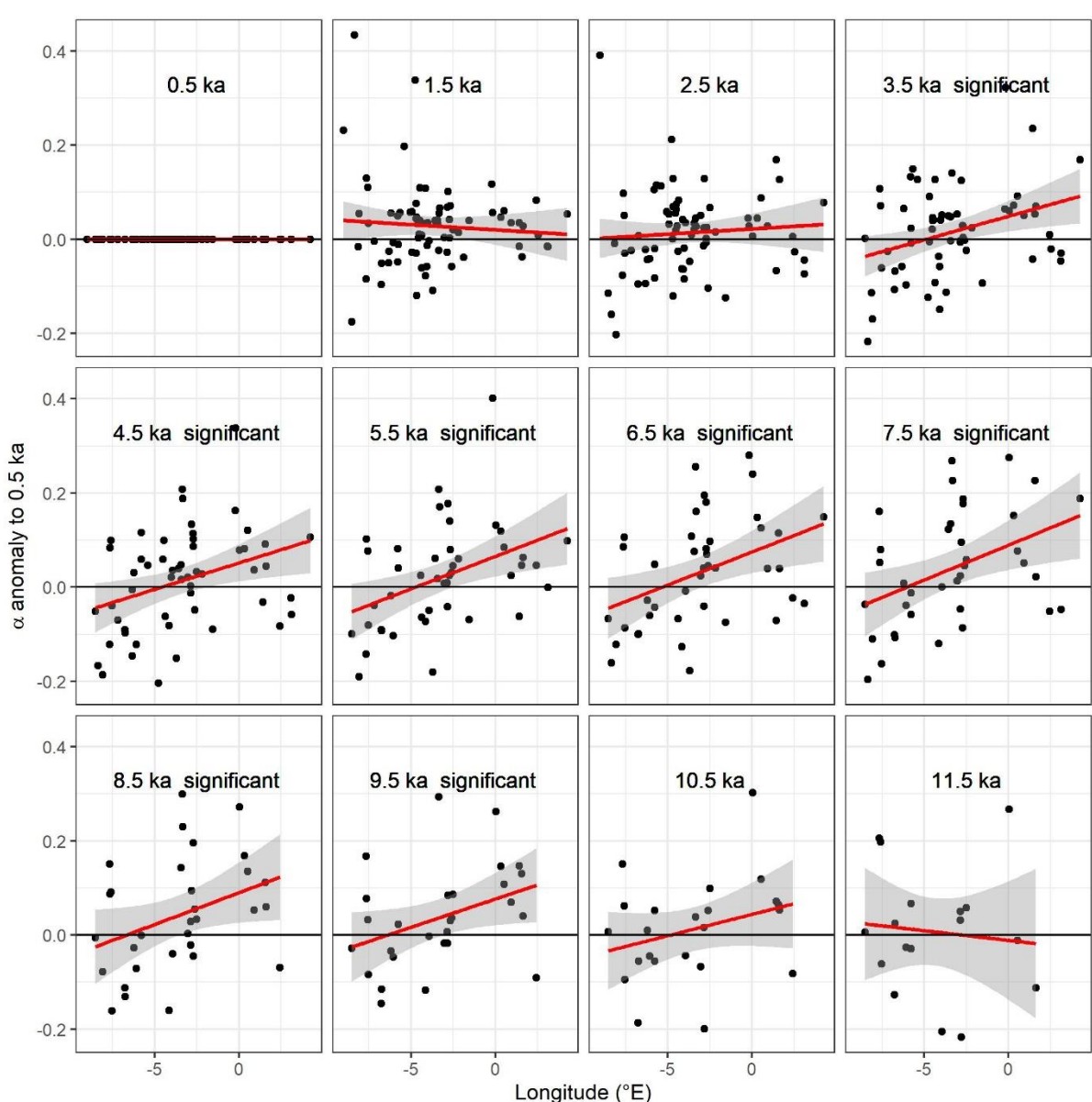


Figure 6. The relationship between mean temperature of the warmest month (MTWA) and
plant-available moisture as represented by α (a) in the modern climate data set, and (b) in the
Holocene reconstructions.

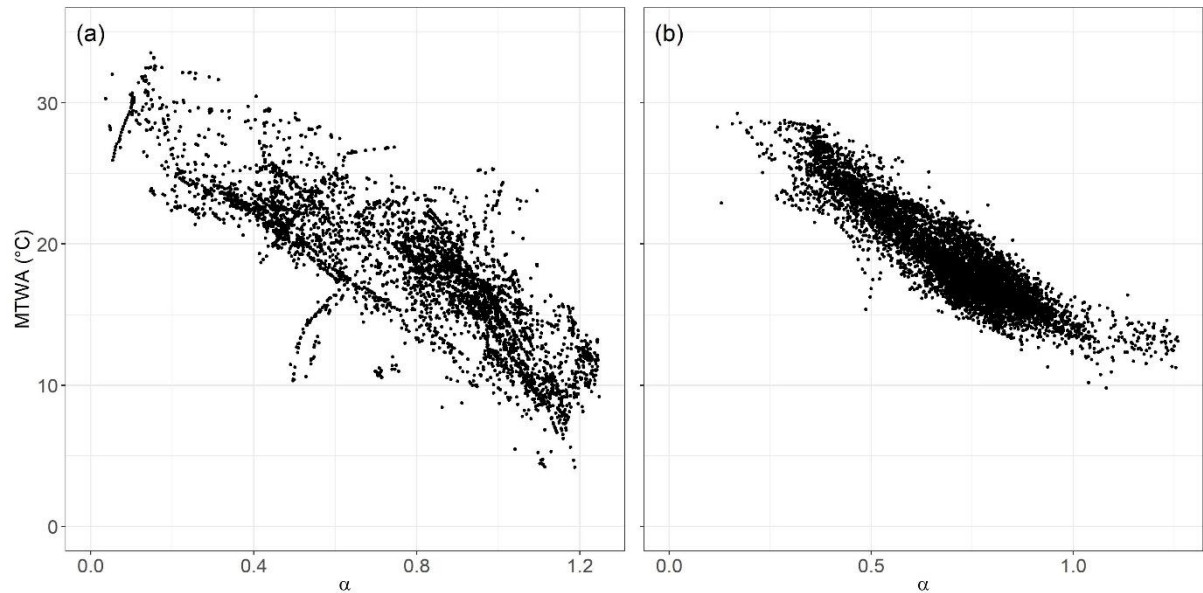


35

Figure 7. Comparison between reconstructed composite changes in climate anomalies. The first column represents this paper, the second column represents Mauri et al. (2015), the third column represents Kaufman et al. (2020), the fourth column represents Tarroso et al. (2016). The composite curves from this paper and Kaufman et al. (2020) are calculated from individual reconstructions, using anomalies to 0.5 ka and a bin of ± 500 years (time slices are 0.5, 1.5, …, 11.5 ka). The composite curves from Mauri et al. (2015) are converted directly from the gridded time slices which are provided with anomalies to 0.1 ka and a bin of ± 500 years (time slices are 1, 2, …, 12 ka). The composite curves from Tarroso et al. (2016) are also converted directly from the gridded time slices provided, with anomalies to 0.5 ka and a bin of ± 500 years (time slices are 3, 4, …, 12 ka). Note that Tarroso et al. (2016) applied a smoothing to the data such that the plots in the paper do not show the excursion in MTWA at 8 ka. In all of the plots, the black lines show mean values across sites, with vertical line bars showing the standard deviation of mean values using 1000 bootstrap cycles of site/grid resampling.

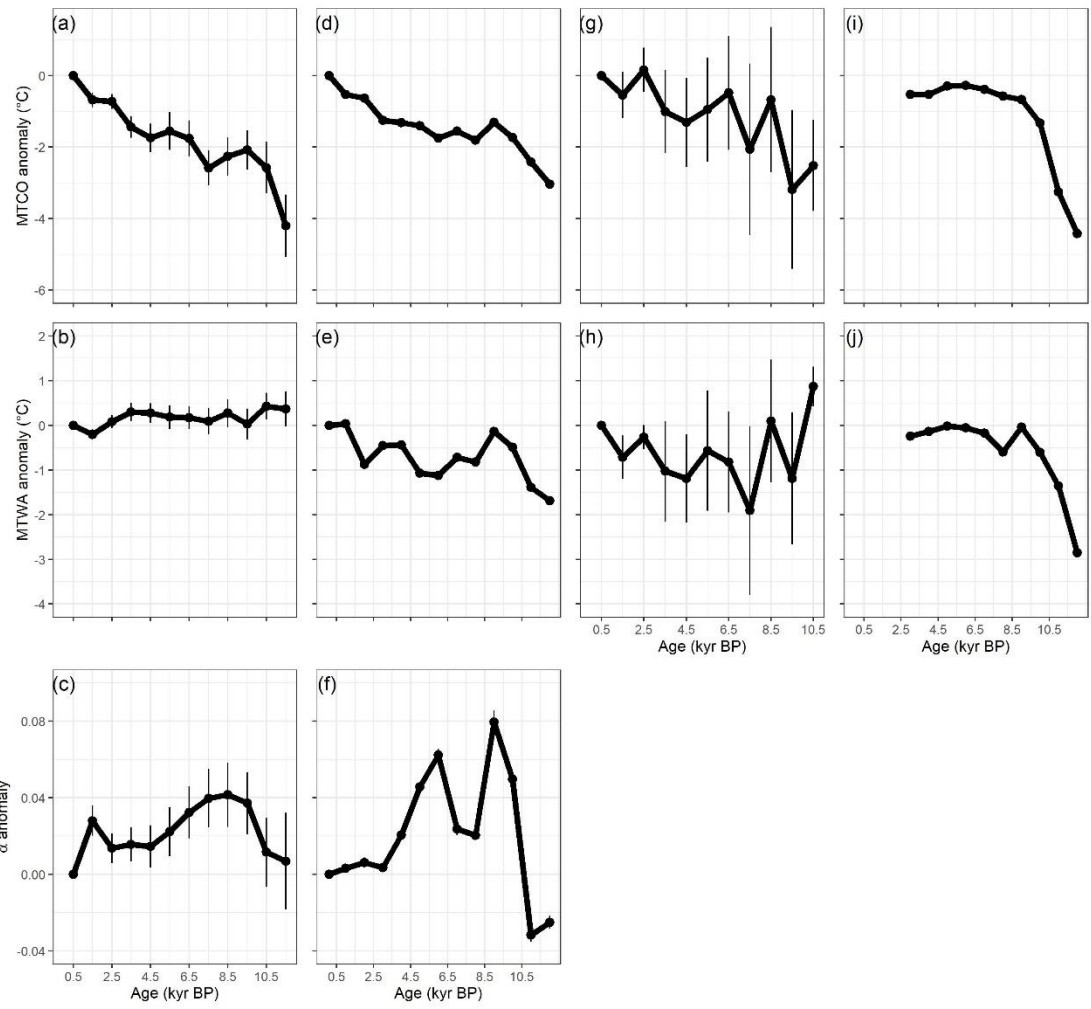

Figure 8. Simulated mean values of mean temperature of the coldest month (MTCO), mean
temperature of the warmest month (MTWA) and mean daily precipitation in Iberian
Peninsula between 8 ka and 0 ka, smoothed using 100 year bins. Here BP means before 1950
AD. The black lines represent Max Planck Institute Earth System Model (MPI) simulations,
the red lines represent Alfred Wagner Insitute Earth System Model (AWI) simulations, the
blue lines represent Institut Pierre Simon Laplace Climate Model (IPSL-CM5) TR5AS
simulations, the orange lines represent Institut Pierre Simon Laplace Climate Model (IPSL-
CM6) TR6AV simulations. The four simulations were forced by evolving orbital parameters
and greenhouse gas concentrations. The four models have different spatial resolution, with
the finest resolution being 1.875° × 1.875° (AWI, MPI) and the coarsest resolution being
1.875° × 3.75° (IPSL-CM5, TR5AS).

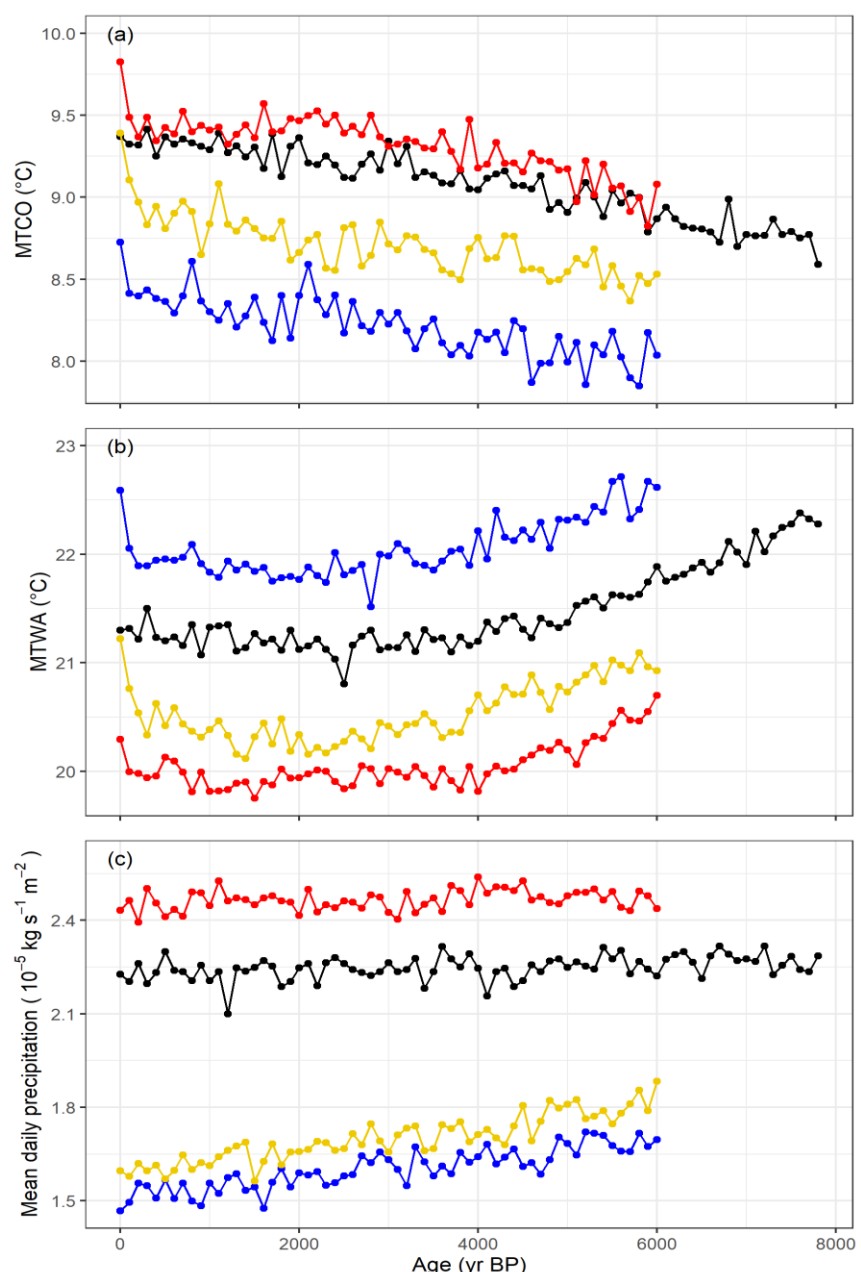


Table 1. Details of the fossil pollen sites used. The fossil pollen data from the Iberian Peninsula were compiled by Shen et al. (2021) and obtained
from https://doi.org/10.17864/1947.000343. The reference list of this table can be found in the supplementary.

| site name | entity name | longitude (°E) | latitude (°N) | elevation (m) | earliest sample (yr BP) | latest sample (yr BP) | length of record (yr) | no of samples | no of dating points | source | reference |
|---|---|---|---|---|---|---|---|---|---|---|---|
| Albufera Alcudia | ALCUDIA | 3.12 | 39.79 | 0 | 7921 | 17 | 7904 | 54 | 4 | EPD | Burjachs et al., (1994) |
| Algendar | ALGENDAR | 3.96 | 39.94 | 21 | 8908 | 3816 | 5092 | 118 | 4 | EPD | Yll et al., (1995, 1997) |
| Almenara de Adaja | ADAJA | −4.67 | 41.19 | 784 | 2830 | 477 | 2353 | 25 | 2 | EPD | López Merino et al., (2009) |
| Alsa | ALSA | −4.02 | 43.12 | 560 | 4908 | 150 | 4758 | 24 | 3 | EPD | Mariscal (1993) |
| Alvor Estuary Ribeira do Farelo Ribeira da Torre | Abi 05/07 | −8.59 | 37.15 | 1 | 7840 | 1699 | 6141 | 76 | 9 | author | Schneider et al., (2010, 2016) |
| Antas | ANTAS | −1.82 | 37.21 | 0 | 11141 | 4309 | 6832 | 95 | 6 | EPD | YII et al., (1995); Cano Villanueva, J. P. (1997); Pantaléon–Cano et al., (2003) |
| Arbarrain Mire | ARBARRAIN | −2.17 | 43.21 | 1004 | 6872 | 78 | 6794 | 91 | 8 | author | Pérez–Díaz et al., (2018) |
| Armacao de Pera Ribeira de Alcantarilha | ADP 01/06 | −8.34 | 37.11 | 2 | 7926 | 8 | 7918 | 17 | 7 | author | Schneider et al., (2010, 2016) |
| Armena | Armena | 0.34 | 42.51 | 2238 | 5668 | 2217 | 3451 | 53 | 27 | author | Leunda et al., (2019) |
| Arroyo de Aguas Frias | AGUASFRIAS | −5.12 | 40.27 | 1120 | 196 | −41 | 237 | 50 | 5 | author | Julio Camarero et al., (2019) |
| Arroyo de las Cárcavas | CARCAVAS | −4.03 | 40.84 | 1300 | 2346 | −57 | 2403 | 40 | 6 | EPD | Morales–Molino et al., (2017a) |
| Arroyo de Navalacarreta | NAVALACA | −4.03 | 40.85 | 1250 | 706 | −60 | 766 | 38 | 6 | EPD | Morales–Molino et al., (2017a) |
| Arroyo de Valdeconejos | VALDECON | −4.06 | 40.86 | 1380 | 611 | −56 | 667 | 44 | 8 | EPD | Morales–Molino et al., (2017a) |
| Atxuri | ATXURI01 | −1.55 | 43.25 | 500 | 6877 | 495 | 6382 | 33 | 2 | EPD | Penalba (1994); Penalba and Garmendia (1989) |
| Ayoó de Vidriales | AYOO | −6.07 | 42.13 | 780 | 11846 | −26 | 11872 | 63 | 15 | EPD | Morales–Molino & García–Antón (2014) |
| Basa de la Mora | BSM08 | 0.33 | 42.55 | 1906 | 9856 | 184 | 9672 | 135 | 16 | author | Pérez–Sanz et al., (2013) |
| Bassa Nera | BSN6 | 0.92 | 42.64 | 1891 | 9599 | −55 | 9654 | 62 | 8 | author | Garces–Pastor et al., (2017) |
| Bermu Mire | BERMU | −4.15 | 39.43 | 783 | 1192 | −25 | 1217 | 38 | 8 | author | Luelmo–Lautenschlaeger et al., (2018a) |
| Borreguil de la Caldera | BdlC–01 | −3.32 | 37.05 | 2992 | 1440 | −56 | 1496 | 80 | 6 | author | Ramos–Román et al., (2016) |

| Bosc dels Estanyons | BOSCESTA | 1.63 | 42.48 | 2180 | 11761 | 26 | 11735 | 91 | 8 | EPD | Miras et al., (2007); De Beaulieu et al., (2005) |
|---|---|---|---|---|---|---|---|---|---|---|---|
| Botija Bog | BOTIJA | −4.7 | 39.6 | 755 | 3773 | 82 | 3691 | 25 | 4 | author | Luelmo−Lautenschlaeger et al., (2018b) |
| Cañada de la Cruz | CANCRUZ | −2.69 | 38.07 | 1595 | 9413 | −6 | 9419 | 39 | 14 | EPD | Yll et al., (1997) |
| Cala'n Porter | CPORTER | 4.13 | 39.87 | 24 | 8809 | 4802 | 4007 | 86 | 4 | EPD | Yll et al., (1994, 1995) |
| Cala Galdana | GALDANA | 3.96 | 39.94 | 47 | 8498 | 4830 | 3668 | 101 | 5 | EPD | López−Merino et al., (2012) |
| Campo Lameiro | PRD4 | −8.52 | 42.53 | 260 | 11948 | −11 | 11959 | 42 | 6 | EPD | Carrión et al., (2007) |
| Canada del Gitano_Sierra de Baza | SBAZA | −2.7 | 37.23 | 1900 | 8460 | 103 | 8357 | 111 | 8 | EPD | Cerrillo Cuenca et al., (2007); Cerrillo Cuenca & González Cordero (2011) |
| Canaleja | CANALEJA | −2.45 | 40.9 | 1029 | 11544 | 5515 | 6029 | 6 | 2 | EPD | Carrion et al., (2001) |
| Castello Lagoon | Castello Lagoon core EM | 3.1 | 42.28 | 2 | 4944 | 307 | 4637 | 85 | 10 | author | Ejarque et al., (2016) |
| Cha das Lameiras | LAMEIRAS | −7.68 | 40.94 | 950 | 11982 | 539 | 11443 | 32 | 8 | author | Burjachs & Expósito (2015) |
| Charco da Candieira | CANDIEIR | −7.58 | 40.34 | 1409 | 11970 | 32 | 11938 | 230 | 31 | EPD | Mariscal Alvarez et al., (1983) |
| Creixell | CreixellT | 1.43 | 41.16 | 1 | 6438 | 723 | 5715 | 32 | 2 | EPD | López−Sáez et al., (2013) |
| Cueto de la Avellanosa | CUETOAV | −4.36 | 43.12 | 1320 | 6969 | 292 | 6677 | 34 | 3 | EPD | López−Sáez et al., (2017) |
| Culazón | CULAZON | −4.49 | 43.23 | 592 | 3895 | −44 | 3939 | 69 | 11 | EPD | van der Knaap & van Leeuwen (1984, 1995, 1997) |
| El Brezosa | BREZOSA | −4.36 | 39.35 | 733 | 3958 | −16 | 3974 | 68 | 11 | author | Burjachs & Expósito (2015); Burjachs et al., (1997) |
| El Carrizal | CARRIZAL | −4.14 | 41.32 | 860 | 9851 | 0 | 9851 | 50 | 6 | EPD | Morales−Molino et al., (2018) |
| El Maíllo mire | MAI | −6.21 | 40.55 | 1100 | 10687 | 91 | 10596 | 104 | 10 | EPD | Franco−Múgica, et al., (2005) |
| El Payo | ELPAYO | −6.77 | 40.25 | 1000 | 571 | −56 | 627 | 50 | 6 | EPD | Morales−Molino et al., (2013) |
| El Perro mire | ELPERRO | −4.76 | 39.05 | 690 | 4694 | −69 | 4763 | 41 | 10 | author | Abel Schaad et al., (2009); Silva−Sánchez et al., (2016) |
| El Portalet | PORTALET | −0.4 | 42.8 | 1802 | 11838 | 2128 | 9710 | 207 | 13 | author | Luelmo−Lautenschlaeger (2019a, 2019b) |
| El Redondo | REDONDO | −5.66 | 40.22 | 1765 | 3222 | 31 | 3191 | 60 | 4 | author | González−Sampériz et al., (2006) |
| El Sabinar | SABINAR | −2.12 | 38.2 | 1117 | 6580 | 1140 | 5440 | 129 | 9 | EPD | López−Sáez et al., (2016) |
| El Tiemblo | TIEMBLO | −4.53 | 40.36 | 1250 | 3184 | 3 | 3181 | 60 | 9 | author | Carrión et al., (2004) |
| Elx | ELX | −0.75 | 38.17 | 1 | 9903 | 3392 | 6511 | 79 | 4 | EPD | López−Sáez et al., (2018a) |

| | | | | | | | | | | | |
|---|---|---|---|---|---|---|---|---|---|---|---|
| Enol | ENOL | −4.99 | 43.27 | 1075 | 10910 | 2487 | 8423 | 30 | 7 | author | Moreno et al., (2011) |
| Es Grau | ESGRAU | 4.26 | 39.95 | 2 | 7648 | −13 | 7661 | 98 | 15 | EPD | Burjachs et al., (2017) |
| Espinosa de Cerrato | CERRATO | −3.94 | 41.96 | 885 | 11578 | 822 | 10756 | 157 | 7 | author | Múgica et al., (2001); Morales−Molino et al., (2017b) |
| Estanilles | ESTANILLES | 1.3 | 42.63 | 2247 | 11908 | 7646 | 4262 | 57 | 11 | EPD | Pérez−Obiol et al., (2012) |
| Estanya | Estanya Catena | 0.53 | 42.03 | 677 | 11882 | −37 | 11919 | 48 | 21 | author | González−Sampériz et al., (2017); Morellón et al., (2011) |
| Fuente de la Leche | LECHE | −5.06 | 40.35 | 1382 | 2783 | −18 | 2801 | 58 | 10 | author | Robles−López et al., (2018) |
| Fuente del Pino Blanco | PINOBLANCO | −4.98 | 40.24 | 1343 | 653 | −38 | 691 | 96 | 5 | author | Robles−López et al., (2018) |
| Hinojos Marsh | HINOJOS | −6.39 | 36.96 | 2 | 4737 | 2682 | 2055 | 46 | 5 | author | López−Sáez et al., (2018b) |
| Hort Timoner | HTIMONER | 4.13 | 39.88 | 40 | 8686 | 5089 | 3597 | 46 | 4 | EPD | Yll et al., (1997) |
| Hoya del Castillo | N−CAS | −0.16 | 41.48 | 258 | 10740 | 5629 | 5111 | 34 | 3 | EPD | Davis & Stevenson (2007) |
| La Cruz | LACRUZ | −1.87 | 39.99 | 1024 | 1521 | 12 | 1509 | 23 | 2 | EPD | Burjachs (1996) |
| La Molina mire | MOLINAES | −6.33 | 43.38 | 650 | 4482 | 388 | 4094 | 152 | 6 | author | López−Merino et al., (2011) |
| Labradillos Mire | LABRADILLOS | −4.57 | 40.34 | 1460 | 1447 | 184 | 1263 | 25 | 5 | author | Robles López et al., (2017) |
| Lago de Ajo | LAGOAJO | −6.15 | 43.05 | 1570 | 11755 | 2175 | 9580 | 44 | 6 | EPD | McKeever et al., (1984); Allen et al., (1996) |
| Lagoa Comprida 2 | LAGOA_CO | −7.64 | 40.36 | 1650 | 9863 | 94 | 9769 | 68 | 4 | EPD | Janssen & Woldringh (1981); Moe & Van Der Knaap (1990); Van Den Brink & Janssen (1985) |
| Lagoa Travessa | TRAVESS1 | −8.77 | 38.3 | 3 | 8174 | 3617 | 4557 | 65 | 4 | EPD | Mateus (1985); Mateus (1989) |
| Laguna de la Mosca | LdlMo composite | −3.31 | 37.06 | 2889 | 8344 | −63 | 8407 | 68 | 18 | author | Manzano et al., (2019) |
| Laguna de la Mula | LdlM 10−02 | −3.42 | 37.06 | 2497 | 4581 | −60 | 4641 | 32 | 8 | author | Jiménez−Moreno et al., (2013) |
| Laguna de la Roya | LAROYA | −6.77 | 42.22 | 1608 | 11927 | −41 | 11968 | 54 | 7 | PANGAEA | Allen et al., (1996) |
| Laguna de Rio Seco | Laguna de Rio Seco core 1 | −3.35 | 37.05 | 3020 | 10455 | −54 | 10509 | 69 | 13 | author | Anderson et al., (2011) |
| Laguna Guallar | N−GUA | −0.23 | 41.41 | 336 | 10654 | 8056 | 2598 | 30 | 6 | EPD | Davis & Stevenson (2007) |
| Laguna Mesagosa | LAGMESAG | −2.81 | 41.97 | 1600 | 11981 | −48 | 12029 | 90 | 5 | EPD | Engelbrechten (1999) |
| Laguna Negra | LAGNEGRA | −2.85 | 42 | 1760 | 11253 | −48 | 11301 | 68 | 9 | EPD | Engelbrechten (1999) |
| Laguna Salada Chiprana | N−SAL | −0.17 | 41.23 | 150 | 6872 | −40 | 6912 | 39 | 4 | EPD | Valero−Garces et al., (2000) |

| Lake Banyoles | BANYOLES_1, Banyoles SB2 | 2.75 | 42.13 | 174 | 11952 | 3316 | 8636 | 141 | 15 | EPD | Pèrez−Obiol & Julià (1994); Revelles et al., (2015) |
|---|---|---|---|---|---|---|---|---|---|---|---|
| Lake Saloio | SALOIO | −9.02 | 39.61 | 70 | 2804 | 313 | 2491 | 24 | 2 | EPD | Gomes (2011) |
| Lanzahíta | LANZBOG | −4.94 | 40.22 | 558 | 2657 | −51 | 2708 | 51 | 8 | author | López−Sáez et al., (1999, 2010) |
| Las Animas Mire | ANIMAS | −5.03 | 36.69 | 1403 | 797 | −57 | 854 | 48 | 10 | author | Alba−Sánchez et al., (2019) |
| Las Lanchas | LANCHAS | −4.89 | 39.59 | 800 | 374 | −8 | 382 | 20 | 2 | author | Luelmo−Lautenschlaeger et al., (2018c) |
| Las Pardillas | LASPARDI | −3.03 | 42.03 | 1850 | 10954 | 404 | 10550 | 74 | 4 | EPD | Goñi & Hannon (1999) |
| Las Vinuelas | VINUELAS | −4.49 | 39.37 | 761 | 4210 | −56 | 4266 | 58 | 9 | author | Morales−Molino et al., (2019) |
| Les Palanques | PALANQUES | 2.44 | 42.16 | 460 | 10011 | 524 | 9487 | 77 | 3 | EPD | Revelles et al., (2018) |
| Manaderos | Manaderos core | −4.69 | 40.34 | 1292 | 1293 | 37 | 1256 | 59 | 9 | author | Robles−López et al., (2020) |
| Marbore | Marbore composite | 0.04 | 42.7 | 2612 | 11683 | −18 | 11701 | 61 | 18 | author | Leunda et al., (2017) |
| Monte Areo mire | AREO | −5.77 | 43.53 | 200 | 11547 | −35 | 11582 | 55 | 12 | EPD | López−Merino et al., (2010) |
| Montes do Buio Cuadramón | CUAII | −7.53 | 43.47 | 700 | 11347 | 241 | 11106 | 19 | 4 | EPD | González et al., (2000) |
| Navamuno | Navamuno_S3 | −5.78 | 40.32 | 1505 | 11971 | −28 | 11999 | 207 | 12 | author | López−Sáez et al., (2020) |
| Navarrés | NAVA1, NAVARRE3 | −0.68 | 39.1 | 225 | 11104 | 3131 | 7973 | 72 | 15 | EPD | Carrion & Dupre (1996); Carrión & Van Geel (1999) |
| Ojos del Tremendal | Ojos del Tremendal core 1 | −2.04 | 40.54 | 1650 | 11875 | 1253 | 10622 | 52 | 4 | author | Stevenson (2000) |
| Pateteros bog | PATATERO | −4.67 | 39.6 | 700 | 2655 | −19 | 2674 | 28 | 4 | EPD | Dorado−Valiño et al., (2014) |
| Peña Negra | PENANEGR | −5.79 | 40.33 | 1000 | 3434 | −62 | 3496 | 63 | 7 | EPD | Stefanini (2008) |
| Pedrido | PEDRIDO | −7.07 | 43.44 | 770 | 5256 | 106 | 5150 | 71 | 30 | EPD | Mighall et al., (2006) |
| Pena de Cadela | CADELA | −7.17 | 42.83 | 970 | 5233 | −14 | 5247 | 91 | 9 | EPD | Abel−Schaad & López−Sáez (2013) |
| Pico del Sertal | SERTAL | −4.44 | 43.22 | 940 | 5200 | 106 | 5094 | 9 | 3 | EPD | Mariscal Alvarez (1986) |
| Pla de l'Estany | PLAESTANY | 2.54 | 42.19 | 520 | 3577 | −37 | 3614 | 43 | 4 | EPD | Burjachs (1994) |
| Planell de Perafita | PERAFITA | 1.57 | 42.48 | 2240 | 10244 | −1 | 10245 | 56 | 11 | EPD | Miras et al., (2010) |
| Posidonia Lligat | LLIGAT | −3.29 | 42.29 | −3 | 779 | 15 | 764 | 32 | 5 | EPD | López−Sáez et al., (2009) |
| Pozo de la Nieve | PozoN_2015 core | −4.55 | 40.35 | 1600 | 2258 | −37 | 2295 | 41 | 10 | author | Robles−López et al., (2017) |

| Name | Code | | | | | | | | | DB | Reference |
|---|---|---|---|---|---|---|---|---|---|---|---|
| Praillos de Bossier Mire | BOSSIER | −4.07 | 36.91 | 1610 | 3428 | 4 | 3424 | 25 | 3 | EPD | Abel−Schaad et al., (2017) |
| Prat de Vila | PRATVILA | 1.43 | 38.92 | 4 | 10776 | 538 | 10238 | 29 | 5 | EPD | Burjachs et al., (2017) |
| Puerto de Belate | BELATE01 | −2.05 | 43.03 | 847 | 8457 | 1746 | 6711 | 60 | 3 | EPD | Penalba (1994); Penalba and Garmendia (1989) |
| Puerto de las Estaces de Trueba | ESTACAS | −3.7 | 43.12 | 1160 | 6263 | 391 | 5872 | 9 | 3 | PANGAEA | Mariscal (1989) |
| Puerto de Los Tornos | TORNOS01 | −3.43 | 43.15 | 920 | 8718 | −34 | 8752 | 47 | 4 | EPD | Penalba and Garmendia (1989) |
| Puerto de Serranillos | SERRANIL | −4.93 | 40.31 | 1700 | 2254 | −50 | 2304 | 34 | 5 | EPD | López−Merino et al., (2009) |
| Quintanar de la Sierra | QUINTA02 | −3.02 | 42.03 | 1470 | 11995 | 1953 | 10042 | 37 | 20 | EPD | Penalba (1994); Penalba and Garmendia (1989) |
| Roquetas de Mar | ROQUETAS | −2.59 | 36.79 | 0 | 6910 | 1057 | 5853 | 32 | 3 | EPD | YII et al., (1995); Cano Villanueva (1997); Pantaléon−Cano (2003); Obiol (1994) |
| Salada Pequeña | N−PEQ | −0.22 | 41.03 | 357 | 4350 | 669 | 3681 | 43 | 5 | EPD | Davis (2010) |
| Saldropo | SALDROPO | −2.72 | 43.05 | 625 | 7577 | 403 | 7174 | 76 | 3 | EPD | Penalba (1994, 1989) |
| Salines playa−lake | SALINES | −0.89 | 38.5 | 475 | 11905 | 1394 | 10511 | 74 | 7 | EPD | Burjachs et al., (2017) |
| San Rafael | SANRAFA | −2.6 | 36.77 | 0 | 10846 | −30 | 10876 | 134 | 6 | EPD | Cano Villanueva (1997); Pantaléon−Cano et al., (2003); Yll et al., (1995) |
| Sanabria Marsh | SANABRIA | −6.73 | 42.1 | 1050 | 11832 | 0 | 11832 | 79 | 9 | EPD | Allen et al., (1996); Hannon (1985); Turner & Hannon (1988) |
| Serra Mitjana Fen | MITJANA | 1.58 | 42.47 | 2406 | 1490 | 412 | 1078 | 15 | 2 | EPD | Miras et al., (2015) |
| Serrania de las Villuercas | VILLUERCAS | −5.4 | 39.48 | 1000 | 4156 | 128 | 4028 | 31 | 4 | author | Gil−Romera et al., (2008) |
| Sierra de Gádor | GADOR | −2.92 | 36.9 | 1530 | 6222 | 1195 | 5027 | 86 | 6 | EPD | Carrión et al., (2003) |
| Siles Lake | SILES | −2.5 | 38.4 | 1320 | 11527 | 189 | 11338 | 67 | 12 | EPD | Carrión (2002) |
| Tubilla del Lago | TUB | −3.57 | 41.81 | 900 | 7436 | 31 | 7405 | 88 | 13 | EPD | Morales−Molino et al., (2017b) |
| Turbera de La Panera Cabras | PANERA | −5.76 | 40.17 | 1648 | 164 | −56 | 220 | 23 | 2 | EPD | Abel Schaad et al., (2009) |
| Valdeyernos bog | VALDEYER | −4.1 | 39.44 | 850 | 3160 | −60 | 3220 | 25 | 4 | EPD | Dorado−Valiño et al., (2014) |
| Valle do Lobo Ribeira de Carcavai | VdL PB2 | −8.07 | 37.06 | 2 | 8331 | 16 | 8315 | 144 | 20 | author | Schneider et al., (2010, 2016) |
| Verdeospesoa mire | VERDEOSPESOA | −2.86 | 43.06 | 1015 | 11137 | 0 | 11137 | 91 | 12 | author | Pérez−Díaz & López−Sáez (2017) |

| | | | | | | | | | | |
|---|---|---|---|---|---|---|---|---|---|---|
| Vilamora Ribeira de Quarteira | Vilamora P01–5 | −8.14 | 37.09 | 4 | 3851 | 919 | 2932 | 30 | 12 | author | Schneider et al., (2010, 2016) |
| Villaverde | VILLAVERDE | −2.37 | 38.8 | 870 | 8066 | 0 | 8066 | 104 | 9 | EPD | Carrión et al., (2001) |
| Xan de Llamas | XL | −6.32 | 42.3 | 1500 | 4113 | 34 | 4079 | 33 | 4 | EPD | Morales–Molino et al., (2011) |
| Zoñar | ZONARcombined | −4.69 | 37.48 | 300 | 3234 | −45 | 3279 | 52 | 17 | author | Martín–Puertas et al., (2008) |


Table 2. Leave-out cross-validation (with geographically and climatically close sites
removed) fitness of the modified version of fxTWA-PLS, for mean temperature of the coldest
month (MTCO), mean temperature of the warmest month (MTWA) and plant-available
moisture (α), with p-spline smoothed fx estimation, using bins of 0.02, 0.02 and 0.002,
showing results for all the components. RMSEP is the root-mean-square error of prediction.
ΔRMSEP is the per cent change of RMSEP using the current number of components than
using one component less. $p$ assesses whether using the current number of components is
significantly different from using one component less, which is used to choose the last
significant number of components (indicated in bold) to avoid over-fitting. The degree of
overall compression is assessed by linear regression of the cross-validated reconstructions
onto the climate variable, $b_1$, $b_1$.se are the slope and the standard error of the slope,
respectively. The closer the slope ($b_1$) is to 1, the less the overall compression is.

| | ncomp | $R^2$ | avg. bias | max. bias | min. bias | RMSEP | ΔRMSEP | $p$ | $b_1$ | $b_1$.se |
|---|---|---|---|---|---|---|---|---|---|---|
| MTCO | 1 | 0.70 | −0.86 | 25.23 | 0.00 | 5.20 | −39.97 | 0.001 | 0.89 | 0.01 |
| | 2 | 0.73 | −0.73 | 25.00 | 0.00 | 4.87 | −6.29 | 0.001 | 0.91 | 0.01 |
| | 3 | 0.74 | −0.71 | 24.38 | 0.00 | 4.86 | −0.32 | 0.001 | 0.91 | 0.01 |
| | **4** | **0.75** | **−0.59** | **24.27** | **0.00** | **4.70** | **−3.26** | **0.001** | **0.91** | **0.01** |
| | 5 | 0.74 | −0.63 | 34.54 | 0.00 | 4.77 | 1.51 | 1.000 | 0.91 | 0.01 |
| MTWA | 1 | 0.52 | −0.29 | 17.13 | 0.00 | 3.72 | −26.88 | 0.001 | 0.69 | 0.01 |
| | 2 | 0.56 | −0.14 | 17.20 | 0.00 | 3.53 | −5.06 | 0.001 | 0.71 | 0.01 |
| | 3 | 0.56 | −0.13 | 17.01 | 0.00 | 3.53 | −0.20 | 0.008 | 0.71 | 0.01 |
| | **4** | **0.57** | **−0.11** | **17.30** | **0.00** | **3.47** | **−1.56** | **0.001** | **0.71** | **0.01** |
| | 5 | 0.57 | −0.11 | 17.34 | 0.00 | 3.48 | 0.10 | 0.780 | 0.71 | 0.01 |
| α | 1 | 0.65 | −0.014 | 0.787 | 0.000 | 0.165 | −39.59 | 0.001 | 0.76 | 0.01 |
| | 2 | 0.68 | −0.016 | 0.781 | 0.000 | 0.159 | −3.55 | 0.001 | 0.77 | 0.01 |
| | **3** | **0.68** | **−0.017** | **0.757** | **0.000** | **0.158** | **−0.61** | **0.023** | **0.78** | **0.01** |
| | 4 | 0.69 | −0.017 | 0.784 | 0.000 | 0.158 | −0.43 | 0.108 | 0.79 | 0.01 |
| | 5 | 0.69 | −0.017 | 0.850 | 0.000 | 0.158 | 0.26 | 0.985 | 0.80 | 0.01 |


Table 3. Canonical Correspondence Analysis (CCA) result of modern and fossil-reconstructed MTCO, MTWA and α. The summary statistics for the ANOVA-like permutation test (999 permutations) are also shown. VIF is the variance inflation factor, Df is the number of degrees of freedom, $\chi^2$ is the constrained eigenvalue (or the sum of constrained eigenvalues for the whole model), F is significance, and Pr (>F) is the probability. The CCA plots can be found in the Supplementary (Fig. S11).

| | Axes | Axis 1 | Axis 2 | Axis 3 | VIF |
|---|---|---|---|---|---|
| **Modern** | Constrained eigenvalues | 0.3819 | 0.1623 | 0.1087 | / |
| | **Correlations of the environmental variables with the axes:** | | | | |
| | MTCO | −0.815 | 0.579 | 0.012 | 1.31 |
| | MTWA | −0.700 | −0.203 | 0.685 | 3.34 |
| | α | 0.883 | 0.430 | −0.187 | 3.39 |
| | | Df | $\chi^2$ | F | Pr (>F) |
| | Whole model | 3 | 0.6530 | 78.113 | 0.001 |
| | MTCO | 1 | 0.3082 | 110.597 | 0.001 |
| | MTWA | 1 | 0.1602 | 57.489 | 0.001 |
| | α | 1 | 0.1846 | 66.252 | 0.001 |
| | CCA 1 | 1 | 0.3819 | 137.076 | 0.001 |
| | CCA 2 | 1 | 0.1623 | 58.252 | 0.001 |
| | CCA 3 | 1 | 0.1087 | 39.011 | 0.001 |
| **Fossil-reconstructed** | Axes | Axis 1 | Axis 2 | Axis 3 | VIF |
| | Constrained eigenvalues | 0.3601 | 0.2266 | 0.2037 | / |
| | **Correlations of the environmental variables with the axes:** | | | | |
| | MTCO | 0.430 | 0.776 | 0.462 | 1.34 |
| | MTWA | 0.987 | 0.141 | −0.076 | 5.40 |
| | α | −0.947 | 0.088 | −0.308 | 5.28 |
| | | Df | $\chi^2$ | F | Pr (>F) |
| | Whole model | 3 | 0.7905 | 226.98 | 0.001 |
| | MTCO | 1 | 0.2465 | 212.34 | 0.001 |
| | MTWA | 1 | 0.3298 | 284.07 | 0.001 |
| | α | 1 | 0.2142 | 184.53 | 0.001 |
| | CCA 1 | 1 | 0.3601 | 310.19 | 0.001 |
| | CCA 2 | 1 | 0.2266 | 195.24 | 0.001 |
| | CCA 3 | 1 | 0.2037 | 175.51 | 0.001 |

Table 4. Assessment of the significance of anomalies to 0.5 ka through time with latitude and
elevation. The slope is obtained by linear regression of the anomaly onto the longitude or
elevation. $p$ is the significance of the slope (bold parts: $p < 0.05$). $x_0$ is the point where the
anomaly is 0 in the linear equation, which indicates longitude or elevation where the anomaly
changes sign.

| | | Longitude (°E) | | | Elevation (km) | | |
|---|---|---|---|---|---|---|---|
| | age (ka) | slope | $p$ | $x_0$ | slope | $p$ | $x_0$ |
| MTCO (°C) | 0.5 | 0.00 | / | / | 0.00 | / | / |
| | 1.5 | −0.07 | 0.411 | −13.02 | −0.30 | 0.411 | −1.21 |
| | 2.5 | −0.15 | 0.095 | −8.56 | −0.52 | 0.179 | −0.40 |
| | 3.5 | −0.13 | 0.314 | −14.83 | −0.81 | 0.142 | −0.77 |
| | 4.5 | −0.12 | 0.444 | −17.28 | −0.69 | 0.319 | −1.46 |
| | 5.5 | −0.24 | 0.247 | −9.49 | −0.61 | 0.503 | −1.43 |
| | 6.5 | −0.18 | 0.372 | −12.74 | −0.87 | 0.293 | −0.88 |
| | 7.5 | −0.15 | 0.421 | −20.39 | −1.38 | 0.080 | −0.67 |
| | 8.5 | −0.03 | 0.890 | −77.87 | −1.58 | 0.065 | −0.10 |
| | 9.5 | 0.01 | 0.954 | 156.31 | −1.79 | 0.060 | 0.11 |
| | 10.5 | 0.20 | 0.474 | 9.25 | −1.38 | 0.241 | −0.64 |
| | 11.5 | 0.23 | 0.528 | 13.77 | 0.12 | 0.947 | 36.35 |
| MTWA (°C) | 0.5 | 0.00 | / | / | 0.00 | / | / |
| | 1.5 | −0.01 | 0.862 | −26.38 | −0.05 | 0.830 | −3.35 |
| | 2.5 | −0.09 | 0.137 | −2.80 | −0.45 | 0.092 | 1.19 |
| | 3.5 | **−0.23** | **0.005** | **−2.03** | −0.40 | 0.284 | 1.74 |
| | 4.5 | **−0.21** | **0.016** | **−2.01** | −0.58 | 0.126 | 1.55 |
| | 5.5 | **−0.26** | **0.011** | **−2.43** | −0.49 | 0.280 | 1.53 |
| | 6.5 | **−0.24** | **0.017** | **−2.30** | −0.62 | 0.137 | 1.41 |
| | 7.5 | **−0.26** | **0.012** | **−3.02** | **−1.05** | **0.019** | **1.28** |
| | 8.5 | −0.24 | 0.061 | −2.43 | **−1.15** | **0.023** | **1.57** |
| | 9.5 | **−0.32** | **0.013** | **−3.20** | −0.44 | 0.459 | 1.34 |
| | 10.5 | −0.18 | 0.115 | −1.23 | 0.54 | 0.276 | 0.44 |
| | 11.5 | 0.13 | 0.453 | −7.25 | 0.37 | 0.663 | 0.22 |
| α | 0.5 | 0.00 | / | / | 0.00 | / | / |
| | 1.5 | 0.00 | 0.508 | 8.99 | −0.01 | 0.393 | 3.40 |
| | 2.5 | 0.00 | 0.517 | −9.89 | 0.02 | 0.249 | 0.19 |
| | 3.5 | **0.01** | **0.006** | **−4.91** | 0.02 | 0.191 | 0.28 |
| | 4.5 | **0.01** | **0.010** | **−4.60** | **0.05** | **0.008** | **0.79** |
| | 5.5 | **0.01** | **0.005** | **−4.75** | **0.05** | **0.027** | **0.67** |
| | 6.5 | **0.01** | **0.007** | **−5.34** | **0.06** | **0.004** | **0.60** |
| | 7.5 | **0.02** | **0.009** | **−6.05** | **0.09** | **0.000** | **0.75** |
| | 8.5 | **0.01** | **0.049** | **−6.67** | **0.09** | **0.000** | **0.88** |
| | 9.5 | **0.01** | **0.048** | **−6.40** | **0.07** | **0.012** | **0.70** |
| | 10.5 | 0.01 | 0.183 | −4.85 | 0.02 | 0.535 | 0.59 |
| | 11.5 | 0.00 | 0.713 | −2.76 | 0.03 | 0.654 | 0.93 |


**Appendix A**
**Theoretical basis:**
**The previous version of fxTWA-PLS (fxTWA-PLS1):**
The estimated optimum ($\hat{u}_k$) and unbiased tolerance ($\hat{t}_k$) of each taxon are calculated from
the modern training data set as follows:
$$\hat{u}_k = \frac{\sum_{i=1}^{n} y_{ik} x_i}{\sum_{i=1}^{n} y_{ik}} \tag{A1}$$

$$\hat{t}_k = \sqrt{\frac{\sum_{i=1}^{n} y_{ik}(x_i - \hat{u}_k)^2}{(1 - 1/N_{2k}) \sum_{i=1}^{n} y_{ik}}} \tag{A2}$$

where
$$N_{2k} = \frac{1}{\sum_{i=1}^{n} \left( \frac{y_{ik}}{\sum_{i'=1}^{n} y_{i'k}} \right)^2} \tag{A3}$$

where $n$ is the total number of sites; $y_{ik}$ is the observed abundance of the $k^{th}$ taxon at the $i^{th}$
site; $x_i$ is the observed climate value at the $i^{th}$ site; $N_{2k}$ is the effective number of occurrences
for the $k^{th}$ taxon.
fx correction is applied as weight in the form of $1/fx^2$ at regression at step 7 in Table 1 in Liu
et al. (2020). The regression step uses robust linear model fitting by the R code:

$$rlm\left(x_i \sim comp_1 + comp_2 + \cdots + comp_{pls}, weights = {}^1/_{fx^2}\right) \tag{A4}$$


**The modified version of fxTWA-PLS (fxTWA-PLS2):**
The distribution of $y_{ik}$ is influenced by the distribution of the climate variable, so we need to
apply the fx correction when calculating optimum and tolerance for each taxon as follows:
$$\hat{u}_k = \frac{\sum_{i=1}^{n} \frac{y_{ik} x_i}{f_{x_i}}}{\sum_{i=1}^{n} \frac{y_{ik}}{f_{x_i}}} \tag{A5}$$

$$\hat{t}_k = \sqrt{\frac{\sum_{i=1}^{n} \frac{y_{ik}(x_i - \hat{u}_k)^2}{f_{x_i}}}{\left(1 - \frac{1}{N_{2k}}\right) \sum_{i=1}^{n} \frac{y_{ik}}{f_{x_i}}}} \tag{A6}$$

where
$$N_{2k} = \frac{1}{\sum_{i=1}^{n} \left( \frac{\frac{y_{ik}}{f_{x_i}}}{\sum_{i'=1}^{n} \frac{y_{i'k}}{f_{x_{i'}}}} \right)^2} \tag{A7}$$

The modified version of fxTWA-PLS applies fx correction separately at taxon calculation
and regression (step 2 and 7 in Table 1 in Liu et al., 2020), both using weight in the form of
1/fx. The regression step (step 7) then becomes:
$$rlm\left(x_i \sim comp_1 + comp_2 + \cdots + comp_{pls}, weights = {}^1\!/_{fx}\right) \qquad (A8)$$
The previous version uses fx values extracted from histograms, and different bin widths may
result in different training results. The modified version applies P-splines histogram
smoothing (Eilers and Marx, 2021) with third order difference penalty, which makes the fx
values almost independent on the bin width. The optimal smoothing parameter of the P-spline
penalty was determined by the HFS (Harville-Fellner-Schall) algorithm (Eilers and Marx,
2021) for the Poisson likelihood for the histogram counts.

Table A1. Leave-out cross-validation (with geographically and climatically close sites removed) fitness of the previous and modified version of fxTWA-PLS (fxTWA-PLS1 and fxTWA-PLS2, respectively), for mean temperature of the coldest month (MTCO), mean temperature of the warmest month (MTWA) and plant-available moisture ($\alpha$), using bins of 0.02, 0.02 and 0.002, respectively. n is the number of components used. RMSEP is the root mean square error of prediction. $\Delta$RMSEP is the per cent change of RMSEP using the current number of components than using one component less. $p$ assesses whether using the current number of components is significantly different from using one component less, which is used to choose the last significant number of components (indicated in bold) to avoid overfitting. The degree of overall compression is assessed by doing linear regression to the cross-validation result and the climate variable. b1, b1.se are the slope and the standard error of the slope, respectively. The closer the slope (b1) is to 1, the lower the overall compression is. fx correction is set intrinsic in functions in `fxTWAPLS` package for both versions in this paper, instead of relying on an outside input in Liu et al. (2020), so the values of fxTWA-PLS1 might be slighted different from values in Table 3 in Liu et al. (2020), but it doesn't affect the conclusion.

| | Method | n | $R^2$ | avg. bias | max. bias | min. bias | RMSEP | $\Delta$RMSEP | $p$ | b1 | b1.se |
|---|---|---|---|---|---|---|---|---|---|---|---|
| MTCO | fxTWA-PLS1 | 1 | 0.66 | −0.86 | 31.17 | 0.00 | 5.21 | −39.87 | 0.001 | 0.76 | 0.01 |
| | | 2 | 0.72 | −0.52 | 36.65 | 0.00 | 4.70 | −9.78 | 0.001 | 0.80 | 0.01 |
| | | 3 | 0.73 | −0.47 | 41.18 | 0.00 | 4.62 | −1.63 | 0.001 | 0.82 | 0.01 |
| | | **4** | **0.73** | **−0.51** | **44.86** | **0.00** | **4.58** | **−1.01** | **0.006** | **0.82** | **0.01** |
| | | 5 | 0.73 | −0.41 | 58.35 | 0.00 | 4.62 | 0.89 | 0.708 | 0.83 | 0.01 |
| | fxTWA-PLS2 | 1 | 0.70 | −0.86 | 25.23 | 0.00 | 5.20 | −39.97 | 0.001 | 0.89 | 0.01 |
| | | 2 | 0.73 | −0.73 | 25.00 | 0.00 | 4.87 | −6.29 | 0.001 | 0.91 | 0.01 |
| | | 3 | 0.74 | −0.71 | 24.38 | 0.00 | 4.86 | −0.32 | 0.001 | 0.91 | 0.01 |
| | | **4** | **0.75** | **−0.59** | **24.27** | **0.00** | **4.70** | **−3.26** | **0.001** | **0.91** | **0.01** |
| | | 5 | 0.74 | −0.63 | 34.54 | 0.00 | 4.77 | 1.51 | 1.000 | 0.91 | 0.01 |
| MTWA | fxTWA-PLS1 | 1 | 0.50 | −0.53 | 17.91 | 0.00 | 3.87 | −24.09 | 0.001 | 0.67 | 0.01 |
| | | **2** | **0.56** | **−0.54** | **17.71** | **0.00** | **3.52** | **−8.98** | **0.001** | **0.69** | **0.01** |
| | | 3 | 0.57 | −0.49 | 25.14 | 0.00 | 3.52 | 0.09 | 0.565 | 0.73 | 0.01 |
| | | 4 | 0.57 | −0.43 | 34.92 | 0.00 | 3.56 | 1.12 | 0.974 | 0.75 | 0.01 |
| | | 5 | 0.57 | −0.46 | 32.23 | 0.00 | 3.55 | −0.23 | 0.139 | 0.74 | 0.01 |
| | fxTWA-PLS2 | 1 | 0.52 | −0.29 | 17.13 | 0.00 | 3.72 | −26.88 | 0.001 | 0.69 | 0.01 |
| | | 2 | 0.56 | −0.14 | 17.20 | 0.00 | 3.53 | −5.06 | 0.001 | 0.71 | 0.01 |
| | | 3 | 0.56 | −0.13 | 17.01 | 0.00 | 3.53 | −0.20 | 0.008 | 0.71 | 0.01 |
| | | **4** | **0.57** | **−0.11** | **17.30** | **0.00** | **3.47** | **−1.56** | **0.001** | **0.71** | **0.01** |
| | | 5 | 0.57 | −0.11 | 17.34 | 0.00 | 3.48 | 0.10 | 0.780 | 0.71 | 0.01 |
| $\alpha$ | fxTWA-PLS1 | 1 | 0.63 | −0.020 | 0.773 | 0.000 | 0.174 | −36.23 | 0.001 | 0.78 | 0.01 |
| | | 2 | 0.69 | −0.012 | 0.902 | 0.000 | 0.157 | −9.66 | 0.001 | 0.79 | 0.01 |
| | | **3** | **0.69** | **−0.011** | **0.820** | **0.000** | **0.155** | **−1.28** | **0.001** | **0.79** | **0.01** |
| | | 4 | 0.70 | −0.010 | 0.786 | 0.000 | 0.156 | 0.25 | 0.867 | 0.81 | 0.01 |
| | | 5 | 0.70 | −0.010 | 0.786 | 0.000 | 0.156 | 0.09 | 1.000 | 0.81 | 0.01 |
| | fxTWA-PLS2 | 1 | 0.65 | −0.014 | 0.787 | 0.000 | 0.165 | −39.59 | 0.001 | 0.76 | 0.01 |
| | | 2 | 0.68 | −0.016 | 0.781 | 0.000 | 0.159 | −3.55 | 0.001 | 0.77 | 0.01 |
| | | **3** | **0.68** | **−0.017** | **0.757** | **0.000** | **0.158** | **−0.61** | **0.023** | **0.78** | **0.01** |
| | | 4 | 0.69 | −0.017 | 0.784 | 0.000 | 0.158 | −0.43 | 0.108 | 0.79 | 0.01 |
| | | 5 | 0.69 | −0.017 | 0.850 | 0.000 | 0.158 | 0.26 | 0.985 | 0.80 | 0.01 |

Figure A1. Training results using the last significant number of components. The left panel shows the previous version (fxTWA-PLS1) and the right panel shows the modified version of fxTWA-PLS (fxTWA-PLS2). The 1: 1 line is shown in black; the linear regression line is shown in red, to show the degree of overall compression. The horizontal dashed lines indicate the natural limit of α (0~1.26).

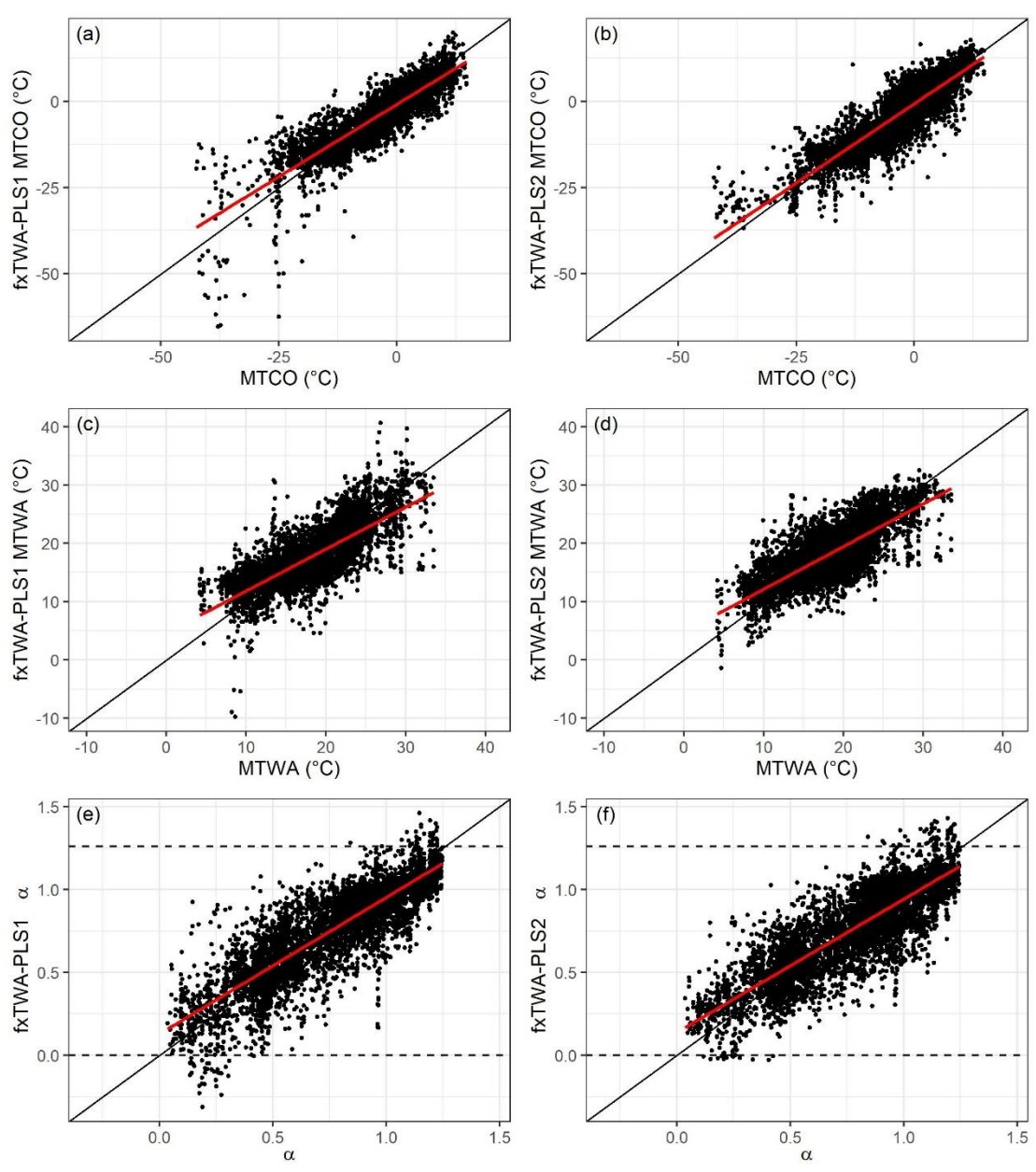

Figure A2. Residuals using the last significant number of components. The left panel shows
the previous version (fxTWA-PLS1) and the right panel shows the modified version (fxTWA-
PLS2) of fxTWA-PLS. The zero line is shown in black; the locally estimated scatterplot
smoothing is shown in red, to show the degree of local compression.

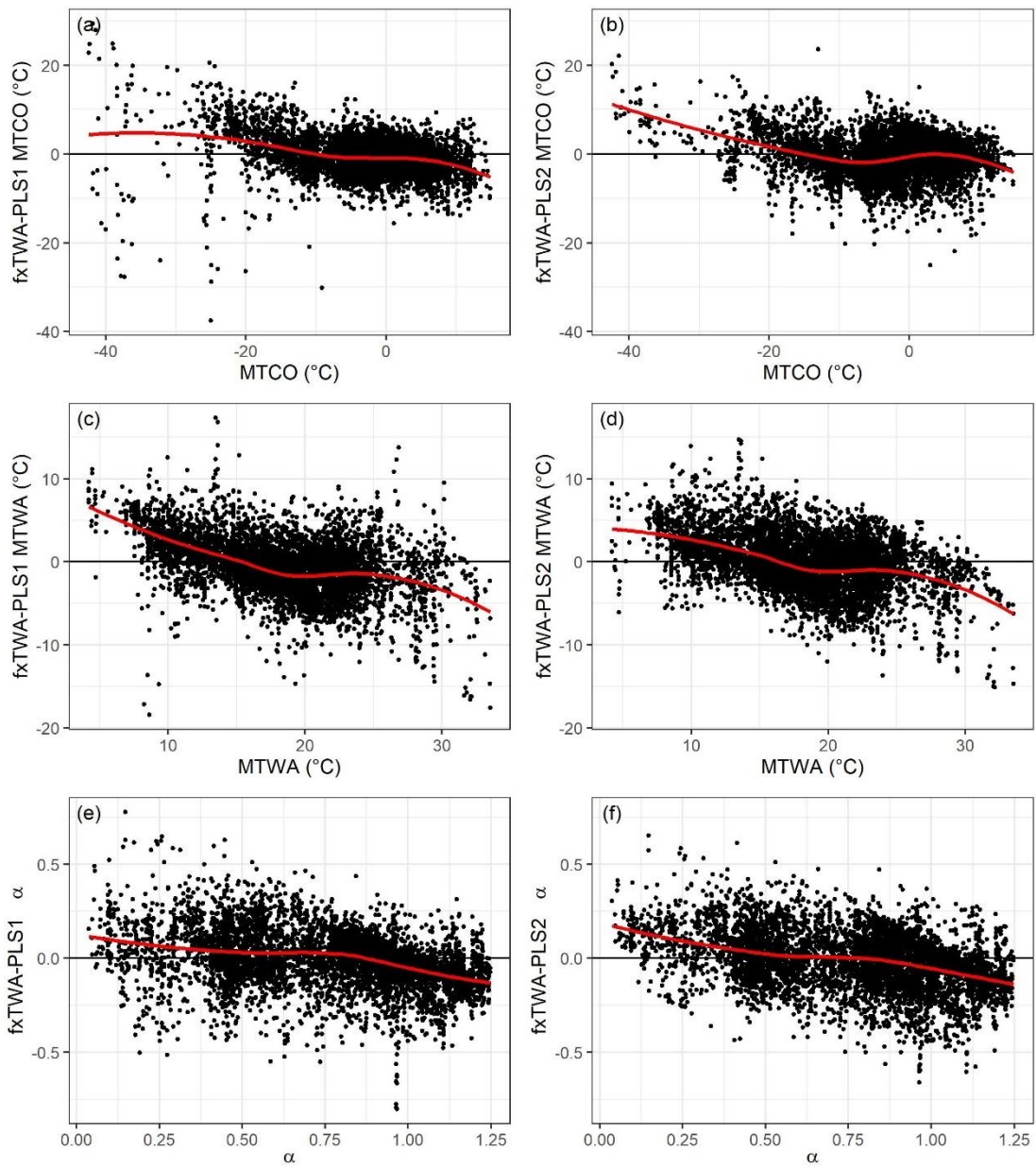

