# Peer review of "Holocene climates of the Iberian Peninsula: pollen-based reconstructions of changes in"

_Climate of the Past, 2021_

## Referee Comment (RC2)

Thank you for the opportunity to review for Climate of the Past the manuscript entitled "Holocene climates of the Iberian Peninsula: pollen-based reconstructions of changes in the west-east gradient of temperature and moisture" by Mengmeng Liu and coauthors.

I think that the paper of Mengmeng Liu et al. presents interesting findings in terms of results to be published in Climate of the Past but I also think that it cannot be published in its current version for several reasons.

-My first point concerns the choice of the method to reconstruct past climate changes. You have selected the WAPLS (a modified version of the transfer function): why the WAPLS and not the MAT or BRT? This method is not appropriate here with because the size of your modern pollen dataset (S1) is high and it covers a wide range of biomes and taxa. The WAPSL is useful at local and regional scale but may not be optimal in continental or global scale studies, as the responses of some pollen taxa to the variable of interest can be multimodal (Chevalier et al., 2020) as it is here. Moreover, its better to use a local calibration than a global one: global versus local calibrations (WAPLS) have been investigated in Dugerdil et al (2021). They show that WAPLS performs better for the local database than for global databases. Moreover, in your study, the relative contributions of individual taxa to the reconstructions of MTCO, MTWA and alpha (Table S2) raises some questions: most of these taxa are very rare in the Iberian Peninsula Holocene pollen records (Parrotia, Huperzia, Dryas, Zelkova….). These taxa can be recorded in the modern pollen dataset but are not representative of Holocene south Mediterranean pollen records. In this frame, I strongly recommend to resize your modern pollen dataset (by excluding biomes not recorded in the fossil assemblages or by spatial selection) and to recalibrate the WAPLS with a smaller but more appropriate training set.

Another way could be you need to validate your results by using another climate reconstruction method (MAT, BRT, RF for example) cf Salonen et al. works.

- I strongly recommend to add also regional composite panels (north, central, south?) of temperature and alpha changes instead of a unique composite curve (fig. 3). Regional climate patterns are important (fig 2) and the signal is too averaged if you only look at composite curves. You may miss important signal, so add a discussion about the additional panels and the regional patterns. This is a key point.

-The paper is too short (the description of the modern and fossil pollen datasets is too short, the discussion needs to be improved and some key figures are missing. In its current form, it's more a report paper than a discussion paper. Many points need further discussion (see below), and this is important because your paper will probably be a key paper.

> - I first suggest to better highlight the innovative side of this study. Your work and those of Tarroso et al (2016) (not cited in your paper!) focus on the reconstruction of the climate (temperature and precipitation) in Iberian Peninsula during the last 15000 years from pollen data. What's new in your paper?

- The paragraph on the modern pollen dataset is too short given that the accuracy of the modern pollen dataset is very important in transfer functions. The ref given for the modern pollen dataset (Harrison, 2019) is not a paper, so more details are needed; how do you calculate the climate parameters? Wordclim1, 2? Chelsea? How do you calculate alpha, which ref? Please add modern values of MTCO and MTWA as you did for alpha (S1). Moreover, the figure with climate values of the training set must be included in the text, not in the Supplementary.

- The paragraph on the fossil pollen dataset is also too short. In the ref cited for the fossil dataset (Shen et al., 2021 CPD) I just found a list of the taxa in the supplementary. It's not enough. Data have been extracted from Neotoma, Pangea, EPD? The description of the data sources of fossil pollen used to reconstruct the climate in the Iberian Peninsula (table S1) must be included directly here in the text and not in supplementary material. Table 1 must be updated with the origin of fossil pollen records: for each site, please add the references of the papers, information about the number of 14C date available, and the temporal range covered as for example, 8000-2000 cal yrs BP (not clear as it is in table S1: what does length mean?). Just keep in mind that without these pollen records you will not be able to provide such regional synthesis.

- The discussion need to be rewritten. **The synthesis figure (S8) must be updated and added in the text not in supplementary**. There is a lack of comparison of your results with the climate parameters available in the Mediterranean area: the study of Tarroso et al (2016) for Iberian Peninsula of course, Dormoy et al (2009) Combourieu-Nebout et al., (2013), Di Rita et al (2018), Jalali et al. (2016) for south Spain and western Mediterranean. It's important to add the curves of Tarroso et al., (2016) which are based on another climate reconstruction method (the PDF) in your figure to discuss regional patterns.

The discussion part on the CO2 impact must be removed, as you work on the Holocene not on the Lateglacial or LGM. You may replace this part by a more in depth discussion on data model comparison (too short!) and atmospheric circulation process.

Other points:

-How do you calculate alpha? A ref is needed. How do you explain values above 1?

- I don't agree with your sentence p 2, line 47 "much of the evidence of the Holocene climates is based on qualitative interpretations of vegetation changes…". A lot of other proxies are available: speleothems, chironomids, alkenones… all give independent **values** of climate parameters.

-- I don't agree with your sentence p 2, line 51 "most of the ca 50 sites from Iberia (Mauri et al 2015) were from the Pyrenees…". Please check and correct: in the Mauri's paper, at least 25 sites of the Iberian Peninsula are not from Pyrenean area and are not extrapolated!

- Some MTWA and MTCO anomalies values are very low for the Holocene period, especially for the last 6 ka: for example, some sites indicate -7° for MTWA (figs S5, S7), it's too low. Could you check your reconstructions?

- Ho do you take into account human impact in your modern and fossil pollen data? Usually we consider that the reconstruction of past climate for the last 2000 years are biased by human impact (check the IPA).

- fig S9: what is PACMEDY, please explain or add a reference.

I realize the authors may find my comments difficult to approach, but I sincerely hope they accept them as well-intentioned guidance. It should not be difficult to address them. Once concerns are addressed, I feel the manuscript will be much closer to being an outstanding contribution to knowledge in this time period.

---

## Author Comment (AC1)

Response to Reviewer 1

1. line 34 ' Projections of future climate change suggest that the region will become both warmer and drier, but nevertheless show that this west-east differentiation is maintained. ' Reference(s) is needed here.

There was some confusion here because we cited one of the Andrade et al. papers that deal with projections later, but we should have cited another paper at this point. We have added a reference to Andrade et al. (2021) which specifically shows the west-east gradients in RCP scenarios.

Andrade, C., Contente, J. and Santos, J. A.: Climate change projections of aridity conditions in the Iberian Peninsula, Water, 13(15), doi:10.3390/w13152035, 2021a.

2. line 16. 'early to mid-Holocene' Here, and in other instances in the text, the authors use terms like mid-Holocene, early Holocene, present, without having defined them explicitly. In particular, the reference to 'present' is relevant, and the manuscript often assesses Holocene temperatures compared to 'present temperatures'. Is 'present' the 20[th] average, pre-industrial, around 1950, temperatures?

The terminology for early Holocene and mid-Holocene is not clear. We did define the mid-Holocene as 8-4 ka (line 42), and since this is the period we are referring to in terms of the change in west-east gradient, we have modified the abstract to "mid-Holocene". Our comparisons to the present day are based on the comparing each bin to the most recent bin (0.5 ka ± 500 years); we have clarified this now in the text.

3. line 49 'Although these records are extensive, they seem to indicate fairly complex spatial patterns of change# I did not understand why the word 'although' needs to be used here. I do not see an implicit contradiction between being extensive and showing complex spatial patterns.

Indeed there is no contradiction. We have changed this to "These records are extensive and they seem to indicate fairly complex spatial patterns of change. "

4. line 53. Peninsular

We have now changed this to

"However, most of the ca 50 sites from Iberia were from the Pyrenees and the inferred patterns across the remainder of the Iberian Peninsula are therefore largely extrapolated."

line 53 'Furthermore, quantitative reconstructions of summer temperature made at individual sites using chironomid data (Muñoz Sobrino et al., 2013; Tarrats et al., 2018) are not consistent with reconstructed changes based on pollen for the same sites.' with reconstructed temperatures. Otherwise the sentence is grammatically somewhat odd.

Thank you. We have changed this as suggested to:

" Furthermore, quantitative reconstructions of summer temperature made at individual sites using chironomid data (Muñoz Sobrino et al., 2013; Tarrats et al., 2018) are not consistent with reconstructed temperatures based on pollen for the same sites. "

5. line 60 'We analyse how these trends are related to external forcing' I think the authors refer here to external climate forcing, but the sentence could be misinterpreted as meaning remote forcing, e.g. from the North Atlantic.

Yes, we do indeed mean external climate forcing and have changed the sentence to read:

" We analyse how these trends are related to changes in external climate forcing and quantify whether there are significant differences in west-east gradients through time. "

6. line 93 'We excluded individual pollen samples with large age uncertainties (standard error larger than 100 years)' What is the typical time resolution of the reconstructions? I think it is nowhere stated.

We are referring here to the standard error on the ages, but since this is not clear we have modified the sentence to read:

We excluded individual pollen samples with large uncertainties (standard error larger than 100 years) on the attributed age in the new age model.

We have also added a sentence here to specify the average resolution of the records:

As a result, the climate analyses are based on a fossil data set of 7121 pollen samples from 117 sites covering part or all of the last 12,000 years (Figure 1). The average temporal resolution of these records is 101 years.

7. line 105 'and assessed the significance of differences in these trends through time compared to 0.5 ka based on p values' compared to 0.5 ka? It is for me unclear.

We compared the 1000-year binned Holocene records to the 0.5 ka bin (0~1 ka) in order to avoid problems with post-industrial climate changes including recent anthropogenic changes in climate. In fact, it makes little difference to the detection of when the trends are different. We will clarify the choice of 0.5 ka as follows:

"..... and assessed the significance of differences in these trends through time compared to the most recent bin (0.5 ka ± 500 years) based on $p$ values ...."

8. line 141 'Summer temperatures are strongly correlated with changes in α' spatially or temporally correlated?

Both. Figure 5a shows correlations in the modern data set, and therefore reflects spatial correlations. Figure 5b shows correlations across the fossil samples and therefore shows spatial and temporal correlations. We will modify the sentence to clarify this as follows:

"Summer temperatures are strongly correlated with changes in α, both in terms of spatial correlations in the modern data set at a European scale and in terms of spatial and temporal correlations the fossil data set from Iberian Peninsula (Fig. 5)."

9. line 158 'their reconstructions show a cooling of 3°C in the early Holocene are comparable in magnitude# I guess that 'are' should be deleted.

Thanks for pointing this out, we have deleted this "are".

10. line 165 'change at some of the individual sites is much larger (ca 10°C) and there is no assessment off the uncertainty on these reconstructions. ' of

Thanks for pointing this out, we have corrected this typo.

11. line 193 'However, they show a persistent cooling of 1.5 °C compared to present between 4.5 and 2 ka, not seen in these reconstructions' do the authors mean persistent cooling trend or persistent cool conditions?

We meant persistently cool conditions. We have modified the sentence to make this clearer, as follows:

".... they show persistently conditions cooler than present by ca 1.5 °C between 4.5 and 2 ka, ...."

12. line 214 'Specifically, the increased advection of moisture into eastern Iberia created wetter conditions leading to increased evapotranspiration, less allocation of available net radiation to sensible heating, and resulting in cooler air temperatures.' I think the authors' point is not necessarily that increased evapotranspiration leads to colder temperatures, but rather to less temperature variations through time and space. The study suggests that summer temperatures did not fall as expected from the solar insolation alone, so this sentence is a bit confusing. Please, clarify.

We have shown that there is a change in the west-east gradient of moisture which implies that there is increased moisture advection into the eastern part of the Iberian Peninsula. Here we were trying to provide an explanation of how this would impact summer temperatures. We have rewritten the sentence to clarify this, as follows:

"The change in moisture gradient during the mid-Holocene, however, suggests an alternative explanation whereby changes in summer temperature are a response to land-surface feedbacks associated with changes in moisture. Specifically, the observed increased advection of moisture into eastern Iberia would have created wetter conditions there, which in turn would permit increased evapotranspiration, implying less allocation of available net radiation to sensible heating, and resulting in cooler air temperatures."

13. line 222 'Stronger moisture advection is not a feature of the transient climate model simulations, which may explain why these simulations do not show a strong modification of the insolation-driven changes in summer temperature. The failure of the current generation of climate models to simulate the observed strengthening of moisture transport into Europe and Eurasia during the mid-Holocene has been noted by other studies (e.g. Bartlein et al., 2017; Mauri et al., 2014)' The first sentence is confusing in view of Figure S9. The models do show temperature evolution through time. I think the authors mean that the models are not able to counteract or shield the insolation forcing. Also, is this conclusion (moisture advection) derived by other studies (then please cite references) or did the authors look into the

simulated moisture advection or is this conclusion reached by indirect reasoning? Please, be here as clear as possible.

The transient simulations do not show an increase in moisture advection during the mid-Holocene and the changes in summer temperature follow the changes in insolation. Our argument is therefore that this is why they are not consistent with our reconstructions. The Bartlein et al and Mauri et al papers both showed that the models do not advect sufficient moisture into Europe and Eurasia, but both focused on time-slice 6 ka simulations. Figure S9 only shows the regionally averaged changes in precipitation in the transient simulations and we agree that our point would be made more clearly if we showed the changes in the west-east gradient. We have now included such a plot in the Supplementary (Fig. S10).

Fig S10. The difference between the westmost and eastmost simulated mean daily precipitation in Iberian Peninsula between 8 ka and 0 ka, smoothed using 100 year bins. Here BP means before 1950 AD. The black lines represent Max Planck Institute Earth System Model (MPI) simulations, the red lines represent Alfred Wagner Insitute Earth System Model (AWI) simulations, the blue lines represent Institut Pierre Simon Laplace Climate Model (IPSL-CM5) TR5AS simulations, the orange lines represent Institut Pierre Simon Laplace Climate Model (IPSL-CM6) TR6AV simulations.

[Figure]

We have modified the text for clarification, as follows:

Stronger moisture advection is not a feature of the transient climate model simulations, which may explain why these simulations do not show a strong modification of the insolation-driven changes in summer temperature. The failure of the current generation of climate models to simulate the observed strengthening of moisture transport into Europe and Eurasia during the mid-Holocene has been noted by other studies (e.g. Bartlein et al., 2017; Mauri et

al., 2014) and is shown in Fig. S10. This data-model mismatch highlights the need for better modelling of land-surface feedbacks on atmospheric circulation and moisture.

14. Conclusion sections. The usual conclusion section is missing. This is to some extent a matter of style (or editorial guideline), but I find useful that a manuscript finishes off with a few bullet-point style list of most important take-home messages

We have added a concluding statement, as follows:

"We have used a pollen data set representing 117 sites across the Iberian Peninsula to make quantitative reconstructions of summer and winter temperature and an index of annual moisture through the Holocene. We show that the trends in winter temperature broadly follow the changes orbital forcing. Summer temperatures, however, do not follow the changes in orbital forcing but appear to be influenced by land-surface feedbacks associated with changes in the west-east gradient in moisture, which was considerably less pronounced during the mid-Holocene (8-4 ka)."

15. Fig S9. The reader will benefit from a reference to the model runs. I guess that the authors are using the runs described in Braconnot et al. (2019; doi:10.5194/cp-15-997-2019) and Bader et al. (2020; doi:10.1038/s41467-020-18478-6), but please, spell the names of the models in full, e.g. MPI-ESM-P, and give references to the runs used and shown in this figure. Also, some additional information could be useful for the reader as well, such as the spatial resolution. Also, the time axis is not clear enough: years BP?

We realise that we did not provide sufficient information about the transient simulations, and we have added text in the Supplementary to describe these simulations (and have referenced this in the main text), as follows:

Transient climate model simulations

We compared our reconstructions to outputs from four transient climate model simulations run as part of the PACMEDY project (https://pacmedy.lsce.ipsl.fr/wiki/doku.php), specifically two versions of the Institut Pierre Simon Laplace (IPSL) model, one with dynamic vegetation (IPSL-CM6, TR6AV) and one in which the dynamic vegetation was turned off (IPSL-CM5, TR5AS), version 2 of the Alfred Wagner Insitute (AWI) Earth System Model (AWI-ESM2) and the Max Planck Institute Earth System Model (MPI-ESM). The four simulations were forced by evolving orbital parameters and greenhouse gas concentrations. The four models have different spatial resolution, with the finest resolution being 1.875 x 1.875° (AWI, MPI) and the coarsest resolution being 1.875 x 3.75° (IPSL-CM5). Results from these palaeo-simulations have been discussed in Braconnot et al. (2019a, 2019b); Carré et al. (2021); Dallmeyer et al. (2019).

Here are the references:

Braconnot, P., Crétat, J., Marti, O., Balkanski, Y., Caubel, A., Cozic, A., Foujols, M.-A. and Sanogo, S.: Impact of multiscale variability on last 6,000 years Indian and West African monsoon rain, Geophys. Res. Lett., 46(23), 14021–14029, doi:https://doi.org/10.1029/2019GL084797, 2019a.

Braconnot, P., Zhu, D., Marti, O. and Servonnat, J.: Strengths and challenges for transient Mid- to Late Holocene simulations with dynamical vegetation, Clim. Past, 15(3), 997–1024, doi:10.5194/cp-15-997-2019, 2019b.

Carré, M., Braconnot, P., Elliot, M., d'Agostino, R., Schurer, A., Shi, X., Marti, O., Lohmann, G., Jungclaus, J., Cheddadi, R., Abdelkader di Carlo, I., Cardich, J., Ochoa, D., Salas Gismondi, R., Pérez, A., Romero, P. E., Turcq, B., Corrège, T. and Harrison, S. P.: High-resolution marine data and transient simulations support orbital forcing of ENSO amplitude since the mid-Holocene, Quat. Sci. Rev., 268, 107125, doi:https://doi.org/10.1016/j.quascirev.2021.107125, 2021.

Dallmeyer, A., Claussen, M., Lorenz, S. J. and Shanahan, T.: The end of the African humid period as seen by a transient comprehensive Earth system model simulation of the last 8000 years, , doi:10.5194/cp-2019-86, 2019.

We have also modified the time axis to "(yr BP)" and modified the caption of the Supplementary Figure to make it clear which models we have used, as follows:

" Simulated mean values of mean temperature of the coldest month (MTCO), mean temperature of the warmest month (MTWA) and mean daily precipitation in Iberian Peninsula between 8 ka and 0 ka, smoothed using 100 year bins. Here BP means before 1950 AD. The black lines represent Max Planck Institute Earth System Model (MPI) simulations, the red lines represent Alfred Wagner Institute Earth System Model (AWI) simulations, the blue lines represent Institut Pierre Simon Laplace Climate Model (IPSL-CM5) TR5AS simulations, the orange lines represent Institut Pierre Simon Laplace Climate Model (IPSL-CM6) TR6AV simulations."

---

## Author Comment (AC2)

Response to reviewer 2

Our responses are given in *blue italics* and proposed changes to the text are given in normal blue font.

-My first point concerns the choice of the method to reconstruct past climate changes. You have selected the WAPLS (a modified version of the transfer function): why the WAPLS and not the MAT or BRT? This method is not appropriate here with because the size of your modern pollen dataset (S1) is high and it covers a wide range of biomes and taxa. The WAPLS is useful at local and regional scale but may not be optimal in continental or global scale studies, as the responses of some pollen taxa to the variable of interest can be multimodal (Chevalier et al., 2020) as it is here.

*We do not use WA-PLS. As explained in the Methods section (lines 64-80), we use a modified version of fxTWA-PLS, which is explicitly designed to reduce compression biases that affect WA-PLS. The original version of fxTWA-PLS was published in Liu et al., 2020 and we provide detailed information about the new modification in the Appendix of the current paper. In the original paper (Liu et al., 2020), we compared the performance of fxTWA-PLS with the BRT approach as encoded in BUMPER. Although BUMPER performed better than the standard WA-PLS approach, it did not perform as well as fxTWA-PLS: the best BUMPER model had an RMSEP of 4.42, 882, 0.166 and $R^2$ of 0.74, 0.72, 0.71 for MTCO, $GDD_0$ and α compared to the fxTWA-PLS RMSEP of 4.37, 830, 0.148 and an $R^2$ of 0.76, 0.73, 0.72 for MTCO, $GDD_0$ and α in leave-one-out cross validation (BUMPER can't produce leave-pseudo-out cross validation as in Liu et al., 2020 and this paper, so we compared leave-one-out cross validation result); BUMPER model also has biased residuals in training (see Figure S5.2 in Liu et al., 2020). We provide an evaluation of the improved version of the fxTWA-PLS method using modern data in the current manuscript (Table A1, Figure A1, Figure A2), but we do not think it necessary to repeat the comparison with BUMPER here. In introducing this new methodology, we explain why it was used in preference to existing methods.*

*The importance of using a large and climatically extensive data set for pollen calibrations is increasingly recognised. The MAT-based reconstructions for Europe made by Mauri et al. (2015) includes 4700 samples covering Europe, the Middle East and Northern Africa. The Eurasian Modern Pollen Database (Chevalier et al., 2019; Davis et al., 2020), which includes over 8000 records and covers the much larger area of the Eurasian continent, was explicitly developed to serve as a calibration data set for pollen-based climate reconstructions. Analyses of the impact of the size of the training data set (Turner et al., 2020) show that training data sets that cover a more limited climate space result in poorer correlations between observed and reconstructed modern climates. Equation 2.14 in Liu et al. (2020) also shows that the standard error of the estimate to the climate will be reduced by increasing the number of taxa used. Small local calibration data sets can have better performance in some circumstances (see next response) but obviously make it impossible to reconstruct climate states outside the range of the modern climate in that locality.*

*It is difficult to make direct comparisons with the published MAT reconstructions for Europe by Mauri et al. (2015) because they reconstruct different climate variables. However, one of the known issues with the MAT approach is that it produces reconstructions that are more variable than e.g. Bayesian techniques and can also produce unrealistic jumps in the reconstructions because of switches between available analogues (see e.g. Brewer et al., 2008)*

*Multimodality in the response of pollen taxa to a specific climate variable often happens when there is inadequate sampling of the climate space (see e.g. Wei et al., 2020, Ecology), and to taxa with large tolerance and low abundance (see Supplementary Material 7 in Liu et al., 2020). In our original paper on fxTWA-PLS, we established that the multimodal peaks in abundance had almost no impact on the final climate reconstructions given the large number of taxa used (see Liu et al., 2020).*

Moreover, it's better to use a local calibration than a global one: global versus local calibrations (WAPLS) have been investigated in Dugerdil et al (2021). They show that WAPLS performs better for the local database than for global databases.

*For the reasons given above, we disagree that it is better to use a local calibration. The Dugerdil et al. (2021) paper shows that local calibration reduces the amplitude of the reconstructed climate changes compared to the global calibration. The reason that the local calibration gives a better result with WA-PLS is that the reconstructed climate is near to the 0-compression point in local calibration but far from the 0-compression point in global calibration. (see part 3c in Liu et al., 2020 for the explanation of 0-compression point.) The two 0-compression points can be very different in extreme climates, such as that examined by Dugerdil et al, and hence the difference between local calibration and global calibration shows up very clearly. However, the fact that they reduce the amplitude of climate changes indicates that the local calibration compresses the range of the reconstructions towards the central part of the climate range. This is exactly what out new method was designed to address and indeed the results shown in Appendix A show that there is reduced bias at the extremes of each climate variable. It is also interesting to note that the Dugerdil et al. paper shows that MAT performs less well than WA-PLS for the Mongolian and Baikal reconstructions.*

Moreover, in your study, the relative contributions of individual taxa to the reconstructions of MTCO, MTWA and alpha (Table S2) raises some questions: most of these taxa are very rare in the Iberian Peninsula Holocene pollen records (Parrotia, Huperzia, Dryas, Zelkova....). These taxa can be recorded in the modern pollen dataset but are not representative of Holocene south Mediterranean pollen records. In this frame, I strongly recommend to resize your modern pollen dataset (by excluding biomes not recorded in the fossil assemblages or by spatial selection) and to recalibrate the WAPLS with a smaller but more appropriate training set.

*The purpose of Table S2 is to illustrate the method since it shows the 10 most important taxa in the modern data set that contribute to making warmer/colder or wetter/drier reconstructions. It does not show the loadings of all the taxa that contribute to the reconstructions, only the top 10 in each category. We agree that some of the taxa in the current version are rare (although present in some samples) in the Holocene records for the Iberian Peninsula. For this reason, we have provided a new version in which we select the taxa that occur ≥ the median number of occurrences across samples and then show the top 10 of these taxa contributing to making reconstructions warmer/colder, wetter/drier. We have modified the caption describing this Figure, as follows:*

Table S2. Relative contributions of individual taxa to the reconstructions of mean temperature of the coldest month (MTCO), mean temperature of the warmest month (MTWA) and plant-available moisture (α). The plots show the top 10 taxa for each end of the climate gradient after

first screening out taxa that are relatively rare (i.e. occur < the median number of occurrences of all taxa in the fossil pollen record, which is 178 samples).

*The new version is shown below:*

| | MTCO | MTWA | | α |
|---|---|---|---|---|
| Increasing cold ↑ | *Abies* | *Ilex* | Increasing wet ↑ | *Myrica* |
| | *Tilia* | *Taxus* | | *Taxus* |
| | *Thalictrum* | *Saxifragaceae* | | *Ilex* |
| | *Betula* | *Myrica* | | *Calluna* |
| | *Onagraceae* | *Orobanchaceae* | | *Orobanchaceae* |
| | *Ericaceae* | *Calluna* | | *Sorbus* |
| | *Lycopodium* | *Potentilla* | | *Potentilla* |
| | *Salix* | *Onagraceae* | | *Betula* |
| | *Ulmus* | *Sorbus* | | *Onagraceae* |
| | *Orobanchaceae* | *Salix* | | *Ericaceae* |
| | ... | | | |
| Increasing warm ↓ | *Quercus.intermediate* | *Amaranthaceae* | Increasing dry ↓ | *Olea* |
| | *Phillyrea* | *Myrtaceae* | | *Myrtaceae* |
| | *Cistaceae* | *Cistus* | | *Quercus.evergreen* |
| | *Oleaceae* | *Amaryllidaceae* | | *Pistacia* |
| | *Cistus* | *Phillyrea* | | *Cistaceae* |
| | *Arbutus* | *Pistacia* | | *Amaryllidaceae* |
| | *Pistacia* | *Ephedra* | | *Cistus* |
| | *Myrtaceae* | *Thymelaeaceae* | | *Ephedra* |
| | *Ilex* | *Tamarix* | | *Tamarix* |
| | *Thymelaeaceae* | *Oleaceae* | | *Thymelaeaceae* |

*Obviously, taxa that are not present in the fossil samples do not contribute to the climate reconstructions for these samples. The inclusion of these taxa in the training data set does not impact the reconstructions for the fossil samples from Iberia. Similarly, the rare taxa in the Iberian fossil samples, even if they are strongly weighted towards one end of a climate gradient, will not contribute significantly to the climate reconstructions.*

*We have already explained why we do not think it necessary to use a smaller modern pollen data set. We would also like to point out that it is not possible to exclude modern samples from biomes that are not recorded in the fossil assemblages because this requires that the biomes present are known a priori.*

Another way could be you need to validate your results by using another climate reconstruction method (MAT, BRT, RF for example) cf Salonen et al. works.
*Comparison with other reconstruction methods does not provide a validation of the results. While several papers have used multiple reconstruction techniques at individual sites, these generally show that there are differences in the reconstructions based on different methods (see e.g. Brewer et al., 2008; Sinopoli et al., 2019) but cannot determine which is more correct except through comparison of goodness-of-fit and errors with the modern training data. As pointed out by Chevalier et al (2020) in their review of pollen-based climate reconstruction techniques, the primary purpose of using multiple techniques is to compare the methodologies rather than to determine which reconstructions are more accurate.*

*As we state in response to a previous comment, we compared the original version of fxTWA-PLS with BUMPER, which is a BRT approach, and have shown that our method produces better results. The modified version used in the current manuscript reduces compression bias even further and has better performance than our original version. Therefore, it's better to BUMPER reconstructions. Note that we compared our modified version and the original version based on leave-pseudo-out cross validation (see Appendix A), however, BUMPER can't produce leave-pseudo-out cross validation as in Liu et al. (2020) and this paper, so we compared leave-one-out cross validation result in Liu et al., 2020. Leave-one-out cross validation has inflated statistics than leave-pseudo-out cross validation, so the values of $R^2$, RMSEP and compression are not directly comparable between the leave-one-out BUMPER results in Liu et al. (2020) and the leave-pseudo-out modified fxTWA-PLS results in this paper.*
*We agree with the reviewer that it would be useful to make it very clear why we have chosen to use fxTWA-PLS instead of alternative methods, and we have therefore added text at the beginning of the methods section to clarify this and modified the current first paragraph to link it to the new text better, as follows:*

Multiple techniques have been developed to make quantitative climate reconstructions from pollen (see reviews in Bartlein et al., 2011; Salonenen et al., 2011; Chevalier et al., 2020). Modern analogue techniques (MAT: Overpeck et al., 1985) tend to produce rapid shifts in reconstructed values corresponding to changes in the selection of the specific analogue samples, although this tendency is less marked in the conceptually analogous response surface technique (Bartlein et al., 1986). Regression-based techniques, including weighted averaging methods such as Weighted Average Partial Least-Square (WA-PLS: ter Braak and Juggins, 1993), do not produce step-changes in the reconstructions but suffer from the tendency to compress the reconstructions towards the central part of the sampled climate range. However, this tendency can be substantially reduced by accounting for the sampling frequency (fx) and the climate tolerance of the pollen taxa present in the training data set (fxTWA-PLS: Liu et al., 2020). Bayesian approaches have also been applied to derive climate reconstructions from pollen assemblages (Peyron et al., 1998). However, comparison of fxTWA-PLS with the Bayesian model BUMPER (Holden et al., 2017), shows that fxTWA-PLS performs better in capturing the climate of the modern training data set from Europe (Liu et al., 2020).

Although fxTWA-PLS has clear advantages over other quantitative reconstructions techniques, there is still a slight tendency towards compression. We have therefore made a further modification to the approach as described in Liu et al. (2020). In the original version of fxTWA-PLS, the fx correction is applied as a weight with the form of $1/fx^2$ in the regression (step 7 in Table 1 in Liu et al., 2020). Here (see Appendix A) we make a further modification of fxTWA-PLS by (a) applying the fx correction separately in both the taxon calculation and the regression (step 2 and 7 in Table 1 in Liu et al., 2020) as a weight with the form of 1/fx and (b) applying P-splines smoothing (Eilers and Marx, 2021) in order to reduce the dependence of the fx estimation on bin width. The modified version further reduces the biases at the extremes of the sampled climate range. We used this modified version of fxTWA-PLS to reconstruct three climate variables: mean temperature of the coldest month (MTCO), mean temperature of the warmest month (MTWA) and plant-available moisture represented by α, an estimate of the ratio of actual evapotranspiration to equilibrium evapotranspiration. The individual and joint effects of MTCO, MTWA and α were tested explicitly using canonical correspondence analysis (CCA).

*Additional references:*

Bartlein PJ, Prentice IC, Webb T III (1986) Climatic response surfaces from pollen data for some eastern North American taxa. J Biogeogr 13:35–57

Chevalier, M., Davis, B.A.S., Heiri, O., Seppä, H., Chase, B.M., Gajewski, K., Lacourse, T., Telford, R.J., Finsinger, W., Guiot, J., Kühl, N., Maezumi, S.Y., Tipton, J.R., Carter, V.A., Brussel, T., Phelps, L.N., Dawson, A., Zanon, M., Vallé, F., Nolan, C., Mauri, A., de Vernal, A., Izumi, K., Holmström, L., Marsicek, J., Goring, S., Sommer, P.S., Chaput, M., Kupriyanov, D., 2020. Pollen-based climate reconstruction techniques for late Quaternary studies, Earth-Science Reviews, 210, 103384, https://doi.org/10.1016/j.earscirev.2020.103384.

Holden PB, Birks HJB, Brooks SJ, Bush MB, Hwang GM, Matthews-Bird F, Valencia BG, van Woesik R. 2017 BUMPER v1.0: a Bayesian user-friendly model for palaeo-environmental reconstruction. *Geosci. Model Dev.* **10**, 483–498. (doi:10.5194/gmd-10-483-2017)

Overpeck JT, Webb T III, Prentice IC (1985) Quantitative interpretation of fossil pollen spectra: dissimilarity coefficients and the method of modern analogs. Quat Res 23:87–108

Peyron O, Guiot J, Cheddadi R, Tarasov P, Reille M, de Beaulieu J-L, Bottema S, Andrieu V (1998) Climatic reconstruction in Europe for 18000 yr BP from pollen data. Quat Res 49:183–196

Salonen JS, Ilvonen L, Seppä H, Holmström L, Telford RJ, Gaidamavičˇius A, Stancˇikaitė M, Subetto D. 2011 Comparing different calibration methods (WA/WA-PLS regression and Bayesian modelling) and different-sized calibration sets in pollen-based quantitative climate reconstruction. *Holocene* **22**, 413–424. (doi:10.1177/0959683611425548)

ter Braak CJF, Juggins S. 1993 Weighted averaging partial least squares regression (WA-PLS): an improved method for reconstructing environmental variables from species assemblages. *Hydrobiologia* **269**, 485–502. (doi:10.1007/BF00028046)

- I strongly recommend to add also regional composite panels (north, central, south?) of temperature and alpha changes instead of a unique composite curve (fig. 3). Regional climate patterns are important (fig 2) and the signal is too averaged if you only look at composite curves. You may miss important signal, so add a discussion about the additional panels and the regional patterns.

*Figure 2 provides a summary of the results at individual sites. We have provided composite curves for the Iberian Peninsula to test explicit hypotheses about the controls of climate changes during the Holocene. While we agree that there might be interesting local patterns, there are good reasons for not making composites for smaller arbitrarily defined areas. Firstly, there is the question of how to divide Iberia into coherent regions: the current climate does not show straightforward north-south, east-west patterns. Furthermore, as we show in our analysis of the changes in moisture gradients, the appropriate coherent regions today would not necessarily be coherent in the past. Secondly, composites based on a limited number of sites will be inherently noisier. We will expand the section describing the construction of the composite curves in the Methods to clarify the purpose of these, as follows:*

In addition to examining the reconstructions for individual sites, we constructed composite curves for the Iberian Peninsula as a whole. The composite curves provide a way of

comparing the relationship between trends in the reconstructed climate changes and insolation changes. The curves were constructed after binning the site-based reconstructions using ± 500-year bins. We did 1000 bootstrap resampling of the reconstructed climate values in each ± 500-year bin to avoid the influence of a single value or a single site on the mean climate value in this bin, and use the standard deviation of the 1000 values to represent the uncertainty of the mean climate value. We constructed linear regression plots to examine the longitudinal and elevational patterns in the reconstructed climate variables, and assessed the significance of differences in these trends through time compared to the most recent bin (0.5 ka ± 500 years) based on $p$ values, with the customary threshold of 0.05. We then compared the climate trends with changes in summer and winter insolation.

- I first suggest to better highlight the innovative side of this study. Your work and those of Tarroso et al (2016) (not cited in your paper!) focus on the reconstruction of the climate (temperature and precipitation) in Iberian Peninsula during the last 15000 years from pollen data. What's new in your paper?

*The differences between our paper and the Tarroso et al. (2016) paper are (a) that we use a larger number of sites for the reconstructions, (b) we use a better reconstruction technique, and (c) they show no Holocene signal in temperature after 9 ka although they do show a trend in precipitation. In response to another comment, we have now added a comparison of the Tarroso et al reconstructions with our reconstructions and the other reconstructions available for Iberia. The principle focus of our paper, however, is to use reconstructions to investigate postulated changes in the west-east gradient of temperature and moisture (here represented by α, ratio of actual evapotranspiration to equilibrium evapotranspiration, rather than precipitation) through time. This focus is encapsulated in the title (Holocene climates of the Iberian Peninsula: pollen-based reconstructions of changes in the west-east gradient of temperature and moisture) and we discuss the nature of this gradient and the evidence for changes in the Holocene in the first paragraph of the Introduction. However, since this was obviously insufficiently clear to the reviewer, we will modify the final paragraph of the Introduction to be more explicit about this, as follows:*

Here we re-examine the trends in summer and winter temperature and plant-available moisture through the Holocene across Iberia, using a new and relatively comprehensive compilation of pollen data (Shen et al., 2021) with age models based on the latest radiocarbon calibration curve (IntCal20: Reimer et al., 2020). We explicitly test whether there are significant differences in the west-east gradient of moisture and temperature through time. We then analyse the relationships between the changes in the three climate variables and how trends in these variables are related to external climate forcing. These analyses allow us to confirm that the west-east gradient in moisture was less steep during the mid-Holocene and indicate the importance of changes in atmospheric circulation in explaining observed patterns of climate change across the region.

- The paragraph on the modern pollen dataset is too short given that the accuracy of the modern pollen dataset is very important in transfer functions. The ref given for the modern pollen dataset (Harrison, 2019) is not a paper, so more details are needed; how do you calculate the climate parameters? Wordclim1, 2? Chelsea? How do you calculate alpha, which ref?

*The Harrison, 2019 reference is to the modern pollen data set that we used, and the contents of that data set are described in the readme file. Please see response below about the*

*calculation of α, and also the expanded text describing the climate data set. We will also expand the description of the SMPDS as follows:*

The modern pollen training dataset was derived from the SPECIAL Modern Pollen Data Set (SMPDS: Harrison, 2019). The SMPDS consists of relative abundance records from 6458 terrestrial sites from Europe, northern Africa, the Middle East and northern Eurasia (SI Figure S1) assembled from multiple different published sources. The pollen records were taxonomically standardized, and filtered (as recommended by Chevalier et al, 2020) to remove obligate aquatics, insectivorous species, introduced species, and taxa that only occur in cultivation. Taxa (mainly herbaceous) with only sporadic occurrences were amalgamated to higher taxonomic levels (genus, sub-family or family) after ensuring consistency with their distribution in climate space. As a result of these amalgamations, the SMPDS contains data on 247 pollen taxa. For our analysis, we use the 195 taxa that occur at more than 10 sites.

*New reference:*
Chevalier, M., Davis, B.A.S., Heiri, O., Seppä, H., Chase, B.M., Gajewski, K., Lacourse, T., Telford, R.J., Finsinger, W., Guiot, J., Kühl, N., Maezumi, S.Y., Tipton, J.R., Carter, V.A., Brussel, T., Phelps, L.N., Dawson, A., Zanon, M., Vallé, F., Nolan, C., Mauri, A., de Vernal, A., Izumi, K., Holmström, L., Marsicek, J., Goring, S., Sommer, P.S., Chaput, M., Dmitry Kupriyanov, D., 2020. Pollen-based climate reconstruction techniques for late Quaternary studies. Earth-Science Reviews 210: 103384

Please add modern values of MTCO and MTWA as you did for alpha (S1). Moreover, the figure with climate values of the training set must be included in the text, not in the Supplementary.
*We have added two new panels to the Supplementary figure showing the modern values of MTCO and MTWA at the sites in the training data set.*

[Figure]

*Rather than moving these figures into the main text, which would not be appropriate given the focus of our paper, we have added a new two-panel figure showing the sites in climate space described by MTCO and MTWA, and MTWA and α respectively. This will now be Figure 1 and we will re-number the other figures accordingly.*

Figure 1. Climate space represented by mean temperature of the coldest month (MTCO), mean temperature of the warmest month (MTWA), and plant-available moisture as represented by α, an estimate of the ratio of actual evapotranspiration to equilibrium evapotranspiration. The grey points show climate values for a rectangular area (21° W ~ 150° E, 29° N ~ 82° N) enclosing the SMPDS data set, derived from the Climate Research Unit CRU CL 2.0 database (New et al., 2002). The black points show climate values of the SMPDS dataset. The red points show climate values of the Iberian Peninsula region in the SMPDS dataset.

[Figure]

*We have expanded the text describing the climate data as follows:*

Modern climate data at each of the sites in the training data set were obtained from Harrison (2019). This data set contains climate reconstructions of MTCO, growing degree days above a baseline of 0° C ($GDD_0$) and a moisture index (MI), defined as the ratio of annual precipitation to annual potential evapotranspiration. The climate at each site was obtained using geographically-weighted regression of the CRU CL v2.0 gridded dataset of modern (1961-1990) surface climate at 10 arc minute resolution (New et al., 2002) in order to correct for elevation differences between each pollen site and the corresponding grid cell. The geographically-weighted regression used a fixed bandwidth kernel of 1.06 ° (~140km) to optimize model diagnostics and reduce spatial clustering of residuals relative to other bandwidths. The climate of each pollen site was then estimated based on its longitude, latitude, and elevation. MTCO and $GDD_0$ was taken directly from the GWR regression and MI was calculated for each pollen site using code modified from SPLASH v1.0 (Davis et al., 2017) based on daily values of precipitation, temperature and sunshine hours again obtained using a mean-conserving interpolation of the monthly values of each. For this application, we used MTCO directly from the data set but calculated MTWA from MTCO and $GDD_0$, based on the relationship between MTCO, MTWA and $GDD_0$ given by Appendix 2 of Wei et al. (2021). We derived α from MI following Liu et al. (2020). The modern training data set provides records spanning a range of MTCO from – 42.4 °C to 14.8 °C, of MTWA from 4.2 °C to 33.5 °C, and of α from 0.04 to 1.25 (Figure 1, SI Figure 1).

*Additional references:*

Davis, T. W., I. C. Prentice, B. D. Stocker, R. T. Thomas, R. J. Whitley, H. Wang, B. J. Evans, A. V. Gallego-Sala, M. T. Sykes, and W. Cramer. 2017 Simple process-led algorithms for simulating habitats (SPLASH v.1.0): Robust indices of radiation, evapotranspiration and plant-available moisture. *Geoscientific Model Development* **10**: 689-708, https://doi.org/10.5194/gmd-10-689-2017

New M., Lister D., Hulme M., Makin I., 2002. A high-resolution data set of surface climate over global land areas. Climate Research 21, 1–25. https://doi.org/10.3354/cr021001.

- The paragraph on the fossil pollen dataset is also too short. In the ref cited for the fossil dataset (Shen et al., 2021 CPD) I just found a list of the taxa in the supplementary. It's not enough. Data have been extracted from Neotoma, Pangea, EPD?
*We should have provided a full reference for the data set rather than saying where we obtained it. The data set description now provides information on whether the data were obtained from the original authors or from a public-access data set.*

The fossil pollen data from the Iberian Peninsula were compiled by Shen et al. (2021) and the data set (Harrison et al., 2021) was obtained from https://doi.org/10.17864/1947.000343. The taxonomy used by Shen et al. (2021) is consistent with that employed in the SMPDS. Shen et al. (2021) provides consistent age models for all the records based on the IntCal20 calibration curve (Reimer et al., 2020) and the BACON Bayesian age-modelling tool (Blaauw et al., 2021; Blaauw and Christeny, 2011) using the supervised modelling approach implemented in the ageR package (Villegas-Diaz et al, 2021). We excluded individual pollen samples with large uncertainties (standard error larger than 100 years) on the attributed in the new age model. As a result, the climate reconstructions are based on a fossil data set of 7294 pollen samples from 117 records covering part or all of the last 12,000 years (Figure 2), with 42 individual records provided by the original authors, 73 records obtained from the European Pollen Database (EPD, www.europeanpollendatabase.net) and 2 records from PANGAEA (www.pangaea.de/). Details of the records are given in SI Table S1. The average temporal resolution of these records is 101 years. We then excluded a few samples where the reconstructed values of α exceed the natural limit of 0 and 1.26. Finally, 7121 samples from 117 records are used for the analyses of the climate reconstructions.

*The revised version of Table S1 in this paper and the Supplementary to the Shen et al. paper now provide a list of sites, the source for each site and the original references.*

The description of the data sources of fossil pollen used to reconstruct the climate in the Iberian Peninsula (table S1) must be included directly here in the text and not in supplementary material. Table S1 must be updated with the origin of fossil pollen records: for each site, please add the references of the papers, information about the number of 14C date available, and the temporal range covered as for example, 8000-2000 cal yrs BP (not clear as it is in table S1: what does length mean?).
*We do not think it is necessary to move this large table from the Supplementary into the main text, particularly since we have expanded it as suggested by the reviewer. We have added the source of each record and the publications to Table S1 (as in the Supplementary to Shen et al., 2021). We have also added the number of dating points used to construct the age models - noting that some of these sites have other types of date than radiocarbon. We emphasised the length of the period covered and the number of samples available in the original version of this table because this is important for temporal resolution. However, we have now added the start and end dates of each record.*

There is a lack of comparison of your results with the climate parameters available in the Mediterranean area: the study of Tarroso et al (2016) for Iberian Peninsula of course, Dormoy et al (2009), Combourieu-Nebout et al., (2013), Di Rita et al (2018), Jalali et al. (2016) for south Spain and western Mediterranean. It's important to add the curves of

Tarroso et al., (2016) which are based on another climate reconstruction method (the PDF) in your figure to discuss regional patterns.

*We did not originally include the Tarroso et al. (2016) paper because they show almost no change in either MTWA or MTCO during 9 ~ 3 ka. We suspect that this is a problem with the methodology - in that the modern distribution data is based on species occurrence data from Atlas Florae Europeae or the Global Biodiversity Information Facility (GBIF), both of which have incomplete coverage. Although they validated the PDFs using core top data from the 31 sites, they did not perform a wider validation using surface samples e.g. from the European Modern Pollen Dataset. Nevertheless, we are happy to cite this paper as a source of quantitative climate reconstructions for the region (and we will also take the opportunity to mention the Kaufman et al data set that we refer to later), and will expand the text in the Introduction to do so as follows:*

However, much of the evidence for Holocene climates of the Iberian Peninsula is based on qualitative interpretations of vegetation changes, generally interpreted as reflecting changes in moisture availability (Morellón et al., 2018). These records are extensive and they seem to indicate fairly complex spatial patterns of change. Kaufman et al. (2020) provides quantitative reconstructions of summer and winter temperature in their compilation of Holocene climate information, but there are only 5 terrestrial sites from the Iberian Peninsula. Iberia was also included in the quantitative pollen-based reconstructions of European climate through the Holocene (Mauri et al., 2015). However, the geographical distribution of sites included is uneven and a large fraction of the records were from the Pyrenees and the Cantabrian mountains, with additional clustering of sites in coastal regions. Thus, the inferred patterns of climate over most of the central part of the Peninsula are therefore largely extrapolated. Tarroso et al. (2016) has provided reconstructions of summer and winter temperature and mean annual precipitation since the Last Glacial Maximum for the Iberian Peninsula, by using modern species distribution data to develop climate probability distribution functions (PDFs) and applying these to 31 fossil records. However, although they identified trends in precipitation during the Holocene, the temperature reconstructions do not seem to be reliable since they show no changes through time (9 ~ 3 ka), either for the Iberian Peninsula as a whole or for individual sub-regions, in contra-distinction to the other reconstructions. The current state of uncertainty about Holocene climate changes in Iberia is further exacerbated because quantitative reconstructions of summer temperature made at individual sites using chironomid data (Muñoz Sobrino et al., 2013; Tarrats et al., 2018) are not consistent with reconstructed summer temperatures based on pollen for the same sites.

*We have included plots of the reconstructed MTCO and MTWA from Tarroso et al. in Supplementary Figure S8. However, α can't be directly transformed from precipitation due to a lack of other parameters.*

Figure S8. Comparison between reconstructed composite changes in climate anomalies. The first column represents this paper, the second column represents Mauri et al. (2015), the third column represents Kaufman et al. (2020), the fourth column represents Tarroso et al. (2016). The composite curves from this paper and Kaufman et al. (2020) are calculated from individual reconstructions, using anomalies to 0.5 ka and a bin of ± 500 years (time slices are 0.5, 1.5, …, 11.5 ka). The composite curves from Mauri et al. (2015) are converted directly from the gridded time slices which are provided with anomalies to 0.1 ka and a bin of ± 500 years (time slices are 1, 2, …, 12 ka). The composite curves from Tarroso et al. (2016) are also converted directly from the gridded time slices provided, with anomalies to 0.5 ka and a bin of ± 500 years (time

slices are 3, 4, …, 12 ka). Note that Tarroso et al. (2016) applied a smoothing to the data such that the plots in the paper do not show the excursion in MTWA at 8 ka. In all of the plots, the black lines show mean values across sites, with vertical line bars showing the standard deviation of mean values using 1000 bootstrap cycles of site/grid resampling.

[Figure]

*We will revise the Discussion section to include a comparison of these results with our reconstructions, as follows:*

We have shown that there was a gradual increase in MTCO over the Holocene, both for most of the individual sites represented in the data set and for Iberia as a whole. Colder winters in southern Europe during the mid-Holocene (6 ka) are a feature of many earlier reconstructions (e.g. Cheddadi et al., 1997; Wu et al., 2007). A general warming trend over the Holocene is seen in gridded reconstructions of winter season (December, January, February) temperatures as reconstructed using the modern analogue approach by Mauri et al. (2015), although there is somewhat less millennial-scale variability in these reconstructions (SI Fig. S8). Nevertheless, their reconstructions show a cooling of 3°C in the early Holocene comparable in magnitude to the ca 4°C cooling at 11.5 ka reconstructed here. Although they show conditions slightly cooler than present persisting up to 1 ka, the differences are very small (ca 0.5°C) after 2 ka, again consistent with our reconstructions of MTCO similar to present by 2.5 ka. Quantitative reconstructions of winter temperature for the 5 terrestrial sites from the Iberian Peninsula in the Kaufman et al. (2020) compilation all show a general trend of winter warming over the Holocene, but the magnitude of the change at some of the individual sites is much larger (ca 10°C) and there is no assessment of the uncertainty on these reconstructions. The composite curve of Kaufman et al. (2020) shows an increasing trend in MTCO through the Holocene although with large uncertainties (SI Fig. S8). In contrast to the consistency of the increasing trend in MTCO during the Holocene between our reconstructions and those of Mauri et al. (2015) and Kaufman et al. (2020), there is no discernible trend in MTCO during the Holocene reconstruction of Tarroso et al. (2016). Indeed, there is no significant change in their MTCO values after ca 9 ka, either for the Peninsula as a whole (SI Fig. S8) or for any of the four sub-regions they considered.

*When discussing the MTWA trends, we will add:*

The differences between the three data sets probably reflect differences in the number of records used, but the lack of coherency points to there not being a strong, regionally coherent signal of summer temperature changes during the Holocene. Tarroso et al (2016) also showed no significant changes in MTWA after ca 9 ka (SI Fig. S8).

*The Dormoy et al (2009), Combourieu-Nebout et al., (2013), Jalali et al. (2016) and Di Rita et al (2018) papers do not provide reconstructions from terrestrial sites from the Iberian Peninsula, although Demoy et al (2009) and Di Rita et al (2018) include reconstructions respectively for one/two marine records south of Iberia. Given our focus on the climate gradients across the Iberian Peninsula, it does not seem appropriate to cite these papers. A pan-Mediterranean analysis of changes in temperature and moisture gradients during the Holocene is beyond the scope of the current paper.*

The synthesis figure (S8) must be updated and added in the text not in supplementary. *We have updated this Figure to include the Tarroso et al. (2016) curves for MTWA and MTCO. However, we do not think it is necessary to move this Figure into the main text. Our purpose here is to discuss the degree to which our reconstructions are consistent (or not) with previous reconstructions, but we are not aiming to provide detailed comparisons of the methods used or to evaluate which of these reconstructions is most accurate.*

The discussion part on the CO₂ impact must be removed, as you work on the Holocene not on the Late glacial or LGM.

*We disagree. We include this because we have previously published on the potential impact of CO₂ on quantitative climate reconstructions based on modern training data sets, and furthermore have developed a robust method to account for this based on known plant physiology responses linking ambient CO₂ levels with changing water use efficiency. Our point here is that this will have an impact, even during the Holocene (see e.g. Figure 6 in Wei et al, 2019). However, the impact of a 40 ppm reduction in CO₂ on reconstructed moisture is less than the uncertainties in our Holocene reconstructions, and will not affect the reconstruction of changes in the west-east gradient, and this is why we do not make this correction in the current analyses. Rather than removing this text, we will clarify why it could be an issue and why we do not think it important for the conclusions of the current paper as follows:*

Nevertheless, climate is not the only driver of vegetation changes. On glacial-interglacial timescales, changes in $CO_2$ have a direct impact on plant physiological processes and reductions in plant water-use efficiency at low $CO_2$ result in vegetation appearing to reflect drier conditions than were experienced in reality (Farquhar, 1997; Gerhart and Ward, 2010; Prentice et al., 2017; Prentice and Harrison, 2009). The difference between post- and pre-industrial $CO_2$ levels could also influence the reliability of moisture reconstructions based on modern training data sets. However, the change in $CO_2$ over the Holocene was only 40 ppm. Prentice et al. (2022) shows that this change relative to modern levels has only a small impact on pollen-based reconstructed moisture indices. The magnitude of this impact is within the uncertainties on our reconstructions. Furthermore, accounting for the effect of this change in $CO_2$ or not won't affect the reconstructed west-east gradient through time. Therefore, we have not accounted for the impact of changing $CO_2$ in our reconstructions of α, although there are techniques to do this (Prentice et al., 2011, 2017; Wei et al., 2021).

*We will update the Prentice et al. reference, originally given as 2021, to 2022.*

You may replace this part by a more in depth discussion on data model comparison (too short!) and atmospheric circulation process.

*The key point that we want to make in referring to climate model simulations is that they do not show increased moisture advection and this is why the simulated changes in summer temperature are inconsistent with the reconstructions. This point has been made before in the papers we cite with respect to mid-Holocene changes across Europe and in central Eurasia. We do not need to make detailed data-model comparisons for this. Nevertheless, we will expand the Discussion to make the evidence clearer, as follows:*

We have shown that stronger moisture advection is not a feature of transient climate model simulations of the Holocene, which may explain why these simulations do not show a strong modification of the insolation-driven changes in summer temperature (Fig. S9). Although the amplitude differs, all of the models show a general decline in summer temperature. The failure of the current generation of climate models to simulate the observed strengthening of moisture transport into Europe and Eurasia during the mid-Holocene has been noted for previous versions of these models (e.g. Bartlein et al., 2017; Mauri et al., 2014) and also shown in Fig. S10. Mauri et al. (2014), for example, showed that climate models participating in the last phase of the Coupled Model Intercomparison Project (CMIP5/PMIP3) were unable to reproduce reconstructed climate patterns over Europe at 6000 yr B.P. and indicated that

this resulted from over-sensitivity to changes in insolation forcing and the failure to simulate increased moisture transport into the continent. Bartlein et al. (2017) showed that the CMIP5/PMIP3 models simulated warmer and drier conditions in mid-continental Eurasia at 6000 yr B.P., inconsistent with palaeo-environmental reconstructions from the region, as a result of the simulated reduction in the zonal temperature gradient which resulted in weaker westerly flow and reduced moisture fluxes into the mid-continent. They also pointed out the strong feedback between drier conditions and summer temperatures. The drying of the mid-continent is also a strong feature of the mid-Holocene simulations made with the current generation of CMIP6/PMIP4 models (Brierley et al., 2020). The persistence of these data-model mismatches highlights the need for better modelling of land-surface feedbacks on atmospheric circulation and moisture.

*New reference*
Brierley, C., Zhao, A., Harrison, S.P., Braconnot, P., Williams, C., Thornalley, D., Shi, X., Peterschmitt, J-Y., Ohgaito, R., Kaufman, D.S., Kagayama, M., Hargreaves, J.C., Erb, M., Emile-Geay, J., D'Agostino, R., Chandan, D., Carré, M., Bartlein, P.J., Zheng, W., Zhang, Z., Zhang, Q., Yang, H., Volodin, E.M., Routsen, C., Peltier, W.R., Otto-Bliesner, B., Morozova, P.A., McKay, N.P., Lohmann, G., LeGrande, A.N., Guo, C., Cao, J., Brady, E., Annan, J.D., Abe-Ouchi, A., 2020. Large-scale features and evaluation of the PMIP4-CMIP6 *midHolocene* simulations. *Climate of the Past* 16: 1847-1872.

Fig S10. The difference between the westmost and eastmost simulated mean daily precipitation in Iberian Peninsula between 8 ka and 0 ka, smoothed using 100 year bins. Here BP means before 1950 AD. The black lines represent Max Planck Institute Earth System Model (MPI) simulations, the red lines represent Alfred Wagner Insitute Earth System Model (AWI) simulations, the blue lines represent Institut Pierre Simon Laplace Climate Model (IPSL-CM5) TR5AS simulations, the orange lines represent Institut Pierre Simon Laplace Climate Model (IPSL-CM6) TR6AV simulations.

[Figure]

-How do you calculate alpha? A ref is needed. How do you explain values above 1?

*The calculation of α is based on the Priestley-Taylor formulation and as such has a range between 0 and 1.26 (see Davis et al., 2017 and Supplementary Material 3 in Liu et al., 2020). As explained in the revised text about the climate data given in response to an earlier comment about these data, we derived site-based climate values from Harrison et al. (2019) and we calculated α from the MI values provided in that data set, using the following equation:*

$$\alpha = 1.26 \cdot MI \cdot \left( 1 + \frac{1}{MI} - \left( 1 + \left( \frac{1}{MI} \right)^{\omega} \right)^{\frac{1}{\omega}} \right)$$

*using a value for ω of 3, as in Liu et al. (2020).*

- I don't agree with your sentence p 2, line 47 "much of the evidence of the Holocene climates is based on qualitative interpretations of vegetation changes...". A lot of other proxies are available: speleothems, chironomids, alkenones... all give independent **values** of climate parameters.
*We were not precise enough here. We are in fact referring to the evidence for Holocene climates of the Iberian Peninsula. There are two quantitative chironomid reconstructions from Iberia, and we do indeed compare our reconstructions with these (lines 186 et seq). There are speleothem records from Iberia, but these provide information about oxygen isotopic changes. While these are used to infer changes in precipitation and (in some cases) temperature, they are not a direct quantitative estimate of the climate parameters, and indeed in some cases it is difficult to infer what specifically is driving the changes in isotopic composition (see e.g. Parker et al., 2021). There are alkenone records from the seas around Iberia, but these provide estimates of sea-surface temperature and so are not directly comparable with our reconstructions. We will make the meaning of this sentence clearer as follows:*

However, much of the evidence for Holocene climates of the Iberian Peninsula is based on qualitative interpretations of vegetation changes, generally interpreted as reflecting changes in moisture availability (Morellón et al., 2018). These records are extensive, they seem to indicate fairly complex spatial patterns of change.

-- I don't agree with your sentence p 2, line 51 "most of the ca 50 sites from Iberia (Mauri et al 2015) were from the Pyrenees...". Please check and correct: in the Mauri's paper, at least 25 sites of the Iberian Peninsula are not from Pyrenean area and are not extrapolated!
*Since this is a gridded data set, and there are large areas of the Peninsula which are not represented in the Mauri et al data set, the inferred patterns of change are indeed extrapolated. A substantial proportion of the sites in the data set are from the Pyrenees and the Cantabrian mountains. The rather "blobby" reconstructions for Iberia in this paper compared to other parts of Europe suggest that individual sites are playing a large role in*

*the extrapolated surfaces. We will be more precise in our description of the data set and the importance of site distribution in creating gridded surfaces, as follows:*

Iberia was also included in the quantitative pollen-based reconstructions of European climate through the Holocene (Mauri et al., 2015). However, the geographical distribution of sites included is uneven and a large fraction of the records were from the Pyrenees and the Cantabrian mountains, with additional clustering of sites in coastal regions. Thus, the inferred patterns of climate over most of the Peninsula are largely extrapolated.

- Some MTWA and MTCO anomalies values are very low for the Holocene period, especially for the last 6 ka: for example, some sites indicate -7° for MTWA (figs S5, S7), it's too low. Could you check your reconstructions?

*There are indeed three individual data points that indicate values of -7° for MTWA (figs S5, S7). These individual samples may be depauperate or otherwise unreliable. In previous work (e.g. Wei et al., 2021) we have removed suspect samples of this sort. In the absence of evidence to exclude these specific samples, we have not excluded them here. However, their contribution to the composite curve is negligible.*

- How do you take into account human impact in your modern and fossil pollen data? Usually, we consider that the reconstruction of past climate for the last 2000 years is biased by human impact (check the IPA).

*We have removed introduced and cultivated species from our training data set (see revised text describing this data set above) in order to focus on species that can be expected to be diagnostic of climate. We do not otherwise take account of potential human impact on the pollen assemblages. Attempts to quantify human impacts on the vegetation of Europe (e.g. Marquer et al., 2017; Roberts et al., 2018) have only limited coverage for Iberia, although they do imply that major anthropogenic changes in forest cover in northern Iberia occurred only in the last 2-3000 years. There is no obvious break in our reconstructions at this time that would suggest they are less reliable because of human influence.*

- fig S9: what is PACMEDY, please explain or add a reference.

*PACMEDY was the "PAleao-Constraints on Monsoon Evolution and Dynamics" project which coordinated the transient climate models simulations. Rather than adding a reference to the project, we will add the references to the publications describing the individual simulations to Supplementary, as follows:*

We compared our reconstructions to outputs from four transient climate model simulations run as part of the "PAleao-Constraints on Monsoon Evolution and Dynamics" (PACMEDY) project (https://pacmedy.lsce.ipsl.fr/wiki/doku.php): version 1.2 of the MPI (Max Planck Institute) Earth System model (Dallmeyer et al., 2020), version 2 of the AWI (Alfred Wegener Institute) Earth System model (Sidorenko et al., 2019), a version of the IPSL (Institut Pierre Simon Laplace) Earth system model with prescribed vegetation (IPSL-CM5, TR5AS), and one with a dynamic vegetation module (IPSL-CM6, TR6AV) (Braconnot et al., 2019a, 2019b). The four simulations were forced by evolving orbital parameters and greenhouse gas concentrations. The four models have different spatial resolution, with the finest resolution

being 1.875 x 1.875° (AWI, MPI) and the coarsest resolution being 1.875 x 3.75° (IPSL-CM5, TR5AS).

Fig S9. Simulated mean values of mean temperature of the coldest month (MTCO), mean temperature of the warmest month (MTWA) and mean daily precipitation in Iberian Peninsula between 8 ka and 0 ka, smoothed using 100 year bins. Here BP means before 1950 AD. The black lines represent Max Planck Institute Earth System Model (MPI) simulations, the red lines represent Alfred Wagner Insitute Earth System Model (AWI) simulations, the blue lines represent Institut Pierre Simon Laplace Climate Model (IPSL-CM5) TR5AS simulations, the orange lines represent Institut Pierre Simon Laplace Climate Model (IPSL-CM6) TR6AV simulations.

*Additional references*

Braconnot, P., Crétat, J., Marti, O., Balkanski, Y., Caubel, A., Cozic, A., Foujols, M.-A. and Sanogo, S.: Impact of multiscale variability on last 6,000 years Indian and West African monsoon rain, Geophys. Res. Lett., 46(23), 14021–14029, doi:https://doi.org/10.1029/2019GL084797, 2019a.

Braconnot, P., Zhu, D., Marti, O. and Servonnat, J.: Strengths and challenges for transient Mid- to Late Holocene simulations with dynamical vegetation, Clim. Past, 15(3), 997–1024, doi:10.5194/cp-15-997-2019, 2019b.

Dallmeyer, A., Claussen, M., Lorenz, S. J. and Shanahan, T.: The end of the African humid period as seen by a transient comprehensive Earth system model simulation of the last 8000 years, , doi:10.5194/cp-2019-86, 2020.

Sidorenko, D., Goessling, H. F., Koldunov, N. V, Scholz, P., Danilov, S., Barbi, D., Cabos, W., Gurses, O., Harig, S., Hinrichs, C., Juricke, S., Lohmann, G., Losch, M., Mu, L., Rackow, T., Rakowsky, N., Sein, D., Semmler, T., Shi, X., Stepanek, C., Streffing, J., Wang, Q., Wekerle, C., Yang, H. and Jung, T.: Evaluation of FESOM2.0 coupled to ECHAM6.3: Preindustrial and HighResMIP simulations, J. Adv. Model. Earth Syst., 11(11), 3794–3815, doi:https://doi.org/10.1029/2019MS001696, 2019.

---

## Referee Report (RR1)

**Comments on "Holocene climates of the Iberian Peninsula: pollen-based reconstructions of changes in 1 the west-east gradient of temperature and moisture"**

The manuscript provides new insights in the field of paleoclimate studies, and in particular, in quantitative paleoclimate reconstructions. The authors have used a recently developed technique (fxTWA-PLS method) for reconstructing three climatic variables (MTCO, MTWA and plant-available moisture) in the Iberian Peninsula during the last 11.5 ka.

**Main points**

One of the main goal of the paper is to use the recently developed fxTWA-PLS method (Liu et al., 2020). Using this method provides new knowledge in the area of quantitative paleoclimate reconstructions, and therefore, it is one of the main points why the paper is worth publishing. However, since this is a new method and very few pollen records have used this methodology (as far as I know: Liu et al., 2020; Wei et al., 2020; Prentice et al., 2022), more information should be included. For example, the authors mention that the WAPLS suffers from the tendency to compress the reconstructions towards the central part of the sampled climate range (Line 87-88) but they do not mention that the WAPLS is robust to spatial autocorrelation (Telford et al., 2005). Has this method been demonstrated to be robust to spatial autocorrelation?

With respect to fossil pollen records, the authors did a great work compiling a large number of sites, using different data sources such as the European Pollen Database (EPD), Pangaea and even contacting directly with the authors from the original studies ("Author" in the excel file "Iberia_pollen_records_v3_0307.xlsx"). However, the authors should have also checked the relatively new and opened ACER database (Sánchez-Goñi et al., 2017), which provides high-resolution global-scale fossil pollen data. Did the authors checked the high-resolution Spanish pollen records from this database? In addition, other new high-resolution Holocene Iberian pollen records should have also been included in the paper, such as lake Medina or Padul (both in southern Spain), especially when the number of fossil pollen records in the southern Iberian Peninsula is not as good as in the north, which could lead to uncertainties in the reconstructions and interpretations (as you mentioned in Lines 350-354).

I really appreciate the work done on the re-calibration of the age models. The authors have calibrated the age-models of the fossil records based on the newest IntCal20 calibration curve (Reimer et al., 2020). This work along with the removal of the samples with large uncertainties (stnd error >100yr) make the paper reliable from the point of view of the age-model and age uncertainties.

As we know, the pollen-based quantitative reconstructions are controlled by the modern pollen training dataset, and thus by the modern pollen vegetation. Although the MTCO reconstructions appear to be consistent with climate model simulations and consequence of insolation forcing (Lines 233-237), I wonder whether the MTCO reconstructions are completely reliable. Looking at the statistical performance of the MTCO ($R^2$ 0.75 for Component 4, Table 1), the MTCO reconstructions should be reliable. However, the reconstructed MTCO anomalies at individual sites (Fig. 3) do not show a clear E-W trend (neither at low nor high altitude). This seems to be a consequence of the recent MTCO variability over a NW-SE (or E-W) transect (Figure 2), which shows no clear E-W changes in temperature. Actually, the MTCO in Figure 2 seems to be related

with elevation changes in Iberia, with the lowest temperatures in Pyrenees, Central and Iberian mountain chains, etc. Although the winter temperature can influence the vegetation at high-altitudes, at low elevations it is well known (you have also pointed it out in the Discussion) that the vegetation in the Iberian Peninsula is strongly influenced and controlled by the precipitation and moisture availability, and not by the winter temperature. The authors have pointed out a similar problem in the Discussion with the MTWA (Lines 260-268). Therefore, taking into account the issues related with the influence of the MTCO (and even MTWA) in the Iberian vegetation, at least at low-altitudes, the authors should consider whether including the MTCO is correct from a scientific point of view.

The authors have briefly compared their moisture reconstruction with speleothem records from the Iberian region (Lines 310-320). Since the study is based on pollen records, it would also be interesting to compare and discuss similarities/differences with other Iberian pollen records showing humidity changes throughout the Holocene period. In particular, comparing the results with other recent pollen-based quantitative moisture index or precipitation reconstructions from the Iberian Peninsula (e.g., Ilvonen et al., 2022) would show a more complete picture of the quantitative precipitation/humidity changes in the region and would definitely improve the Discussion section.

The authors have included a paragraph about the impact of $CO_2$ on plant physiological processes and how this affects the reconstructions. However, as the paper deals with Holocene records, they have not included any reconstruction that takes into account the effect of the $CO_2$. I agree with that, but then: what is the purpose of including this paragraph? The authors should avoid taking about a methodology that has not been used in this paper. I suggest removing this paragraph or, they could briefly explain that the variability of $CO_2$ during the Holocene is very low, and therefore, the effect of the $CO_2$ has not been taken into account for the moisture reconstructions (in contrast to other reconstructions, such as Wei et al., 2021).

I strongly recommend including the CCA plots (not only the numerical results as in the Table 2) in order to observe the relationship between the climatic variables and the pollen data.

The Variance Inflation Factor (VIF) analysis is used to demonstrate that the climatic variables are independent from each other. Your VIF results suggest that the collinearity between variables is not high, and therefore, variables seem to be independent from each other. However, the authors should better explained the VIF analysis. There is no single mention about the VIF analysis in Methods. In the current version of the paper, the methodology should be improved and these issues/questions should be clarified in the main text. Since the current version of the article is short, the authors could further clarify the methodology.

**Minor points**

Line 49: Add other references about new Iberian Holocene records related with changes in moisture conditions (e.g., Schröder et al., 2020; Ramos-Roman et al., 2018).

Lines 113-118: You should specify which taxa have been removed.

Line 131: "a modified code from SPLASH…"

Line 140: Remove the Doi number. Include the citation: Harrison et al. (2022)

Line 173: "The variance inflation factor (VIF) scores…"

Lines 173-174: Add reference.

**Figure 2:** Figure caption: use acronyms for "m above sea level", for example "m a.s.l.".

**Figure 4:** You must include the references for the insolation values. Where are insolation values taken from?

**Table 2:** The methodology related to this table needs to be better explained. For example, include in Methods the reason for using the VIF analysis.

**REFERENCES**

Harrison et al. (2022). Pollen data and charcoal data of the Iberian Peninsula (version 3). University of Reading. Dataset.

Ilvonen et al. (2022). Spatial and temporal patterns of Holocene precipitation change in the Iberian Peninsula. Boreas.

Liu et al. (2020). An improved statistical approach for reconstructing past climates from biotic assemblages. Proceeding of the royal Society.

Prentice et al. (2022). Accounting for atmospheric carbon dioxide variations in pollen-based reconstruction of past hydroclimates. Global and Planetary Change.

Ramos-Román et al. (2018). Holocene climate aridification trend and human impact interrupted by millennial- and centennial-scale climate fluctuations from a new sedimentary record from Padul (Sierra Nevada, southern Iberian Peninsula). Climate of the Past.

Reimer et al. (2020). The IntCal20 Northern Hemisphere radiocarbon age calibration curve (0-55 cal kBP). Radiocarbon.

Sánchez-Goñi et al. (2017). The ACER pollen and charcoal database: a global resource to document vegetation and fire response to abrupt climate changes during the last glacial period. Earth System Science Data.

Schröder et al. (2020). Unravelling the Holocene environmental history of south-western Iberia through a palynological study of Lake Medina sediments. Quaternary Science Reviews.

Telford et al. (2005). The secret assumption of transfer functions: problems with spatial autocorrelation in evaluating model performance. Quaternary Science Reviews.

Wei et al. (2020). Seasonal temperature and moisture changes in interior semi-arid Spain from the last interglacial to the Late Holocene. Quaternary Research.

Wei et al. (2021). Seasonal temperature and moisture changes in interior semi-arid Spain from the last interglacial to the Late Holocene. Quaternary Research.

---

## Referee Report (RR2)

Thank you for the opportunity to review the second draft of the manuscript entitled "Holocene climates of the Iberian Peninsula: pollen-based reconstructions of changes in the west-east gradient of temperature and moisture" by Mengmeng Liu and coauthors.

The text has been improved, especially the introduction and the parts on pollen data (modern and fossil) and method. It is much clearer. Thanks to the authors for that.

However for the figures, most of the changes I asked for were not taken into account (see below).

Introduction is better, papers are now cited (Tarroso et al); the paper of Davis et al., 2003 is still lacking.

 line 73: contradiction better than contra-distinction;

line 80 and in the text: update the ref Shen et al 2021  as Shen et al 2022 Clim. Past, 18, 1189–1201, https://doi.org/10.5194/cp-18-1189-2022, 2022

Lines 84-87: "These analyses allow us to confirm that the west-east gradient in moisture was less steep during the mid-Holocene and indicate the importance of changes in atmospheric circulation in explaining observed patterns of climate change across the region".

This sentence is a result, avoid it in the introduction

Line 78 :The term pollen and transfer function is required here; better as: Here, using pollen-inferred transfer functions,  we re-examine the trends in summer and winter temperature…

Methods

Line 92, could also add Salonen et al 2019?

Line 103: add a brief sentence on the recent RForest and BRT new methods (Salonen papers).

BRT is a nice and powerful tool to provide robust climate reconstructions

Results

Line 238: Mid-Holocene not Middle

Discussion

Line 300: add the ref for the transient output

Line 311: "The differences between the three data sets probably reflect differences in the number of records used, but the lack of coherency points to there not being a strong, regionally coherent signal of summer temperature changes during the Holocene".

I think that the differences are also probably linked to the method used (MAT with PFT for Mauri et al. and Davis et al, PDF for Tarroso et al. and improved WAPLS for your study). Please add a sentence on that.

What about the results from Davis et al., 2003? Did you compare with your results? I think their reconstruction of MTWA indicate cooler conditions in south west Mediterranean during mid-Holocene.

Line 412: human impact: this part is still too short. Human impact on pollen data is probably the most important problem on climate reconstruction during the mid to late Holocene. Even if archeological evidences are not found, human societies may influence vegetation for the Bronze age, especially in Mediterranean regions. So please, add more sentences on this topic.(Did you find NPP or specific pollen taxa related to human impact in your dataset). What about fires and its possible impact on vegetation?

Line 425; "Thus, the finding that winter temperatures are a direct reflection of insolation forcing whereas summer temperatures are influenced by land-surface feedbacks and changes in atmospheric circulation is robust to the method used."

I agree that results are close if you use WAPLS, but if you use another method (MAT, BRT, RF, Bayesien, ANN…), results could be strongly different, so please modify your sentence.

Conclusion: too short, you can improve it!

Figures

- **I already asked to include the synthesis figures (S8 and S9) in the text and not in supplementary material. This has not been done.**

**Most of the discussion is based on these figures: it must be included in the text.**

**I ask the editor to carefully check this point before acceptation of the manuscript.**

Tables

- **Pollen data must be better taken into account in the text.**

**I already kindly asked to include Table S1 in the text as table 1. The description of the data sources of fossil pollen used to reconstruct the climate in the Iberian Peninsula must be included directly here in the text and not in supplementary material.**

**This has not been done. I insist because we need to check easily the pollen sites, the chronological frame…**

**I ask the editor to carefully check this point before acceptation of the manuscript.**

---

## Referee Report (RR3)

**General observations :**
This paper presents in the same time two main goals which are maybe not sufficiently clearly announced in introduction and maybe not discussed in enough details in the results and discussion:

- **first**, a methodological improvement: the validation of not commonly used transfer function (both fx and tolerant-weighted improvement) presented in Liu et al., 2020 but still not clearly known and commonly tested in the climate reconstruction community. And, more than that, a new version of the fx correction (called fxTWA-PLS2) is presented. In conclusion it appears that this new version of the method is more reliable (especially in the climate extremes) and some supplementary figures show the same conclusion. However, this topic is not sufficiently and clearly presented in the text of the manuscript.

- **second**: the study present a new and more complete, detailed and qualitative climate reconstructions for the Holocene in the Iberian peninsula. Here, the discussion is focused on the west-east climatic gradient and his connection with orbital forcing.

It appears that this last version of the manuscript have really improved the discussion about the Holocene climate in the Iberian peninsula (lots of references added, comparison with other reconstructions, other proxies and other study cases, figures inserted in the main text...). However the methodological improvement of this paper is still not visible in the introduction and in the discussion as well.

**Major comments and modifications :**
As a specialist of climate reconstruction made on pollen samples, but not in this area of study, I will not focused on the Iberian Holocene climate discussion but mainly on the methodological discussion.

First, the last paragraph of the introduction should highlight the two main goals of the papers (methodological test and Holocene climate improvement such as explain in the general observations).

Then, the methodological part is far richer and clearer than the previous version of the manuscript with a larger and more exhaustive list of existing methods to convert the pollen signal to climate parameters. However, we do think that some point in the methodological choice should be clarified in the manuscript (maybe in introduction and certainly in discussion):

- **1.** How did you selected the studied climate parameters ? Why MTWA, MTCO and alpha instead of MAAT, MAP, GDD0, etc ? This is a important point.

- **2.** About the independence between climate parameters, it is also not really clear. The CCA and VIF show than the climate parameters are independent but in the same time you show Fig. 6 and l. 212-214 than they are closely correlated. This have also to be discussed.

- **3.** Why using 10° resolution climate database instead of already interpolated and discuss climate databases with 1° resolution (such as WorldClim2 or CHELSA data based) ?
All these choices have to be defended in introduction / methods.

About the results and discussion also some modifications are necessary. We think that the text, especially in the discussion is not sufficiently connecting with figures. Figs. 1, 5 and 6 are called only once and all the Figs. 1 to 6 are only called in results and not in discussions.

About the discussion, only the last paragraph of the discussion focus on the improvement made with the fxTWA-PLS2 version of the transfer function. The first sentence of the conclusion is "We have developed an improved version of fxTWA-PLS which further reduces compression bias and provides robust climate reconstructions", however this as not be proved neither discuss in the manuscript. We argue that this topic should be discussed and validated in the first part of the discussion before presenting the climate composite reconstruction and comparing it with other proxies, climate modeling and so on. Especially using the material in Appendix (Table A1 and Figs. A1 and A2).

---

## Author Response (AR2)

**Response to both reviewers**

Our responses are given in normal blue font and proposed changes to the text are given in *blue italics*. References to line numbers refer to the revised manuscript.

As suggested by the reviewers, we have moved two figures and one table from Supplementary to the main text, and renumbered the figures and tables accordingly.

**Response to Reviewer 1**

1. One of the main goals of the paper is to use the recently developed fxTWA-PLS method (Liu et al., 2020). Using this method provides new knowledge in the area of quantitative paleoclimate reconstructions, and therefore, it is one of the main points why the paper is worth publishing. However, since this is a new method and very few pollen records have used this methodology (as far as I know: Liu et al., 2020; Wei et al., 2020; Prentice et al., 2022), more information should be included. For example, the authors mention that the WAPLS suffers from the tendency to compress the reconstructions towards the central part of the sampled climate range (Line 87-88) but they do not mention that the WAPLS is robust to spatial autocorrelation (Telford et al., 2005). Has this method been demonstrated to be robust to spatial autocorrelation?

Since fxTWA-PLS is a modification to WA-PLS, it is robust to spatial autocorrelation as we showed in our previous paper. Furthermore, the cross-validation excludes spatially correlated pseudo sites. We agree that we should make this clearer in the text. We have modified the text at line 111-113 in the revised text without tracked changes as follows:

*The modified version further reduces the biases at the extremes of the sampled climate range, while retaining the desirable properties of WA-PLS in terms of robustness to spatial autocorrelation (fxTWA-PLS: Liu et al., 2020).*

2. With respect to fossil pollen records, the authors did a great work compiling a large number of sites, using different data sources such as the European Pollen Database (EPD), Pangaea and even contacting directly with the authors from the original studies ("Author" in the excel file "Iberia_pollen_records_v3_0307.xlsx"). However, the authors should have also checked the relatively new and opened ACER database (Sánchez-Goñi et al., 2017), which provides high-resolution global-scale fossil pollen data. Did the authors check the high-resolution Spanish pollen records from this database? In addition, other new high-resolution Holocene Iberian pollen records should have also been included in the paper, such as lake Medina or Padul (both in southern Spain), especially when the number of fossil pollen records in the southern Iberian Peninsula is not as good as in the north, which could lead to uncertainties in the reconstructions and interpretations (as you mentioned in Lines 350-354).

The ACER database focuses on records covering D-O cycle and some of the records included are truncated before the Holocene, our period of interest. The database contains 3 terrestrial sites from Iberia: Navarres, Abric Romani, Lake Banyoles. Navarres and Banyoles are both included in our compilation. Abric Romani is an archaeological site and not suitable for the purpose of climate reconstruction. The new high-resolution data from Padul is not publicly available as the author is still working on these records. The Medina record has numerous hiatuses and we were advised not to use this record by one of the authors of the 2020 paper. This is also, presumably, the reason that Medina was only used for lake level reconstructions (and not for climate reconstructions) by Ilvonen et al. (2022). Thus, our data set is currently the most comprehensive one available for Iberia.

3. As we know, the pollen-based quantitative reconstructions are controlled by the modern pollen training dataset, and thus by the modern pollen vegetation. Although the MTCO reconstructions appear to be consistent with climate model simulations and consequence of insolation forcing (Lines 233-237), I wonder whether the MTCO reconstructions are completely reliable. Looking at the statistical performance of the MTCO ($R^2$ 0.75 for Component 4, Table 1), the MTCO reconstructions should be reliable. However, the reconstructed MTCO anomalies at individual sites (Fig. 3) do not show a clear E-W trend (neither at low nor high altitude). This seems to be a consequence of the recent MTCO variability over a NW-SE (or E-W) transect (Figure 2), which shows no clear E-W changes in temperature. Actually, the MTCO in Figure 2 seems to be related with elevation changes in Iberia, with the lowest temperatures in Pyrenees, Central and Iberian Mountain chains, etc. Although the winter temperature can influence the vegetation at high-altitudes, at low elevations it is well known (you have also pointed it out in the Discussion) that the vegetation in the Iberian Peninsula is strongly influenced and controlled by the precipitation and moisture availability, and not by the winter temperature. The authors have pointed out a similar problem in the Discussion with the MTWA (Lines 260-268). Therefore, taking into account the issues related with the influence of the MTCO (and even MTWA) in the Iberian vegetation, at least at low-altitudes, the authors should consider whether including the MTCO is correct from a scientific point of view.

We think that we may have confused the reviewer by stating that there is no spatial differentiation between western and eastern Iberia in the changes in winter temperature (line 182-184 in the first revision without tracked changes). We do not expect MTCO to show a west-east trend. Winter temperatures today generally increase from north to south, although are also affected by elevation. Changing winter insolation over the Holocene should not affect this pattern. Although, the vegetation of Iberia is strongly influenced by moisture availability, nevertheless winter temperature affects the distribution of woody species because of the necessity to have physiological mechanisms to resist frost damage which could have been important in the early Holocene. Given that we obtain a reasonable $R^2$ for MTCO, as pointed out by the reviewer, and that the VIFs for our reconstruction show that our reconstructions of all three climate variables are independent, we feel that it is perfectly correct from a scientific point of view to include MTCO here. However, we have modified the text to ensure that other readers are not confused by our statement about the lack of east-west differences as follows:

*The composite curve also shows a general increase in winter temperatures through time (Fig. 4a), consistent with the trend in winter insolation (Fig. 4d). The composite curve shows that it was ca 4°C cooler than today at 11.5 ka and conditions remained cooler than present until ca 2.5 ka. Winter temperatures today increase from north to south and are also affected by elevation; these patterns are still present in the Holocene reconstructions, but there is no spatial differentiation between western and eastern Iberia in the anomalies (Table 4, SI Fig. S2). The similarity of the changes compared to present geographically is consistent with the idea that the changes in winter temperature are driven by changes in winter insolation.*

4. The authors have briefly compared their moisture reconstruction with speleothem records from the Iberian region (Lines 310-320). Since the study is based on pollen records, it would also be interesting to compare and discuss similarities/differences with other Iberian pollen records showing humidity changes throughout the Holocene period. In particular, comparing the results with other recent pollen-based quantitative moisture index or precipitation reconstructions from the Iberian Peninsula (e.g., Ilvonen et al., 2022) would show a more complete picture of the quantitative precipitation/humidity changes in the region and would definitely improve the Discussion section.

Ilvonen et al. (2022) provide pollen-based reconstructions of mean annual and summer and winter precipitation from 8 sites in Iberia, using WAPLS and a Bayesian modelling approach. The training

data set that is confined to samples from Spain, which may explain the somewhat limited variability shown by these records for most of the Holocene. They draw attention to a distinction between northern sites and southern sites in terms of hydroclimate trends but they do not specifically address the idea that the west-east gradient in moisture was reduced. Nevertheless, the increased precipitation during the early to mid-Holocene that they claim is characteristic of both northern and southern sites is largely caused by increases in the sites in the eastern part of Iberia, specifically San Rafael, Navarres, and Qintanar. With the exception of Monte Areo (which shows an increase for a short period), the sites in the western part of the peninsula either show little change or in some cases a decrease in precipitation (Zalamar, El Maillo). There are only a few other pollen-based reconstructions of moisture changes from Iberia, specifically precipitation reconstructions from Padul and Villarquemado (Camuera et al., 2022, also Garcia-Alex et al., 2021) and moisture index from Villarquemada (Wei et al., 2021). These sites are also in the east of the peninsula, thus the increased moisture recorded there during the early to mid-Holocene is also consistent with our reconstructions. Although it is difficult to compare changes in precipitation directly with changes in plant-available moisture, we have added a brief description of these reconstructions and pointed out they support our conclusion of a reduced west-east gradient during the early to mid-Holocene as follows:

*There are comparatively few pollen-based reconstructions of moisture changes during the Holocene from Iberia. Records from Padul show increased mean annual and winter precipitation during the early and mid-Holocene (Garcia-Alex et al., 2021; Camuera et al., 2022). Reconstructions of mean annual and winter precipitation (Camuera et al., 2022) and the ratio of annual precipitation to annual potential evapotranspiration (Wei et al., 2021) also show wetter conditions at this time at El Cañizar de Villarquemado. Both of these sites lie in the eastern part of the Iberian Peninsula, so these reconstructions are consistent with our interpretation of wetter conditions in this region during the interval between 9.5 and 3.5 ka. Ilvonen et al. (2022) provide pollen-based reconstructions of mean annual, summer and winter precipitation from 8 sites in Iberia, using WAPLS and a Bayesian modelling approach. Although they focus on the contrasting pattern of hydroclimate evolution between northern and southern Iberia, the three easternmost sites (San Rafael, Navarres, and Qintanar de la Sierra) show much wetter conditions during the early to mid-Holocene. With the exception of the record from Monte Areo, the records from further west are relatively complacent and indeed two sites (Zalamar, El Maillo) show decreased precipitation between 8 and 4 ka. Thus, these records are consistent with our interpretation that the west-east gradient of moisture was reduced between 9.5 and 4.5 ka.*
> > *Speleothem oxygen-isotope data from the Iberian Peninsula also provide support for our pollen-based reconstructions of changes in the west-east gradient of moisture through the Holocene. .......*

We have added the following references:

*Camuera, J., Ramos-Román, M. J., Jiménez-Moreno, G., García-Alix, A., Ilvonen, L., Ruha, L., Gil-Romera, G., González-Sampériz, P. and Seppä, H.: Past 200 kyr hydroclimate variability in the western Mediterranean and its connection to the African Humid Periods, Sci. Rep., 12(1), 9050, doi:10.1038/s41598-022-12047-1, 2022.*

*García-Alix, A., Camuera, J., Ramos-Román, M. J., Toney, J. L., Sachse, D., Schefuß, E., Jiménez-Moreno, G., Jiménez-Espejo, F. J., López-Avilés, A., Anderson, R. S. and Yanes, Y.: Paleohydrological dynamics in the Western Mediterranean during the last glacial cycle, Glob. Planet. Change, 202, 103527, doi:https://doi.org/10.1016/j.gloplacha.2021.103527, 2021.*

*Ilvonen, L., López-Sáez, J. A., Holmström, L., Alba-Sánchez, F., Pérez-Díaz, S., Carrión, J. S.,*

*Ramos-Román, M. J., Camuera, J., Jiménez-Moreno, G., Ruha, L. and Seppä, H.: Spatial and temporal patterns of Holocene precipitation change in the Iberian Peninsula, Boreas, doi:https://doi.org/10.1111/bor.12586, 2022.*

5. The authors have included a paragraph about the impact of CO2 on plant physiological processes and how this affects the reconstructions. However, as the paper deals with Holocene records, they have not included any reconstruction that takes into account the effect of the CO2. I agree with that, but then: what is the purpose of including this paragraph? The authors should avoid taking about a methodology that has not been used in this paper. I suggest removing this paragraph or, they could briefly explain that the variability of CO2 during the Holocene is very low, and therefore, the effect of the CO2 has not been taken into account for the moisture reconstructions (in contrast to other reconstructions, such as Wei et al., 2021).

We included this paragraph because the direct impacts of changing $CO_2$ are significant on longer timescales, as we have shown, and we wanted to make clear why it is unnecessary to do this in the current case. Specifically, as we state (line 322-334 in the first revision without tracked changes) the impact of the relatively small change in $CO_2$ over the Holocene is within the uncertainties of our reconstructions. However, we agree with the reviewer that this paragraph is long and rather belabours the point, so we have rewritten it more succinctly as follows:

*Pollen data are widely used for the quantitative reconstruction of past climates (see discussion in Bartlein et al., 2011), but reconstructions of moisture indices are also affected by changes in water-use efficiency caused by the impact of changing atmospheric $CO_2$ levels on plant physiology (Farquhar, 1997; Gerhart and Ward, 2010; Prentice et al., 2017; Prentice and Harrison, 2009). This has been shown to be important on glacial-interglacial timescales, when intervals of lower-than-present $CO_2$ result in vegetation appearing to reflect drier conditions than were experienced in reality (Prentice et al., 2011, 2017; Wei et al., 2021). We do not account for this $CO_2$ effect in our reconstructions of α because the change in $CO_2$ over the Holocene was only 40 ppm. This change relative to modern levels has only a small impact on the reconstructions (Prentice et al., 2022) and is sufficiently small to be within the reconstruction uncertainties. Furthermore, accounting for changes in $CO_2$ would not affect the reconstructed west-east gradient through time.*

6. I strongly recommend including the CCA plots (not only the numerical results as in the Table 2) in order to observe the relationship between the climatic variables and the pollen data. The Variance Inflation Factor (VIF) analysis is used to demonstrate that the climatic variables are independent from each other. Your VIF results suggest that the collinearity between variables is not high, and therefore, variables seem to be independent from each other. However, the authors should better explain the VIF analysis. There is no single mention about the VIF analysis in Methods. In the current version of the paper, the methodology should be improved and these issues/questions should be clarified in the main text. Since the current version of the article is short, the authors could further clarify the methodology.

We have now included the CCA plots in Supplementary Materials (Fig. S11) and an explanation of the VIF analysis, at line 161-164 in the revised text without tracked changes, as follows:

*Variance inflation factor (VIF) scores are calculated for both the modern climates and the climates reconstructed from fossil pollen records, in order to avoid multicollinearity problems and thus guarantee the climate variables (MTCO, MTWA, α) used here represent independent features of the pollen records.*

7. Line 49: Add other references about new Iberian Holocene records related with changes in moisture conditions (e.g., Schröder et al., 2020; Ramos-Roman et al., 2018).

We have now added the references:

*However, much of the evidence for Holocene climates of the Iberian Peninsula is based on qualitative interpretations of vegetation changes, generally interpreted as reflecting changes in moisture availability (Morellón et al., 2018; Ramos-Román et al., 2018; Schröder et al., 2019).*

coherent signal of summer temperature changes during the Holocene". I think that the differences are also probably linked to the method used (MAT with PFT for Mauri et al. and Davis et al, PDF for Tarroso et al. and improved WAPLS for your study). Please add a sentence on that.

It is possible that the differences are related to the methods used, but we cannot demonstrate this, whereas it is clear that there are differences and spatial biases in the number of records used. The key point here is that whereas there is broad agreement between different data sets about the changes in winter temperature, there is little agreement in terms of summer temperature. We argue that this reflects the absence of a strong, regionally coherent signal of summer temperature changes. We have modified the sentence to make the point of our argument clearer as follows:

*The differences between the three data sets could reflect differences in the reconstruction methods, or differences in the number of records used and in the geographic sampling. However, given the fact that all three data sets show similar trends in winter temperature, the lack of coherency between the data sets for MTWA points to there not being a strong, regionally coherent signal of summer temperature changes during the Holocene.*

What about the results from Davis et al., 2003? Did you compare with your results? I think their reconstruction of MTWA indicate cooler conditions in south west Mediterranean during mid-Holocene.

We did not explicitly compare our results to Davis et al. (2003) because the Mauri et al. (2015) paper is an update of Davis et al. (2003), and as Mauri et al. (2015) explicitly states it was made "using the same methodology, but with a greatly expanded fossil and surface-sample dataset and more rigorous quality-control". Thus, they argue that the newer results are more reliable than the earlier work.

11. Line 412: human impact: this part is still too short. Human impact on pollen data is probably the most important problem on climate reconstruction during the mid to late Holocene. Even if archeological evidences are not found, human societies may influence vegetation for the Bronze age, especially in Mediterranean regions. So please, add more sentences on this topic. (Did you find NPP or specific pollen taxa related to human impact in your dataset). What about fires and its possible impact on vegetation?

The degree of human influence on land-use in Iberia is still a matter of debate, with some authors arguing for evidence of human influence on the vegetation at particular sites and no evidence at others. Archaeological evidence indicates that the Neolithic transition in Iberia was non-synchronous across the region and occurred over a period of several thousand years, which makes it unlikely that anthropogenic signals would produce coherent patterns of vegetation change through time. There is also an issue about the methods used to identify anthropogenic land-use change. The presence of fungal spores associated with animal faeces has been used to identify the presence of domesticated animals at individual archaeological sites in but can also indicate non-domesticates and has not been investigated systematically across multiple sites. Evidence of land-use changes based on increases in weed plants, which has also been used to identify human impact in particular sites in Iberia, are unreliable because they can also reflect non-anthropogenic disturbance. The most reliable evidence of anthropogenic land use changes is the presence of cereals. However, with the exception of rye which is wind pollinated, most cereals do not release pollen until they are threshed and this means that they are generally poorly represented in pollen diagrams. Indeed, the latest reconstructions of changes in the abundance of cereals across Europe (Githumbi et al., 2022) shows that it was not until ca 1000 years ago that cereal pollen was present in more sites in the Iberian Peninsula than absent. Change in fire regime have also been attributed to human activities in Iberia. However, our recent analyses of fire history across Iberia (Sweeney et al., 2022) shows no relationship between the timing of first agriculture in different parts of the Peninsula and changes in fire regime, nor between intervals of rapid population growth and fire. It is clear that a more

thorough analysis of the evidence for human impacts on vegetation, based on a comprehensive analysis of various sources of data, is required in order to provide a better understanding of how this might impact the pollen-based climate reconstructions for Iberia, but this is beyond the scope of the present paper. Nevertheless, we are happy to provide more discussion of this issue and have expanded the text in the Discussions as follows:

[revised manuscript text omitted]

12. Line 425; "Thus, the finding that winter temperatures are a direct reflection of insolation forcing whereas summer temperatures are influenced by land-surface feedbacks and changes in atmospheric circulation is robust to the method used." I agree that results are close if you use WAPLS, but if you use another method (MAT, BRT, RF, Bayesien, ANN…), results could be strongly different, so please modify your sentence.
We agree that the results might be different if we used non-regression based techniques, as indeed previous papers comparing WAPLS and other types of reconstruction method (e.g. Ilvonen et al., 2022) have shown. Here we are simply arguing that our improvements of fxTWA-PLS produce a better model but do not change our conclusions. We have modified the sentence as follows:

*Thus, the finding that winter temperatures are a direct reflection of insolation forcing whereas summer temperatures are influenced by land-surface feedbacks and changes in atmospheric circulation is robust to the version of fxTWA-PLS used.*

13. Conclusion: too short, you can improve it!
We tried to keep the conclusion short to emphasise the key findings about the Holocene climate of Iberia. However, we are happy to expand this and have rewritten the text as follows:

*We have developed an improved version of fxWA-PLS which further reduces compression bias and provides robust climate reconstructions. We have used this technique with a large pollen data set representing 117 sites across the Iberian Peninsula to make quantitative reconstructions of summer and winter temperature and an index of plant-available moisture through the Holocene. We show that there was a gradual increase in winter temperature through the Holocene and that this trend*

*broadly follows the changes in orbital forcing. Summer temperatures, however, do not follow the changes in orbital forcing but appear to be influenced by land-surface feedbacks associated with changes in moisture. We show that the west-east gradient in moisture was considerably less pronounced during the mid-Holocene (8~4 ka), implying a significant increase in moisture advection into the continental interior resulting from changes in circulation. Our reconstructions of temperature changes are broadly consistent with previous reconstructions, but are more solidly based because of the increased site coverage. Our reconstructions of changes in the west-east gradient of moisture during the early part of the Holocene are also consistent with previous reconstructions, although this change is not simulated by state-of-the-art climate models, implying that there are still issues to resolve the associated land-surface feedbacks in these models. Our work provides an improved foundation for documenting and understanding the Holocene palaeoclimates of Iberia.*

14. For figures, I already asked to include the synthesis figures (S8 and S9) in the text and not in supplementary material. This has not been done. Most of the discussion is based on these figures: it must be included in the text. I ask the editor to carefully check this point before acceptation of the manuscript.

In our previous response we explained that we did not think these figures were central to the paper since our goal was to compare the reconstructions rather than the methodologies. We were concerned that, as we already have 9 display items, adding two new figures and a new table (see C15 below) would be rather too much. However, since the editor seems to be happy to have 12 display items, we have now moved figure S8 and S9 into the main text and numbered them accordingly. Figure S8 and S9 are now Figure 7 and 8 in the revised version.

15. For tables, pollen data must be better taken into account in the text. I already kindly asked to include Table S1 in the text as table 1. The description of the data sources of fossil pollen used to reconstruct the climate in the Iberian Peninsula must be included directly here in the text and not in supplementary material. This has not been done. I insist because we need to check easily the pollen sites, the chronological frame… I ask the editor to carefully check this point before acceptation of the manuscript.

We have now moved Table S1 into the main text and renumbered the Tables accordingly. Table S1 is now Table 1 in the revised version.

---

## Author Response (AR3)

**Response to reviewer**

Our responses are given in blue and proposed changes to the text in *blue italics*. References to line numbers refer to the revised manuscript.

1. The last paragraph of the introduction should highlight the two main goals of the papers (methodological test and Holocene climate improvement such as explain in the general observations).

The methodological test was originally presented as a minor aspect of the paper for a good reason, namely that the modification made is relatively slight. It concerns the algorithm used to implement the sampling frequency correction, without changing the underlying principles of the method as presented (and extensively tested) by Liu et al. (2020). The modification does nonetheless produce a further improvement in reconstruction accuracy, which we agree is worth greater emphasis. We have accordingly modified the last paragraph of the Introduction as follows:

*We used the method Tolerance-weighted Weighted Average Partial Least-Squares regression with a sampling frequency correction (fxTWA-PLS), introduced by Liu et al. (2020) as an improvement of the widely used Weighted Average Partial Least-Squares (WAPLS: ter Braak and Juggins, 1993) method for reconstructing past climates from pollen assemblages. As presented in depth by Liu et al. (2020), this method is a more complete implementation of the theory underlying WA-PLS because it takes greater account of the climatic information provided by taxa with more limited climatic ranges and also applies the sampling frequency correction to reduce the impact of uneven sampling in the training data set. Liu et al. (2020) showed that fxTWA-PLS does indeed provide better reconstructions than WA-PLS.*

*Here we have further modified the algorithm implementing fxTWA-PLS, achieving an additional gain in performance. In the algorithm as published by Liu et al. (2020), sampling frequencies were extracted from a histogram. In the modified algorithm they are estimated using P-splines smoothing (Eilers and Marx, 2021), which makes the estimates almost independent on the chosen bin width (see Appendix A for details). In addition, the modified method applies the sampling frequency correction at two separate steps – the estimation of optima and tolerances, and the regression step – a measure intended to produce more stable results. Indeed, the modified method produces both improved $R^2$ values and reduced compression and maximum bias in reconstructed climate variables (see Table A1 and Figs A1–A2). We will return to this point in the Discussion.*

*We have used this improved method to reconstruct Holocene climates across Iberia, and re-examined the trends in summer and winter temperature and plant-available moisture, using a new and*

*relatively comprehensive compilation of pollen data (Shen et al., 2022) with age models based on the latest radiocarbon calibration curve (IntCal20: Reimer et al., 2020)...*

2. The methodological part is far richer and clearer than the previous version of the manuscript with a larger and more exhaustive list of existing methods to convert the pollen signal to climate parameters. However, we do think that some point in the methodological choice should be clarified in the manuscript (maybe in introduction and certainly in discussion) and all these choices have to be defended in introduction / methods:

2.1. How did you select the studied climate parameters? Why MTWA, MTCO and alpha instead of MAAT, MAP, GDD$_0$, etc? This is an important point.

We have modified the Methods as follows:

*There are no generally accepted rules as to the choice of variables for palaeoclimate reconstruction. No systematic comparison of these choices has been made. However, it is widely understood that plant taxon distributions reflect distinct, largely independent controls by summer temperatures, winter temperatures, and moisture availability (see e.g. Harrison et al., 2010). Therefore, in common with many other studies (Cheddadi et al., 1997; Jiang et al., 2010; Peyron et al., 1998; Wei et al., 2021; Zhang et al., 2007), we have chosen bioclimatic variables that reflect these independent controls, with mean temperature of the coldest month (MTCO) to represent winter temperatures, mean temperature of the warmest month (MTWA) to represent summer temperatures and α, an estimate of the ratio of actual evapotranspiration to equilibrium evapotranspiration, to represent plant-available moisture. We choose not to use mean annual air temperature (MAAT) because it is a composite of summer and winter conditions; and we prefer to use an index of effective moisture availability (our estimate of α being one such index) to mean annual precipitation (MAP), whose significance for plant function depends strongly on potential evaporation (a function of temperature and net radiation). Our calculation of α takes account of this dependence. Growing degree days above a baseline of 0 °C (GDD$_0$) would be a possible alternative to MTWA as an expression of summer conditions but is most relevant as a predictor of "cold limits" of trees in cool climates, whereas MTWA better reflects the high-temperature stress on plants in Mediterranean-type climates.*

2.2. About the independence between climate parameters, it is also not really clear. The CCA and VIF show than the climate parameters are independent but in the same time you show Fig. 6 and l. 212-214 than they are closely correlated. This have also to be discussed.

The CCA and the VIFs do indeed show that there is sufficient independent information in pollen data to allow all three variables to be reconstructed. In other words, if their true values varied independently in the past, we would expect these variations to be manifested in the fossil pollen data – and we would expect the reconstruction to reveal them. VIFs are the standard metric to assess such independence. Acceptable VIFs can be achieved even if there is some correlation between variables in the training set (although very high correlations would be problematic, leading to high VIFs).

Fig. 6 shows a quite different point. It shows that there was, in fact, a high correlation between the variations of α and MTWA in this region during the period studied (panel b, which reflects both spatial and temporal correlation)– which we can then interpret in terms of potential climatic mechanisms. Panel a only reflects spatial correlation and the correlation is acceptable to reflect independent information, showing that the relationship in past reconstructed values of the two variables is not an artefact of correlations existing in the training data set.

We have modified text as follows:

*The variance inflation factor (VIF) scores are all less than 6, so there are no multicollinearity problems (Table 3) (Allison, 1994), making it possible to independently reconstruct all three climate variables based on pollen data.*

2.3. Why using 10° resolution climate database instead of already interpolated and discuss climate databases with 1° resolution (such as WorldClim2 or CHELSA data based)?

We obtained the climates for each site using geographically weighted regression (GWR), allowing a built-in correction for site elevation. By far the most important difference between CRU and the more recent, higher-resolution data sets is simply that they account for elevation with much finer resolution. However, even 1 km resolution is not necessarily sufficient to pinpoint the climate of a site; GWR would still be needed. Moreover, other work in Sandy Harrison's group has confirmed that so long as GWR is used, the use of higher-resolution data results in only minimal changes to the interpolated site climates.

We have modified text as follows:

*The climate at each site was obtained using geographically weighted regression (GWR) of the CRU CL v2.0 gridded dataset of modern (1961-1990) surface climate at 10 arc minute resolution (New et al., 2002) in order to (a) correct for elevation differences between each pollen site and the*

*corresponding grid cell and (b) make the resulting climate independent of the resolution of the underlying data set.*

3. About the results and discussion also some modifications are necessary. We think that the text, especially in the discussion is not sufficiently connecting with figures. Figs. 1, 5 and 6 are called only once and all the Figs. 1 to 6 are only called in results and not in discussions.

We intended to separate the result and discussions, now we have included more reference to them at line 278, 279, 305, 343, 344, 349, 353, 354 at the revised manuscript without tracked changes.

4. About the discussion, only the last paragraph of the discussion focuses on the improvement made with the fxTWA-PLS2 version of the transfer function. The first sentence of the conclusion is "We have developed an improved version of fxTWA-PLS which further reduces compression bias and provides robust climate reconstructions", however this as not be proved neither discuss in the manuscript. We argue that this topic should be discussed and validated in the first part of the discussion before presenting the climate composite reconstruction and comparing it with other proxies, climate modeling and so on. Especially using the material in Appendix (Table A1 and Figs. A1 and A2).

We have now modified the beginning of the Discussion as follows, in response to this request.

*The modified version of fxTWA-PLS (fxTWA-PLS2) (Table 2, Table A1) shows a few differences compared to the previous version (fxTWA-PLS1). Cross-validation $R^2$ values are higher for MTCO and MTWA, and almost unchanged for α. The maximum bias shows a decrease for all the three variables, especially for MTCO. The compression problem is also reduced for MTCO ($b_1$ increases from 0.82 to 0.91) and MTWA ($b_1$ increases from 0.69 to 0.71) while remaining roughly the same for α. The overall performance statistics thus show substantial improvements for MTCO and MTWA, while they show little change for α. However, Figure A1 shows that "unphysical" reconstructions beyond the natural limits of α (0–1.26) are greatly reduced, especially for the lower limit. There are also fewer outliers in Figure A1 and A2 for all three variables. Thus overall, the modified version further reduces the reconstruction biases, especially at the extremes of the sampled climate range. This improvement probably occurs because of the separate application of 1/fx correction during both the calculation of optima and tolerances of taxa and during the regression step – instead of applying an overall weight of $1/fx^2$ at the regression step, which can result in some extreme values (with low sampling frequency) being weighed too strongly and appearing as outliers.*

*fxTWA-PLS2 reconstructed climates have shown that there was a gradual increase in …*